# Stochastic Distributed Optimization under Average Second-order Similarity: Algorithms and Analysis

**Dachao Lin** [*][†]          **Yuze Han** [*][‡]          **Haishan Ye** [§]          **Zhihua Zhang** [¶]

## Abstract

We study finite-sum distributed optimization problems involving a master node and $n-1$ local nodes under the popular $\delta$-similarity and $\mu$-strong convexity conditions. We propose two new algorithms, SVRS and AccSVRS, motivated by previous works. The non-accelerated SVRS method combines the techniques of gradient sliding and variance reduction and achieves a better communication complexity of $\tilde{\mathcal{O}}(n+\sqrt{n}\delta/\mu)$ compared to existing non-accelerated algorithms. Applying the framework proposed in Katyusha X [6], we also develop a directly accelerated version named AccSVRS with the $\tilde{\mathcal{O}}(n+n^{3/4}\sqrt{\delta/\mu})$ communication complexity. In contrast to existing results, our complexity bounds are entirely smoothness-free and exhibit superiority in ill-conditioned cases. Furthermore, we establish a nearly matched lower bound to verify the tightness of our AccSVRS method.

## 1   Introduction

We have witnessed the development of distributed optimization in recent years. Distributed optimization aims to cooperatively solve a learning task over a predefined social network by exchanging information exclusively with immediate neighbors. This class of problems has found extensive applications in various fields, including machine learning, healthcare, network information processing, telecommunications, manufacturing, natural language processing tasks, and multi-agent control [54, 30, 48, 45, 60, 8]. In this paper, we focus on the following classical finite-sum optimization problem in a centralized setting:

$$\min_{\boldsymbol{x} \in \mathbb{R}^d} f(\boldsymbol{x}) := \frac{1}{n} \sum_{i=1}^{n} f_i(\boldsymbol{x}), \tag{1}$$

where each $f_i$ is differentiable and corresponds to a client or node, and the target objective is their average function $f$. Without loss of generality, we assume $f_1$ is the master node and the others are local nodes. In each round, every local node can communicate with the master node certain information, such as the local parameter $\boldsymbol{x}$, local gradient $\nabla f_i(\boldsymbol{x})$, and some global information gathered at the master node. Such a scheme can also be viewed as decentralized optimization over a star network [55].

Following the wisdom of statistical similarity residing in the data at different nodes, many previous works study scenarios where the individual functions exhibit relationships or, more specifically, certain homogeneity shared among the local $f_i$'s and $f$. The most common one is under the $\delta$-second-order

---

[*]Equal Contribution.

[†]Academy for Advanced Interdisciplinary Studies; Peking University; `lindachao@pku.edu.cn`;

[‡]School of Mathematical Sciences; Peking University; `hanyuze97@pku.edu.cn`;

[§]Corresponding Author; School of Management; Xi'an Jiaotong University; SGIT AI Lab, State Grid Corporation of China; `yehaishan@xjtu.edu.cn`;

[¶]School of Mathematical Sciences; Peking University; `zhzhang@math.pku.edu.cn`.

37th Conference on Neural Information Processing Systems (NeurIPS 2023).

similarity assumption [33, 35], that is,

$$\left\|\nabla^2 f_i(\boldsymbol{x}) - \nabla^2 f(\boldsymbol{x})\right\| \le \delta, \forall \boldsymbol{x} \in \mathbb{R}^d, i \in \{1, \dots, n\}.$$

Such an assumption also has different names in the literature, such as $\delta$-related assumption, bounded Hessian dissimilarity, or function similarity [7, 32, 51, 54, 62]. The rigorous definitions are deferred to Section 2. Moreover, the second-order similarity assumption can hold with a relatively small $\delta$ compared to the smoothness coefficient of $f_i$'s in many practical settings, such as statistical learning. More insights on this can be found in the discussion presented in [54, Section 2]. The similarity assumption indicates that the data across different clients share common information on the second-order derivative, potentially leading to a reduction in communication among clients. Meanwhile, the cost of communication is often much higher than that of local computation in distributed optimization settings [9, 44, 30]. Hence, researchers are motivated to develop efficient algorithms characterized by low communication complexity, which is the primary objective of this paper as well.

Furthermore, we need to emphasize that prior research [25, 56, 63, 19, 43, 5] has shown tightly matched lower and upper bounds on computation complexity for the finite-sum objective in Eq. (1). These works focus on gradient complexity under (average) smoothness [63] instead of communication complexity under similarity. Indeed, we will also discuss and compare the gradient complexity as shown in [35], to explore the trade-off between communication and gradient complexity.

Although the development of distributed optimization with similarity has lasted for years, the optimal complexity under full participation was only recently achieved by Kovalev et al. [35]. They employed gradient-sliding [37] and obtained the optimal communication complexity $\tilde{\mathcal{O}}(n\sqrt{\delta/\mu})$ for $\mu$-strongly convex $f$ and $\delta$-related $f_i$'s in Eq. (1). However, the full participation model requires the calculation of the whole gradient $\nabla f(\cdot)$, which incurs a communication cost of $n-1$ in each round. In contrast, partial participation could reduce the communication burden and yield improved complexity. Hence, Khaled and Jin [33] introduced client sampling, a technique that selects one client for updating in each round. They developed a non-accelerated algorithm SVRP, which achieves the communication complexity of $\tilde{\mathcal{O}}(n+\delta^2/\mu^2)$. Additionally, they proposed a Catalyzed version of SVRP with the complexity $\tilde{\mathcal{O}}(n+n^{3/4}\sqrt{\delta/\mu})$, which is better than the rates obtained in the full participation setting.

We believe there are several potential avenues for improvement inspired by [33]. 1) Khaled and Jin [33] introduced the requirement that each individual function is strongly convex (see [33, Assumption 2]). However, this constraint is absent in prior works. Notably, in the context of full participation, even non-convexity is deemed acceptable[6]. A prominent example is the shift-and-invert approach to solving PCA [52, 23], where each component is smooth and non-convex, but the average function remains convex. Thus we doubt the necessity of requiring strong convexity for individual components. 2) In hindsight, it seems that the directly accelerated SVRP could only achieve a bound of $\tilde{\mathcal{O}}(n + \sqrt{n}\cdot\delta/\mu)$ based on the current analysis, which is far from being satisfactory compared to its Catalyzed version. Consequently, there might be room for the development of a more effective algorithm for direct acceleration. 3) It is essential to note that the Catalyst framework introduces an additional log term in the overall complexity, along with the challenge of parameter tuning. This aspect is discussed in detail in [6, Section 1.2]. Therefore, we intend to address the aforementioned concerns, particularly on designing directly accelerated methods under the second-order similarity assumption.

## 1.1 Main Contributions

In this paper, we address the above concerns under the average similarity condition. Our contributions are presented in detail below and we provide a comparison with previous works in Table 1:

- First, we combine gradient sliding and client sampling techniques to develop an improved non-accelerated algorithm named SVRS (Algorithms 1). SVRS achieves a communication complexity of $\tilde{\mathcal{O}}(n + \sqrt{n}\cdot\delta/\mu)$, surpassing SVRP in ill-conditioned cases. Notably, this rate does not need component strong convexity and applies to the function value gap instead of the parameter distance.

- Second, building on SVRS, we employ a classical interpolation framework motivated by Katyusha X [6] to introduce the directly accelerated SVRS (AccSVRS, Algorithm 2).

---

[6]Readers can check that the proof of [35] only requires $f_1(\cdot) + \frac{1}{2\theta}\left\|\cdot\right\|^2$ is strongly convex, which can be guaranteed by $\delta$-second-order similarity since $f$ is $\mu$-strongly convex and $\theta = 1/(2\delta)$ therein.

Table 1: Comparison of communication under similarity for the strongly convex objective.

| | Method/Reference | Communication complexity | Assumptions |
|---|---|---|---|
| **No Sampling** | AccExtragradient [35] | $\mathcal{O}\left(n\sqrt{\frac{\delta}{\mu}}\log\frac{1}{\varepsilon}\right)$ | SS only for $f_1$ |
| | Lower bound [7] | $\Omega\left(n\sqrt{\frac{\delta}{\mu}}\log\frac{1}{\varepsilon}\right)$ | SS for $f_i$'s |
| **Client Sampling** | SVRP [33] | $\mathcal{O}\left(\left(n+\frac{\delta^2}{\mu^2}\right)\log\frac{1}{\varepsilon}\right)^{(1)}$ | SC for $f_i$'s, AveSS |
| | Catalyzed SVRP [33] | $\mathcal{O}\left(\left(n+n^{3/4}\sqrt{\frac{\delta}{\mu}}\right)\log\frac{1}{\varepsilon}\log\frac{L}{\mu}\right)^{(2)}$ | SC for $f_i$'s, AveSS |
| | SVRS (Thm 3.3) | $\mathcal{O}\left(\left(n+\sqrt{n}\cdot\frac{\delta}{\mu}\right)\log\frac{1}{\varepsilon}\right)$ | AveSS |
| | AccSVRS (Thm 3.6) | $\mathcal{O}\left(\left(n+n^{3/4}\sqrt{\frac{\delta}{\mu}}\right)\log\frac{1}{\varepsilon}\right)$ | AveSS |
| | Lower bound (Thm 4.4) | $\Omega\left(n+n^{3/4}\sqrt{\frac{\delta}{\mu}}\log\frac{1}{\varepsilon}\right)^{(3)}$ | AveSS |

(1) The rate only applies to $\mathbb{E}\left\|\boldsymbol{x}_k - \boldsymbol{x}_*\right\|^2$, otherwise it would introduce $L$ in the log term; (2) The term $\log(L/\mu)$ comes from the Catalyst framework. See Appendix C for the detail. (2, 3) Here we only list the rates of the common ill-conditioned case: $\mu = \mathcal{O}(\delta/\sqrt{n})$. See Appendices for the remaining case. *Notation:* $\delta$=similarity parameter (both for SS and AveSS), $L$=smoothness constant of $f$, $\mu$=strong convexity constant of $f$(or $f_i$'s), $\varepsilon$=error of the solution for $\mathbb{E}f(\boldsymbol{x}_k) - f(\boldsymbol{x}_*)$. Here $L \geq \delta \geq \mu \gg \epsilon > 0$. *Abbreviation:* SC=strong convexity, SS=second-order similarity, AveSS=average SS.

AccSVRS achieves the same communication bound of $\tilde{\mathcal{O}}(n + n^{3/4}\sqrt{\delta/\mu})$ as Catalyzed SVRP. Specifically, our bound is entirely smoothness-free and slightly outperforms Catalyzed SVRP, featuring a log improvement and not requiring component strong convexity.

- Third, by considering the proximal incremental first-order oracle in the centralized distributed framework, we establish a lower bound, which nearly matches the upper bound of AccSVRS in ill-conditioned cases.

## 1.2 Related Work

**Gradient sliding/Oracle Complexity Separation.** For optimization problems with a separated structure or multiple building blocks, such as Eq. (1), there are scenarios where computing the gradients/values of some parts (or the whole) is more expensive than the others (or a partial one). In response to this challenge, techniques such as the gradient-sliding method [37] and the concept of oracle complexity separation [28] have emerged. These methods advocate for the infrequent use of more expensive oracles compared to their less resource-intensive counterparts. This strategy has found applications in zero-order [12, 21, 28, 53], first-order [37–39, 31] and high-order methods [31, 24, 3], as well as in addressing saddle point problems [4, 13]. Our algorithms can be viewed as a variance-reduced version of gradient sliding tailored to leverage the similarity assumption.

**Distributed optimization under similarity.** Distributed optimization has a long history with a plethora of existing works and surveys. To streamline our discussion, we only list the most relevant references, particularly under the similarity and strong convexity assumptions. In the full participation setting, which involves deterministic methods, the first algorithm credits to DANE [51], though its analysis is limited to quadratic objectives. Subsequently, AIDE [50], DANE-LS and DANE-HB [58] improved the rates for quadratic objective; Disco [62] SPAG [27], ACN [1] and DiRegINA [18] improved the rates for self-concordant objectives. As for general strongly convex objectives, Sun et al. [54] introduced the SONATA algorithm, and Tian et al. [55] proposed accelerated SONATA. However, their complexity bounds include additional log factors. These factors have recently been removed by Accelerated Extragradient [35], whose complexity bound perfectly matches the lower bound in [7]. We highly recommend the comparison of rates in [35, Table 1] for a comprehensive overview. Once the discussion of deterministic methods is concluded, Khaled and Jin [33] shifted their focus to stochastic methods using client sampling. They proposed SVRP and its Catalyzed version, both of which exhibited superior rates compared to deterministic methods.

## 2 Preliminaries

**Notation.** We denote vectors by lowercase bold letters (e.g., $\boldsymbol{w}, \boldsymbol{x}$), and matrices by capital bold letters (e.g., $\boldsymbol{A}, \boldsymbol{B}$). We let $\|\cdot\|$ be the $\ell_2$-norm for vectors, or induced $\ell_2$-norm for a given matrix: $\|\boldsymbol{A}\| = \sup_{\boldsymbol{u} \neq 0} \|\boldsymbol{A}\boldsymbol{u}\| / \|\boldsymbol{u}\|$. We abbreviate $[n] = \{1, \ldots, n\}$ and $\boldsymbol{I}_d \in \mathbb{R}^{d \times d}$ is the identity matrix. We use $\boldsymbol{0}$ for the all-zero vector/matrix, whose size will be specified by a subscript, if necessary, and otherwise is clear from the context. We denote $\mathrm{Unif}(\mathcal{S})$ as the uniform distribution over set $\mathcal{S}$. We say $T \sim \mathrm{Geom}(p)$ for $p \in (0, 1]$ if $\mathbb{P}(T = k) = (1-p)^{k-1}p, \forall k \in \{1, 2, \ldots\}$, i.e., $T$ obeys a geometric distribution. We adopt $\mathbb{E}_k$ as the expectation for all randomness appeared in step $k$, and $\mathbb{1}_A$ as the indicator function on event $A$, i.e., $\mathbb{1}_A = 1$ if event $A$ holds, and $0$ otherwise. We use $\mathcal{O}(\cdot), \Omega(\cdot), \Theta(\cdot)$ and $\tilde{\mathcal{O}}(\cdot)$ notation to hide universal constants and log-factors. We define the Bregman divergence induced by a differentiable (convex) function $h \colon \mathbb{R}^d \to \mathbb{R}$ as $D_h(\boldsymbol{x}, \boldsymbol{y}) := h(\boldsymbol{x}) - h(\boldsymbol{y}) - \langle \nabla h(\boldsymbol{y}), \boldsymbol{x} - \boldsymbol{y} \rangle$.

**Definitions.** We present the following common definitions used in this paper.

**Definition 2.1** *A differentiable function $g \colon \mathbb{R}^d \to \mathbb{R}$ is $\mu$-strongly convex (SC) if*

$$g(\boldsymbol{y}) \geq g(\boldsymbol{x}) + \langle \nabla g(\boldsymbol{x}), \boldsymbol{y} - \boldsymbol{x} \rangle + \frac{\mu}{2} \|\boldsymbol{y} - \boldsymbol{x}\|^2, \forall \boldsymbol{x}, \boldsymbol{y} \in \mathbb{R}^d. \tag{2}$$

*Particularly, if $\mu = 0$, we say that $g$ is convex.*

**Definition 2.2** *A differentiable function $g \colon \mathbb{R}^d \to \mathbb{R}$ is L-smooth if*

$$g(\boldsymbol{y}) \leq g(\boldsymbol{x}) + \langle \nabla g(\boldsymbol{x}), \boldsymbol{y} - \boldsymbol{x} \rangle + \frac{L}{2} \|\boldsymbol{y} - \boldsymbol{x}\|^2, \forall \boldsymbol{x}, \boldsymbol{y} \in \mathbb{R}^d. \tag{3}$$

There are many basic inequalities involving strong convexity and smoothness, see [22, Appendix A.1] for an introduction. Next, we present the definition of second-order similarity in distributed optimization.

**Definition 2.3** *The differentiable functions $f_i$'s satisfy $\delta$-average second-order similarity (AveSS) if the following inequality holds for $f_i$'s and $f = \frac{1}{n} \sum_{i=1}^{n} f_i$:*

$$(AveSS) \quad \frac{1}{n} \sum_{i=1}^{n} \|[\nabla[f_i - f](\boldsymbol{x}) - \nabla[f_i - f](\boldsymbol{y})]\|^2 \leq \delta^2 \|\boldsymbol{x} - \boldsymbol{y}\|^2, \forall \boldsymbol{x}, \boldsymbol{y} \in \mathbb{R}^d. \tag{4}$$

**Definition 2.4** *The differentiable functions $f_i$'s satisfy $\delta$-component second-order similarity (SS) if the following inequality holds for $f_i$'s and $f = \frac{1}{n} \sum_{i=1}^{n} f_i$:*

$$(SS) \quad \|[\nabla[f_i - f](\boldsymbol{x}) - \nabla[f_i - f](\boldsymbol{y})]\|^2 \leq \delta^2 \|\boldsymbol{x} - \boldsymbol{y}\|^2, \forall \boldsymbol{x}, \boldsymbol{y} \in \mathbb{R}^d, i \in [n]. \tag{5}$$

Definitions 2.3 and 2.4 first appear in [33], which is an analogy to (average) smoothness in prior literature [63]. Particularly, $f_i$'s satisfy $\delta$-AveSS implies that $(f - f_i)$'s satisfy $\delta$-average smoothness, while $f_i$'s satisfy $\delta$-SS implies that $(f - f_i)$'s satisfy $\delta$-smoothness. Additionally, many researchers [32, 7, 51, 62, 54, 35] use the equivalent one defined by Hessian similarity (HS) if assuming that $f_i$'s are twice differentiable. Thus we also list them below and leave the derivation in Appendix B.

$$(AveHS) \left\| \frac{1}{n} \sum_{i=1}^{n} \left[ \nabla^2 f_i(\boldsymbol{x}) - \nabla^2 f(\boldsymbol{x}) \right]^2 \right\| \leq \delta^2; (HS) \left\| \nabla^2 f_i(\boldsymbol{x}) - \nabla^2 f(\boldsymbol{x}) \right\| \leq \delta, \forall i \in [n]. \tag{6}$$

Since our algorithm is a first-order method, we adopt the gradient description of similarity (Definitions 2.3 and 2.4) without assuming twice differentiability for brevity.

As mentioned in [7, 54], if $f_i$'s satisfy $\delta$-AveSS (or SS), and $f$ is $\mu$-strongly convex and $L$-smooth, then generally $L \gg \delta \gg \mu > 0$ for large datasets in practice. Therefore, researchers aim to develop algorithms that achieve communication complexity solely related to $\delta, \mu$ (or log terms of $L$). This is also our objective. To finish this section, we will clarify several straightforward yet essential propositions, and the proofs are deferred to Appendix A.

**Proposition 2.5** *We have the following properties among SS, AveSS, and SC: 1) $\delta$-SS implies $\delta$-AveSS, but $\delta$-AveSS only implies $\sqrt{n}\delta$-SS. 2) If $f_i$'s satisfy $\delta$-SS and $f$ is $\mu$-strongly convex, then for all $i \in [n], f_i(\cdot) + \frac{\delta - \mu}{2} \|\cdot\|^2$ is convex, i.e., $f_i$ is $(\delta - \mu)$-almost convex [14].*

# 3 Algorithm and Theory

In this section, we introduce our main algorithms, which are developed to solve the distributed optimization problem in Eq. (1) under Assumption 1 below:

**Assumption 1** *We assume that $f_i$'s satisfy $\delta$-AveSS, and $f$ is $\mu$-strongly convex with $\delta \geq \mu > 0$.*

Assumption 1 does not need each $f_i$ to be $\mu$-strongly convex. In fact, it is acceptable that $f_i$'s are non-convex, since by Proposition 2.5, $f_i$'s are $(\sqrt{n}\delta - \mu)$-almost convex [14]. In the following, we first propose our new algorithm SVRS, which combines the techniques of gradient sliding and variance reduction, resulting in improved rates. Then we establish the directly accelerated method motivated by [6].

## 3.1 No Acceleration Version: SVRS

We first show the one-epoch Stochastic Variance-Reduced Sliding (SVRS[1ep]) method in Algorithm 1. Before delving into the theoretical analysis, we present some key insights into our method. These insights aim to enhance comprehension and facilitate connections with other algorithms.

**Variance Reduction.** Our algorithm can be viewed as adding variance reduction from [35]. Besides the acceleration step, the main difference lies in the proximal step, where Kovalev et al. [35] solved:

$$\boldsymbol{x}_{t+1} \approx \underset{\boldsymbol{x} \in \mathbb{R}^d}{\arg\min} \, B_\theta^t(\boldsymbol{x}) := \langle \nabla f(\boldsymbol{x}_t) - \nabla f_1(\boldsymbol{x}_t), \boldsymbol{x} - \boldsymbol{x}_t \rangle + \frac{1}{2\theta} \|\boldsymbol{x} - \boldsymbol{x}_t\|^2 + f_1(\boldsymbol{x}).$$

To save the heavy communication burden of calculating $\nabla f(\boldsymbol{x}_t)$, we apply client sampling by selecting a random $\nabla f_{i_t}(\boldsymbol{x}_t)$ in the $t$-th step. However, this substitution introduces significant noise. To mitigate this, we incorporate a correction term $\boldsymbol{g}_t = \nabla f_{i_t}(\boldsymbol{w}_0) - \nabla f(\boldsymbol{w}_0)$ from previous wisdom [29] to reduce the variance.

**Gradient sliding.** Our algorithm can be viewed as adding gradient sliding from SVRP [33]. The main difference also lies in the proximal point problem, where Khaled and Jin [33] solved:

$$\boldsymbol{x}_{t+1} \approx \underset{\boldsymbol{x} \in \mathbb{R}^d}{\arg\min} \, C_\theta^t(\boldsymbol{x}) := \langle -\boldsymbol{g}_t, \boldsymbol{x} - \boldsymbol{x}_t \rangle + \frac{1}{2\theta} \|\boldsymbol{x} - \boldsymbol{x}_t\|^2 + f_{i_t}(\boldsymbol{x}).$$

Here we adopt a fixed proximal function $f_1$ instead of $f_{i_t}$, which can be viewed as approximating $f_{i_t}(\boldsymbol{x}) \approx f_1(\boldsymbol{x}) + [f_{i_t} - f_1](\boldsymbol{x}_t) + \langle \nabla[f_{i_t} - f_1](\boldsymbol{x}_t), \boldsymbol{x} - \boldsymbol{x}_t \rangle + \frac{1}{2\theta'} \|\boldsymbol{x} - \boldsymbol{x}_t\|^2$ with a properly chosen $\theta' > 0$. Such a modification is motivated by [35], where they reformulated the objective as $f(\boldsymbol{x}) = [f(\boldsymbol{x}) - f_1(\boldsymbol{x})] + f_1(\boldsymbol{x})$. Thus they could employ gradient sliding to skip heavy computations of $\nabla[f - f_1](\boldsymbol{x})$ by utilizing the easy computations of $\nabla f_1(\boldsymbol{x})$ more times. Fixing the proximal function $f_1$ leads to the same metric space owned by $f_1$ in each step, which could benefit the analysis and alleviate the requirements on $f_i$'s compared to SVRP. Indeed, in our setting $f_1$ can be replaced by any other **fixed** client $f_b, b \in [n]$. In this case, the master node would be $f_b$ instead of $f_1$.

**Bregman-SVRG.** Our algorithm can be viewed as the classical Bregman-SVRG [20] with the reference function $f_1(\cdot) + \frac{1}{2\theta} \|\cdot\|^2$ after introducing the Bregman divergence:

$$\boldsymbol{x}_{t+1} \approx \underset{\boldsymbol{x} \in \mathbb{R}^d}{\arg\min} \, A_\theta^t(\boldsymbol{x}) \overset{(7)}{=} \underset{\boldsymbol{x} \in \mathbb{R}^d}{\arg\min} \langle \nabla f_{i_t}(\boldsymbol{x}_t) - \nabla[f_{i_t} - f](\boldsymbol{w}_0), \boldsymbol{x} - \boldsymbol{x}_t \rangle + D_{f_1(\cdot) + \frac{1}{2\theta}\|\cdot\|^2}(\boldsymbol{x}, \boldsymbol{x}_t).$$

We need to emphasize that the proof of Bregman-SVRG requires additional structural assumptions [20, Assumption 3], which is not directly applicable in our setting. Hence, the rigorous proof of Bregman-SVRG under our similarity assumption is still meaningful as far as we are concerned.

### 3.1.1 Communication Complexity under Distributed Settings

When applied to the distributed system, the communication complexity of SVRS[1ep] can be described as follows: At the beginning of each epoch, the master (corresponding to $f_1$) sends $\boldsymbol{w}_0$ to all clients. Each client computes $\nabla f_i(\boldsymbol{w}_0)$ from its local data and sends it back to the master. The master then builds $\nabla f(\boldsymbol{w}_0)$ after collecting all $\nabla f_i(\boldsymbol{w}_0)$'s. The communication complexity is $2(n-1)$ in this

---

**Algorithm 1** $\text{SVRS}^{1\text{ep}}(f, \boldsymbol{w}_0, \theta, p)$

---

1: **Input:** $\boldsymbol{w}_0 \in \mathbb{R}^d$, $p \in (0, 1)$, $\theta > 0$
2: Initialize $\boldsymbol{x}_0 = \boldsymbol{w}_0$, compute $\nabla f(\boldsymbol{w}_0)$, and set $T \sim \text{Geom}(p)$
3: **for** $t = 0, 1, 2, \ldots, T - 1$ **do**
4:     Sample $i_t \sim \text{Unif}([n])$ and compute $\boldsymbol{g}_t = \nabla f_{i_t}(\boldsymbol{w}_0) - \nabla f(\boldsymbol{w}_0)$
5:     Approximately solve the local proximal point problem:

$$\boldsymbol{x}_{t+1} \approx \underset{\boldsymbol{x} \in \mathbb{R}^d}{\arg\min} \, A_\theta^t(\boldsymbol{x}) := \langle \nabla f_{i_t}(\boldsymbol{x}_t) - \nabla f_1(\boldsymbol{x}_t) - \boldsymbol{g}_t, \boldsymbol{x} - \boldsymbol{x}_t \rangle + \frac{1}{2\theta} \|\boldsymbol{x} - \boldsymbol{x}_t\|^2 + f_1(\boldsymbol{x}) \tag{7}$$

6: **end for**
7: **Output:** $\boldsymbol{x}_T$

---

case. Next, the algorithm enters into the loop iterations. In each iteration, the master only sends current $\boldsymbol{x}_t$ to the chosen client $i_t$. The $i_t$-th client computes $\nabla f_{i_t}(\boldsymbol{x}_t)$ and sends it to the master (the first client). Then the master solves (inexactly) the local problem (Line 5 in Algorithm 1) to get an inexact solution $\boldsymbol{x}_{t+1}$. The communication complexity is 2 in this case. Thus, the total communication complexity of $\text{SVRS}^{1\text{ep}}$ is $2(n - 1) + 2T$. Note that $\mathbb{E}T = 1/p$ and generally $p = 1/n$. We obtain that one epoch communication complexity is $4n - 2$ in expectation.

We would like to emphasize that our setup differs from that in [41, 46], where the authors assume the nodes can perform calculations and transmit vectors in parallel. We recognize the significance of both setups. However, there are situations where communication is more expensive than computation. For instance, in a business network or communication network the communication between any two nodes can result in charges and the risk of information leakage. To mitigate these costs, we should reduce the frequency of communication. Thus, we focus on the nonparallel setting.

### 3.1.2 Convergence Analysis of SVRS

Based on the one-epoch method $\text{SVRS}^{1\text{ep}}$, we could introduce our non-accelerated algorithm SVRS, which starts from $\boldsymbol{w}_0 \in \mathbb{R}^d$ and repeatedly performs the update[7]

$$\boldsymbol{w}_{k+1} = \text{SVRS}^{1\text{ep}}(f, \boldsymbol{w}_k, \theta, p), \ \forall k \geq 0.$$

Now we derive the convergence rate of SVRS[8]. The main technique we apply is replacing the Euclidean distance with the Bergman divergence. Denote the reference function

$$h(\boldsymbol{x}) := f_1(\boldsymbol{x}) + \frac{1}{2\theta} \|\boldsymbol{x}\|^2 - f(\boldsymbol{x}). \tag{8}$$

By Assumption 1 and 1) in Proposition 2.5, we see that $f_i$'s are $\sqrt{n}\delta$-SS. i.e., $[f_1 - f](\cdot)$ is $(\sqrt{n}\delta)$-smooth. Thus, $h(\cdot)$ is $(\frac{1}{\theta} - \sqrt{n}\delta)$-strongly convex and $(\frac{1}{\theta} + \sqrt{n}\delta)$-smooth if $\theta < \frac{1}{\sqrt{n}\delta}$, that is,

$$0 \leq \frac{1 - \sqrt{n}\theta\delta}{2\theta} \|\boldsymbol{x} - \boldsymbol{y}\|^2 \overset{(2)}{\leq} D_h(\boldsymbol{x}, \boldsymbol{y}) \overset{(3)}{\leq} \frac{1 + \sqrt{n}\theta\delta}{2\theta} \|\boldsymbol{x} - \boldsymbol{y}\|^2. \tag{9}$$

Hence, if $\sqrt{n}\theta\delta = \Theta(1)$, $h(\cdot)$ is nearly a rescaled Euclidean norm since its condition number related to $\|\cdot\|$ is $\frac{1 + \sqrt{n}\theta\delta}{1 - \sqrt{n}\theta\delta} = \Theta(1)$. Next, we employ the properties of the Bregman divergence $D_h(\cdot, \cdot)$ to build the one-epoch progress of $\text{SVRS}^{1\text{ep}}$ as shown below:

**Lemma 3.1** *Suppose Assumption 1 holds. Let $\boldsymbol{w}^+ = \text{SVRS}^{1\text{ep}}(f, \boldsymbol{w}_0, \theta, p)$ with $\theta = 1/(4\sqrt{n}\delta)$, and the approximated solution $\boldsymbol{x}_{t+1}$ satisfies*

$$\left\| \nabla A_\theta^t(\boldsymbol{x}_{t+1}) \right\|^2 \leq \frac{\mu}{20\theta} \left\| \boldsymbol{x}_t - \underset{\boldsymbol{x} \in \mathbb{R}^d}{\arg\min} \, A_\theta^t(\boldsymbol{x}) \right\|^2, \forall t \geq 0. \tag{10}$$

---

[7]See Algorithm 3 in Appendix D for the details.
[8]Similar results for the popular loopless version [34] can also be derived, see Appendix D.5 for the detail.

---

**Algorithm 2** Accelerated SVRS (AccSVRS)

---

1: **Input:** $\boldsymbol{z}_0 = \boldsymbol{y}_0 \in \mathbb{R}^d, p, \tau \in (0,1), \alpha, \theta > 0, K \in \{1, 2, \dots\}$
2: **for** $k = 0, 1, 2, \dots, K-1$ **do**
3:     $\boldsymbol{x}_{k+1} = \tau \boldsymbol{z}_k + (1-\tau)\boldsymbol{y}_k$
4:     $\boldsymbol{y}_{k+1} = \text{SVRS}^{1\text{ep}}(f, \boldsymbol{x}_{k+1}, \theta, p)$
5:     $\boldsymbol{\mathcal{G}}_{k+1} = p\left(\nabla[f_1 - f_{j_k}](\boldsymbol{x}_{k+1}) - \nabla[f_1 - f_{j_k}](\boldsymbol{y}_{k+1}) + \frac{1}{\theta}\left(\boldsymbol{x}_{k+1} - \boldsymbol{y}_{k+1}\right)\right), j_k \sim \text{Unif}([n])$
6:     $\boldsymbol{z}_{k+1} = \arg\min_{\boldsymbol{z} \in \mathbb{R}^d} \frac{1}{2\alpha}\|\boldsymbol{z} - \boldsymbol{z}_k\|^2 + \langle \boldsymbol{\mathcal{G}}_{k+1}, \boldsymbol{z}\rangle + \frac{3\mu}{20}\|\boldsymbol{z} - \boldsymbol{y}_{k+1}\|^2 = \frac{\boldsymbol{z}_k + 0.3\mu\alpha\boldsymbol{y}_{k+1} - \alpha\boldsymbol{\mathcal{G}}_{k+1}}{1 + 0.3\mu\alpha}$
7: **end for**
8: **Output:** $\boldsymbol{y}_K$

---

Then for all $\boldsymbol{x} \in \mathbb{R}^d$ that is independent of the indices $i_1, i_2, \dots, i_T$ in $\text{SVRS}^{1\text{ep}}(f, \boldsymbol{w}_0, \theta, p)$, we have

$$\mathbb{E}f(\boldsymbol{w}^+) - f(\boldsymbol{x}) \leq \mathbb{E}\, p\langle \boldsymbol{x} - \boldsymbol{w}_0, \nabla h(\boldsymbol{w}^+) - \nabla h(\boldsymbol{w}_0)\rangle - \left(p - \frac{2}{9n}\right)D_h(\boldsymbol{w}_0, \boldsymbol{w}^+) - \frac{2\mu\theta}{5}D_h(\boldsymbol{x}, \boldsymbol{w}^+). \tag{11}$$

**Remark 3.2** *We note that some papers [10, 11] assume the smoothness and convexity of component functions, and adopt local updates for solving the proximal step. However, we replace these assumptions with a proximal approximately solvable assumption (10), which could even cover some nonsmooth and non-convex but proximal trackable component functions. We regard our assumption as more essential since the local updates can be viewed as partially solving this proximal step.*

The proof of Lemma 3.1 is left in Appendix D.1. From Lemma 3.1, we find a well-behaved proximal operator is sufficient to ensure favorable progress. Finally, we establish the convergence rate and communication complexity of the SVRS method, and the proof is deferred to Appendix D.2.

**Theorem 3.3** *Suppose Assumption 1 holds. If in* $\text{SVRS}^{1\text{ep}}$*(Algorithm 1), the hyperparameters are set as* $\theta = 1/(4\sqrt{n}\delta), p = 1/n$*, and the approximate solution* $\boldsymbol{x}_{t+1}$ *in each proximal step satisfies Eq.* (10)*. Then for any error* $\varepsilon > 0$*, when*

$$k \geq K_1 := \max\left\{2, \frac{5\delta}{\mu\sqrt{n}}\right\} \log \frac{3\left(1 + \frac{\delta}{\mu\sqrt{n}}\right)[f(\boldsymbol{w}_0) - f(\boldsymbol{x}_*)]}{\varepsilon},$$

*i.e., after* $\tilde{\mathcal{O}}(n + \sqrt{n}\delta/\mu)$ *communications in expectation, we obtain that* $\mathbb{E}f(\boldsymbol{w}_k) - f(\boldsymbol{x}_*) \leq \varepsilon$.

**Remark 3.4** *Our results enjoy the following advantages over SVRP [33]: The convergence of SVRP ([33, Theorem 2]) only applied to* $\mathbb{E}\|\boldsymbol{w}_k - \boldsymbol{x}_*\|^2$*, which can also be derived by our results from strong convexity:* $f(\boldsymbol{w}_k) - f(\boldsymbol{x}_*) \geq \frac{\mu}{2}\|\boldsymbol{w}_k - \boldsymbol{x}_*\|^2$*. However, the reverse is not applicable since we do not assume the smoothness of* $f$*, or indeed the smoothness coefficient is very large. Moreover, for ill-conditioned problems (e.g.,* $\delta/\mu \gg \sqrt{n}$*), our step size* $1/(4\sqrt{n}\delta)$ *is much larger than* $\mu/(2\delta^2)$ *used in SVRP, and the convergence rate is also faster than SVRP:* $\tilde{\mathcal{O}}(n + \sqrt{n}\delta/\mu)$ *vs.* $\tilde{\mathcal{O}}(n + \delta^2/\mu^2)$*. Finally, we do not need the strong convexity assumption of component functions.*

### 3.2 Acceleration Version: AccSVRS

Now we apply the classical interpolation technique motivated by Katyusha X [6] to establish accelerated SVRS (AccSVRS, Algorithm 2). The main difference between AccSVRS and Katyusha X is due to the different choices of distance spaces. Specifically, we adopt $D_h(\cdot, \cdot)$ instead of the Euclidean distance used in Katyusha X. Thus, the gradient mapping step (corresponding to Step 2 in [6, (4.1)]) should be built on the reference function $h(\cdot)$ defined in Eq. (8), i.e., $\nabla h(\boldsymbol{x}_{k+1}) - \nabla h(\boldsymbol{y}_{k+1})$ instead of $(\boldsymbol{x}_{k+1} - \boldsymbol{y}_{k+1})/\theta$. Moreover, noting that $\nabla h(\cdot)$ could involve the heavy gradient computing part $\nabla f(\cdot)$, we further employ its stochastic version (Step 5 in Algorithm 2) to reduce the overall communication complexity.

Next, we delve into the convergence analysis. We first give the core lemma for AccSVRS, which is also motivated by the framework of Katyusha X [6]. The proof is deferred to Appendix D.3.

**Lemma 3.5** *Suppose Assumption 1 holds, and $\theta = 1/(4\sqrt{n}\delta), p = 1/n, \alpha \leq n\theta/(2\tau)$ in Algorithm 2, where $\mathrm{SVRS}^{1\mathrm{ep}}(f, \boldsymbol{x}_{k+1}, \theta, p)$ satisfies Eq. (10) in each iteration. Then for all $\boldsymbol{x} \in \mathbb{R}^d$ that is independent of the random indices $i_1^{(k)}, i_2^{(k)}, \ldots, i_T^{(k)}$ in $\mathrm{SVRS}^{1\mathrm{ep}}(f, \boldsymbol{x}_{k+1}, \theta, p)$, we have that*

$$\mathbb{E}_k \frac{\alpha}{\tau} [f(\boldsymbol{y}_{k+1}) - f(\boldsymbol{x})] \leq \mathbb{E}_k(1-\tau) \cdot \frac{\alpha}{\tau} [f(\boldsymbol{y}_k) - f(\boldsymbol{x})] + \frac{\|\boldsymbol{x} - \boldsymbol{z}_k\|^2}{2} - \frac{1 + 0.3\mu\alpha}{2} \|\boldsymbol{x} - \boldsymbol{z}_{k+1}\|^2. \tag{12}$$

Finally, we present the convergence rate and communication complexity of AccSVRS based on Lemma 3.5, and the proof is left in Appendix D.4.

**Theorem 3.6** *Suppose Assumption 1 holds. Consider AccSVRS with the following hyperparameters*

$$\theta = \frac{1}{4\sqrt{n}\delta}, p = \frac{1}{n}, \tau = \frac{1}{4} \min\left\{1, \frac{n^{1/4}}{2}\sqrt{\frac{\mu}{\delta}}\right\}, \alpha = \frac{\sqrt{n}}{8\delta\tau},$$

*and Eq. (10) is satisfied in each iteration of $\mathrm{SVRS}^{1\mathrm{ep}}(f, \boldsymbol{x}_{k+1}, \theta, p)$. Then for any $\varepsilon > 0$, when*

$$k \geq K_2 := \max\left\{4, 8n^{-1/4}\sqrt{\delta/\mu}\right\} \log \frac{2[f(\boldsymbol{y}_0) - f(\boldsymbol{x}_*)]}{\varepsilon},$$

*i.e., after $\tilde{\mathcal{O}}\left(n + n^{3/4}\sqrt{\delta/\mu}\right)$ communications in expectation, we obtain that $\mathbb{E}f(\boldsymbol{y}_k) - f(\boldsymbol{x}_*) \leq \varepsilon$.*

**Remark 3.7** *Although roughly the same as the communication complexity obtained by Catalyzed SVRP in [33, Theorem 3], our results have the following advantages.*

***Fewer assumptions***. *Except for the strong convexity of $f$ and AveSS of $f_i$'s, we do not need to assume component strong convexity appearing in [33, Assumption 2].*

***Inexact proximal step.*** *Khaled and Jin [33, Theorem 3] require exact evaluations of the proximal operator, though they mention that this is only for the convenience of analysis. Our framework allows approximated solutions in each proximal step, and the approximation criterion (10) is error-independent, i.e., irrelevant to the final error $\varepsilon$. Since the local proximal function is strongly convex, we could solve the problem in a few steps if additionally assuming the smoothness of $f_1$.*

***Smoothness-free bound.*** *As shown in [33, Appendix G.1] or Appendix C, even if an exact proximal step is allowed, a dependence on the smoothness coefficient would be introduced in the total communication iterations of Catalyzed SVRP, though only in a log scale. Our directly accelerated method has no dependence on the smoothness coefficient.*

### 3.3 Gradient Complexity under Smooth Assumption

Due to the importance of total computation in the machine learning and optimization community, we consider a more common setup by **additionally assuming** that $f_1$ is $L$-smooth with $L \geq \delta \geq \mu > 0$, which together with Assumption 1 facilitates the quantification of Eq. (10). Then we can compute the total gradient complexity for AccSVRS as shown below. By Proposition 2.5 and our assumptions, $A_\theta^t(\boldsymbol{x})$ is $(\frac{1}{\theta} - \sqrt{n}\delta)$-strongly convex and $(\frac{1}{\theta} + L)$-smooth. Using accelerated methods starting from $\boldsymbol{x}_t$, we can guarantee that Eq. (10) holds after $T_{\mathrm{app}} = \tilde{\mathcal{O}}\left(\sqrt{\frac{1+\theta L}{1-\sqrt{n}\theta\delta}}\right) = \tilde{\mathcal{O}}\left(1 + n^{-1/4}\sqrt{L/\delta}\right)$ iterations with the choice of $\theta$ in Theorem 3.6. Hence, the total gradient calls in expectation are

$$\mathcal{O}(nT_{\mathrm{app}} \cdot K_2) = \tilde{\mathcal{O}}\left(n + n^{3/4}\left(\sqrt{\delta/\mu} + \sqrt{L/\delta}\right) + \sqrt{nL/\mu}\right).$$

Since $\delta \in [\mu, L]$, we recover the optimal gradient complexity $\tilde{\mathcal{O}}(n + n^{3/4}\sqrt{L/\mu})$ for the average smooth setting [63, Table 1] if neglecting log factors. Particularly, when $\delta = \Theta(\sqrt{\mu L})$, we even obtain the nearly optimal gradient complexity $\tilde{\mathcal{O}}(n + \sqrt{nL/\mu})$ for the component smooth setting [25, 26, 56]. We leave the details in Appendix E. Although the gradient complexity is not the primary focus of our work, we have demonstrated that the gradient complexity bound of AccSVRS is nearly optimal for certain values of $\delta$ in specific cases.

## 4 Lower Bound

In this section, we establish the lower bound of the communication complexity, which nearly matches the upper bound of AccSVRS.

## 4.1 Definition of Algorithms

In this subsection, we specify the class of algorithms to which our lower bound can apply. We first introduce the Proximal Incremental First-order Oracle (PIFO) [56, 25], which is defined as $h_{f_i}^{\mathrm{P}}(\boldsymbol{x}, \gamma) = [f_i(\boldsymbol{x}), \nabla f_i(\boldsymbol{x}), \mathrm{prox}_{f_i}^{\gamma}(\boldsymbol{x})]$ with $\gamma > 0$. Here the proximal operator is defined as $\mathrm{prox}_{f_i}^{\gamma}(\boldsymbol{x}) := \arg\min_{\boldsymbol{u}} \{ f_i(\boldsymbol{u}) + \frac{1}{2\gamma} \|\boldsymbol{x} - \boldsymbol{u}\|^2 \}$. In addition to the local zero-order and first-order information of $f_i$ at $\boldsymbol{x}$, the PIFO $h_{f_i}^{\mathrm{P}}(\boldsymbol{x}, \gamma)$ also provides some global information through the proximal operator[9]. Then we assume the algorithm has access to the PIFO and the definition of algorithms is presented as follows.

**Definition 4.1** *Consider a randomized algorithm $\mathcal{A}$ to solve problem* (1). *Suppose the number of communication rounds is $T$. At the initialization stage, the master node $1$ communicates with all the others. In round $t$ ($0 \le t \le T - 1$), the algorithm samples a node $i_t \sim \mathrm{Unif}([n])$, and node $1$ communicates with node $i_t$. Then the algorithm samples a Bernoulli random variable $a_t$ with constant expectation $c_0/n$. If $a_t = 1$, node $1$ communicates with all the others. Define the information set $\mathcal{I}_{t+1}$ as the set of all the possible points $\mathcal{A}$ can obtain after round $t$. The algorithm updates $\mathcal{I}_{t+1}$ based on the linear-span operation and PIFO, and finally outputs a certain point in $\mathcal{I}_T$.*

At the initialization stage, the communication cost is $2(n - 1)$. In each communication round, the Bernoulli random variable $a_t$ determines whether the master node communicates with all the others, i.e., whether to calculate the full gradient. Since $\mathbb{E}a_t = c_0/n$, the expected communication cost of each round is of the order $\Theta(1)$. Thus the total communication cost is of the order $\Theta(n + T)$ and we can use $T$ to measure the communication complexity. Moreover, one can check Algorithm 2 satisfies Definition 4.1. The formal definition and detailed analysis are deferred to Appendix F.1.

## 4.2 The Construction and Results

In this section, we construct a hard instance of problem (1) and then use it to establish the lower bound. Due to space limitations, we only present several key properties. The complete framework of construction is deferred to Appendix F.2.

Inspired by [25], we consider the class of matrices $\boldsymbol{B}(m, \zeta) = \begin{bmatrix} 1 & -1 & & \\ & \ddots & \ddots & \\ & & 1 & -1 \\ & & & \zeta \end{bmatrix} \in \mathbb{R}^{m \times m}$. This class

of matrices is widely used to establish lower bounds for minimax optimization problems [61, 49, 59], and $\boldsymbol{A}(m, \zeta) := \boldsymbol{B}(m, \zeta)^{\top} \boldsymbol{B}(m, \zeta)$ is the well-known tridiagonal matrix in the analysis of lower bounds for convex optimization [47, 40, 15]. Denote the $l$-th row of $\boldsymbol{B}(m, \zeta)$ as $\boldsymbol{b}_l(m, \zeta)^{\top}$. We partition the row vectors of $\boldsymbol{B}(m, \zeta)$ according to the index sets $\mathcal{L}_i = \{l : 1 \le l \le m, l \equiv i - 1 \,(\mathrm{mod}\,(n-1))\}$ for $2 \le i \le n$ and $\mathcal{L}_1 = \varnothing$[10]. These sets are mutually exclusive and their union is $[m]$. Then we consider the following problem

$$\min_{\boldsymbol{x} \in \mathbb{R}^m} r(\boldsymbol{x}; m, \zeta, c) = \frac{1}{n} \sum_{i=1}^{n} \left[ r_i(\boldsymbol{x}; m, \zeta, c) := \begin{cases} \frac{c}{2} \|\boldsymbol{x}\|^2 - n \langle \boldsymbol{e}_1, \boldsymbol{x} \rangle & \text{for } i = 1, \\ \frac{c}{2} \|\boldsymbol{x}\|^2 + \frac{n}{2} \sum_{l \in \mathcal{L}_i} \|\boldsymbol{b}_l(m, \zeta)^{\top} \boldsymbol{x}\|^2 & \text{for } i \ne 1. \end{cases} \right] \quad (13)$$

Here $\boldsymbol{e}_i \in \mathbb{R}^m$ denotes the unit vector with the $i$-th element equal to $1$ and others equal to $0$. Then one can check $r(\boldsymbol{x}; m, \zeta, c) = \frac{1}{2} \boldsymbol{x}^{\top} \boldsymbol{A}(m, \zeta) \boldsymbol{x} + \frac{c}{2} \|\boldsymbol{x}\|^2 - \langle \boldsymbol{e}_1, \boldsymbol{x} \rangle$. Clearly, $r$ is $c$-strongly convex. We can also determine the AveSS parameter as follows. The proof is deferred to Appendix F.3.

**Proposition 4.2** *Suppose that $0 < \zeta \le \sqrt{2}$, $n \ge 3$ and $m \ge 3$. Then $r_i$'s satisfy $\sqrt{8n+4}$-AveSS.*

Define the subspaces $\{\mathcal{F}_k\}_{k=0}^{m}$ as $\mathcal{F}_0 = \{\boldsymbol{0}\}$ and $\mathcal{F}_k = \mathrm{span}\{\boldsymbol{e}_1, \boldsymbol{e}_2, \ldots, \boldsymbol{e}_k\}$ for $1 \le k \le m$. The next lemma is fundamental to our analysis. The proof is deferred to Appendix F.5.

**Lemma 4.3** *Suppose the algorithm $\mathcal{A}$ satisfies Definition 4.1 and apply it to solve problem* (13) *with $n \ge 3$ and $m \ge 4$. We have (i) $\mathcal{I}_0 = \mathcal{F}_1$. (ii) Suppose $\mathcal{I}_t \subseteq \mathcal{F}_k$ ($1 \le k \le m - 3$). If $i_t$ satisfies $k \in \mathcal{L}_{i_t}$ or $a_t = 1$, then $\mathcal{I}_{t+1} \subseteq \mathcal{F}_{k+3}$; otherwise, $\mathcal{I}_{t+1} \subseteq \mathcal{F}_k$.*

---

[9]If we let $\gamma \to \infty$, $\mathrm{prox}_{f_i}^{\gamma}(\boldsymbol{x})$ converges to the exact minimizer of $f_i$, irrelevant to the choice of $\boldsymbol{x}$.

[10]Such a way of partitioning is also inspired by [25] and similar to that in [36]. However, our setting is different from theirs.

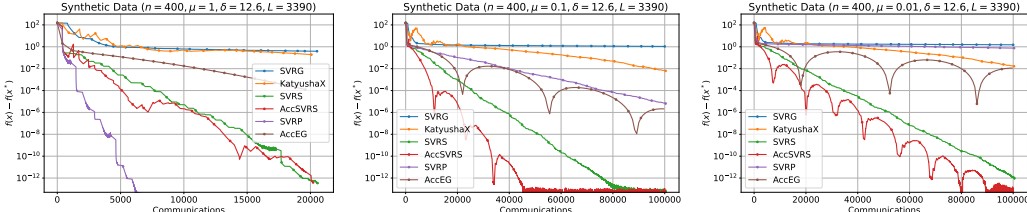

Figure 1: Numerical experiments on synthetic data. The corresponding coefficients are shown in the title of each graph. We plot the function gap on a log scale versus the number of communication steps, where one exchange of vectors counts as a communication step.

Lemma 4.3 guarantees that in each round, only when a specific component is sampled or the full gradient is calculated, can we expand the information set by at most three dimensions. For problem (13), we could never obtain an approximate solution unless we expand the information set to the whole space (see Proposition F.6 in Appendix F.2), while Lemma 4.3 implies that the process of expanding is very slow. Then we can establish the following lower bound.

**Theorem 4.4** *For any $n \geq 3$, $\delta, \mu > 0$, algorithm $\mathcal{A}$ satisfying Definition 4.1 and sufficiently small $\epsilon > 0$, there exists a rescaled version of problem* (13) *such that (i) Assumption 1 holds; (ii) In order to find an $\epsilon$-suboptimal solution $\hat{\boldsymbol{x}}$ such that $\mathbb{E} r(\hat{\boldsymbol{x}}) - \min_{\boldsymbol{x}} r(\boldsymbol{x}) < \epsilon$ by $\mathcal{A}$, the communication complexity in expectation is $\tilde{\Omega}(n + n^{3/4}\sqrt{\delta/\mu})$.*

This lower bound nearly matches the upper bound in Theorem 3.6 up to log factors, implying Algorithm 2 is nearly optimal in terms of communication complexity. The detailed statement and proof are deferred to Appendices F.2 and F.9.

## 5 Experiments

To demonstrate the advantages of our algorithms, we conduct the same numerical experiments as those in [35, 33]. We focus on the linear ridge regression problem with $\ell_2$ regularization, where the average loss $f$ has the formulation: $f(\boldsymbol{x}) = \frac{1}{n} \sum_{i=1}^{n} \left[ f_i(\boldsymbol{x}) := \frac{1}{m} \sum_{j=1}^{m} \left( \boldsymbol{z}_{i,j}^{\top} \boldsymbol{x} - y_{i,j} \right)^2 + \frac{\mu}{2} \|\boldsymbol{x}\|^2 \right]$. Here $\boldsymbol{z}_{i,j} \in \mathbb{R}^d$ and $y_{i,j} \in \mathbb{R}, \forall i \in [n], j \in [m]$ serve as the feature and label respectively, and $m$ can be viewed as data size in each local client. We consider a synthetic dataset generated by adding a small random noise matrix to the center matrix, ensuring a small $\delta$. To capture the differences in convergence rates between our methods and SVRP caused by different magnitudes of $\mu$, we vary $\mu = 10^{-i}, i \in \{0, 1, 2\}$. We compare our methods (SVRS and AccSVRS) against SVRG, KatyushaX, SVRP (Catalyzed SVRP is somehow hard to tune so we omit it), and Accelerated Extragradient (AccEG) using their theoretical step sizes, except that we scale the interpolation parameter $\tau$ in KatyushaX and AccSVRS for producing practical performance (see Appendix G for detail). From Figure 1, we can observe that for a large $\mu$, SVRP outperforms existing algorithms due to its high-order dependence on $\mu$. However, when the problem becomes ill-conditioned with a small $\mu$, AccSVRS exhibits significant improvements compared to other algorithms.

## 6 Conclusion

In this paper, we have introduced two new algorithms, SVRS and its directly accelerated version AccSVRS, and established improved communication complexity bounds for distributed optimization under the similarity assumption. Our rates are entirely smoothness-free and only require strong convexity of the objective, average similarity, and proximal friendliness of components. Moreover, our methods also have nearly optimal gradient complexity (leaving out the log term) when applied to smooth components in specific cases. It would be interesting to remove additional log terms to achieve both optimal communication and local gradient calls as [35], as well as investigating the complexity under other similarity assumptions (such as SS instead of AveSS) in future research.

## Acknowledgments and Disclosure of Funding

Lin, Han, and Zhang have been supported by the National Key Research and Development Project of China (No. 2022YFA1004002) and the National Natural Science Foundation of China (No. 12271011). Ye has been supported by the National Natural Science Foundation of China (No. 12101491).

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
