(x) := \frac{1}{m} \sum_{j=1}^{m} \left( z_{i,j}^\top x - y_{i,j} \right)^2 + \frac{\mu}{2} \|x\|^2 \right]$. Here $z_{i,j} \in \mathbb{R}^d$ and $y_{i,j} \in \mathbb{R}, \forall i \in [n], j \in [m]$ serve as the feature and label respectively, and $m$ can be viewed as data size in each local client. We consider a synthetic dataset generated by adding a small random noise matrix to the center matrix, ensuring a small $\delta$. To capture the differences in convergence rates between our methods and SVRP caused by different magnitudes of $\mu$, we vary $\mu = 10^{-i}, i \in \{0, 1, 2\}$. We compare our methods (SVRS and AccSVRS) against SVRG, KatyushaX, SVRP (Catalyzed SVRP is somehow hard to tune so we omit it), and Accelerated Extragradient (AccEG) using their theoretical step sizes, except that we scale the interpolation parameter $\tau$ in KatyushaX and AccSVRS for producing practical performance (see Appendix G for detail). From Figure 1, we can observe that for a large $\mu$, SVRP outperforms existing algorithms due to its high-order dependence on $\mu$. However, when the problem becomes ill-conditioned with a small $\mu$, AccSVRS exhibits significant improvements compared to other algorithms.

## 6 Conclusion

In this paper, we have introduced two new algorithms, SVRS and its directly accelerated version AccSVRS, and established improved communication complexity bounds for distributed optimization under the similarity assumption. Our rates are entirely smoothness-free and only require strong convexity of the objective, average similarity, and proximal friendliness of components. Moreover, our methods also have nearly optimal gradient complexity (leaving out the log term) when applied to smooth components in specific cases. It would be interesting to remove additional log terms to achieve both optimal communication and local gradient calls as [35], as well as investigating the complexity under other similarity assumptions (such as SS instead of AveSS) in future research.

## Acknowledgments and Disclosure of Funding

Lin, Han, and Zhang have been supported by the National Key Research and Development Project of China (No. 2022YFA1004002) and the National Natural Science Foundation of China (No. 12271011). Ye has been supported by the National Natural Science Foundation of China (No. 12101491).

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

# A  Auxiliary Results

**Proposition A.1 (Three-point identity [17, Lemma3.1])** *Given a differentiable function $h\colon \mathbb{R}^d \to \mathbb{R}$, we have the following equality:*

$$\langle \boldsymbol{x} - \boldsymbol{y}, \nabla h(\boldsymbol{y}) - \nabla h(\boldsymbol{z}) \rangle = D_h(\boldsymbol{x}, \boldsymbol{z}) - D_h(\boldsymbol{x}, \boldsymbol{y}) - D_h(\boldsymbol{y}, \boldsymbol{z}), \forall \boldsymbol{x}, \boldsymbol{y}, \boldsymbol{z} \in \mathbb{R}^d. \qquad (14)$$

**Proposition A.2** *Denote $\forall i \in \mathbb{N}, X_i = \begin{cases} 1 & \text{with probability} \quad p \\ 0 & \text{with probability} \quad 1-p \end{cases}$, and $X_1, X_2, \ldots$ are independent and identically distributed random variables. Then $Y := \inf_i \{i : X_i = 1\} \sim \mathrm{Geom}(p)$.*

Proof: We direct verify the probability distribution:

$$\mathbb{P}(Y = k) = \prod_{i=1}^{k-1} \mathbb{P}(X_i = 0) \mathbb{P}(X_k = 1) = (1-p)^{k-1} p, \quad k \in \{1, 2, \ldots\}.$$

Hence, we see that $Y \sim \mathrm{Geom}(p)$. $\qquad\qquad\qquad\qquad\qquad\qquad\qquad\qquad\qquad\qquad \square$

**Proposition A.3 (Proposition 2.5 in the main text)** *We have the following properties among SS, AveSS, and SC: 1) The $\delta$-SS can deduce $\delta$-AveSS, but $\delta$-AveSS can only deduce $\sqrt{n}\delta$-SS. 2) If $f_i$'s satisfy $\delta$-SS and $f$ is $\mu$-strongly convex, then for all $i \in [n]$, $f_i(\cdot) + \frac{\delta - \mu}{2} \|\cdot\|^2$ is convex, i.e., $f_i$ is $(\delta - \mu)$-almost convex [14].*

Proof: 1) The first part "$\delta$-SS $\Rightarrow$ $\delta$-AveSS" is trivial. The second part is because for all $i \in [n]$,

$$\|[\nabla[f_i - f](\boldsymbol{x}) - \nabla[f_i - f](\boldsymbol{y})]\|^2 \leq \sum_{j=1}^{n} \|[\nabla[f_j - f](\boldsymbol{x}) - \nabla[f_j - f](\boldsymbol{y})]\|^2 \overset{(4)}{\leq} n\delta^2 \|\boldsymbol{x} - \boldsymbol{y}\|^2.$$

Thus Eq. (5) holds with parameter $\sqrt{n}\delta$.

2) Since $f_i$'s satisfy $\delta$-SS, we get $\forall i \in [n], f - f_i$ is $\delta$-smooth, thus $\frac{\delta}{2} \|\boldsymbol{x}\|^2 - [f(\boldsymbol{x}) - f_i(\boldsymbol{x})]$ is convex (e.g., [22, Theorem A.1]). Moreover, we also have $f(\boldsymbol{x}) - \frac{\mu}{2} \|\boldsymbol{x}\|^2$ is a convex function since $f$ is $\mu$-strongly convex (e.g., [22, Theorem A.2]). Therefore, we obtain that

$$f_i(\boldsymbol{x}) + \frac{\delta - \mu}{2} \|\boldsymbol{x}\|^2 = \left( \frac{\delta}{2} \|\boldsymbol{x}\|^2 - [f(\boldsymbol{x}) - f_i(\boldsymbol{x})] \right) + \left( f(\boldsymbol{x}) - \frac{\mu}{2} \|\boldsymbol{x}\|^2 \right)$$

is also convex. The proof is finished. $\qquad\qquad\qquad\qquad\qquad\qquad\qquad\qquad\qquad\qquad \square$

**Lemma A.4 (Allen-Zhu [6, Fact 2.3])** *Given sequence $D_0, D_1, \ldots$ of reals, if $N \sim Geom(p)$, then*

$$\mathbb{E}_N[D_{N-1} - D_N] = p\mathbb{E}[D_0 - D_N], \mathbb{E}_N[D_{N-1}] = (1-p)\mathbb{E}[D_N] + pD_0 \qquad (15)$$

**Lemma A.5 (Allen-Zhu [6, Lemma 2.4])** *If $g(\cdot)$ is proper convex and $\sigma$-strongly convex and $\boldsymbol{z}_{k+1} = \arg\min_{\boldsymbol{z} \in \mathbb{R}^d} \frac{1}{2\alpha} \|\boldsymbol{z} - \boldsymbol{z}_k\|^2 + \langle \boldsymbol{\xi}, \boldsymbol{z} \rangle + g(\boldsymbol{z})$, then for every $\boldsymbol{x} \in \mathbb{R}^d$, we have*

$$\langle \boldsymbol{\xi}, \boldsymbol{z}_k - \boldsymbol{x} \rangle + g(\boldsymbol{z}_{k+1}) - g(\boldsymbol{x}) \leq \frac{\alpha}{2} \|\boldsymbol{\xi}\|^2 + \frac{\|\boldsymbol{x} - \boldsymbol{z}_k\|^2}{2\alpha} - \frac{(1 + \sigma\alpha) \|\boldsymbol{x} - \boldsymbol{z}_{k+1}\|^2}{2\alpha}. \qquad (16)$$

**Lemma A.6 (Han et al. [25, Lemma 2.10])** *Let $\{Y_i\}_{i=1}^{m}$ be independent random variables such that $Y_i \sim \mathrm{Geom}(p_i)$ with $p_i > 0$. Then for $m \geq 2$, we have*

$$\mathbb{P}\left( \sum_{i=1}^{m} Y_i > \frac{m^2}{4(\sum_{i=1}^{m} p_i)} \right) \geq \frac{1}{9}.$$

# B  Hessian Similarity

In this section, we show that AveHS (HS) defined in Eq. (6) is equivalent to AveSS (SS).

**Proposition B.1** *For twice differentiability $f_i$'s and $f$, AveSS $\Leftrightarrow$ AveHS, SS $\Leftrightarrow$ HS.*

Proof: Indeed, we only need to prove the following results for twice differentiability $g$:

$$\frac{1}{n}\sum_{i=1}^{n}\|\nabla g_i(\boldsymbol{x}) - \nabla g_i(\boldsymbol{y})\|^2 \le \delta^2\|\boldsymbol{y}-\boldsymbol{x}\|^2 \Leftrightarrow \left\|\frac{1}{n}\sum_{i=1}^{n}\left(\nabla^2 g_i(\boldsymbol{x})\right)^2\right\| \le \delta^2, \forall \boldsymbol{x},\boldsymbol{y}\in\mathbb{R}^d. \quad (17)$$

"$\Rightarrow$": Taking $\boldsymbol{y} = \boldsymbol{x}+t\boldsymbol{v}, t\in\mathbb{R}\backslash\{0\}, \boldsymbol{v}\in\mathbb{R}^d, \|\boldsymbol{v}\|=1$ and letting $t\to 0$, we get

$$\begin{aligned}
\delta^2 &= \lim_{t\to 0}\frac{\delta^2\|\boldsymbol{x}-\boldsymbol{y}\|^2}{t^2} \ge \lim_{t\to 0}\frac{1}{n}\sum_{i=1}^{n}\left\|\frac{[\nabla g_i(\boldsymbol{x}) - \nabla g_i(\boldsymbol{x}+t\boldsymbol{v})]}{t}\right\|^2 \\
&= \frac{1}{n}\sum_{i=1}^{n}\|\nabla^2 g_i(\boldsymbol{x})\boldsymbol{v}\|^2 = \boldsymbol{v}^\top\left[\frac{1}{n}\sum_{i=1}^{n}\left(\nabla^2 g_i(\boldsymbol{x})\right)^2\right]\boldsymbol{v}.
\end{aligned}$$

The final equality uses the fact that $\nabla^2 g_i(\boldsymbol{x})$ is a symmetric matrix. Now by the arbitrary of $\boldsymbol{v}\in\mathbb{R}^d$ with $\|\boldsymbol{v}\|=1$, we get $\left\|\frac{1}{n}\sum_{i=1}^{n}\left(\nabla^2 g_i(\boldsymbol{x})\right)^2\right\| \le \delta^2$.

"$\Leftarrow$": We use the integral formulation:

$$\begin{aligned}
\frac{1}{n}\sum_{i=1}^{n}\|\nabla g_i(\boldsymbol{x}) - \nabla g_i(\boldsymbol{y})\|^2 &= \frac{1}{n}\sum_{i=1}^{n}\left\|\int_0^1 \nabla^2 g_i(\boldsymbol{x}+t(\boldsymbol{y}-\boldsymbol{x}))(\boldsymbol{y}-\boldsymbol{x})\,dt\right\|^2 \\
&= (\boldsymbol{y}-\boldsymbol{x})\left[\frac{1}{n}\sum_{i=1}^{n}\int_0^1 \nabla^2 g_i(\boldsymbol{x}+s(\boldsymbol{y}-\boldsymbol{x}))\nabla^2 g_i(\boldsymbol{x}+t(\boldsymbol{y}-\boldsymbol{x}))ds\,dt\right](\boldsymbol{y}-\boldsymbol{x}) \\
&\overset{(i)}{\le} (\boldsymbol{y}-\boldsymbol{x})\left[\frac{1}{n}\sum_{i=1}^{n}\int_0^1 \left(\nabla^2 g_i(\boldsymbol{x}+t(\boldsymbol{y}-\boldsymbol{x}))\right)^2 dt\right](\boldsymbol{y}-\boldsymbol{x}) \\
&\le \int_0^1\left\|\frac{1}{n}\sum_{i=1}^{n}\left(\nabla^2 g_i(\boldsymbol{x}+t(\boldsymbol{y}-\boldsymbol{x}))\right)^2\right\|\cdot\|\boldsymbol{y}-\boldsymbol{x}\|^2\,dt \le \delta^2\|\boldsymbol{y}-\boldsymbol{x}\|^2,
\end{aligned}$$

where $(i)$ uses the inequality $\boldsymbol{A}_t^2 + \boldsymbol{A}_s^2 \succeq \boldsymbol{A}_s\boldsymbol{A}_t + \boldsymbol{A}_t\boldsymbol{A}_s$ for symmetric matrices $\boldsymbol{A}_s, \forall s\in[0,1]$ since $(\boldsymbol{A}_t - \boldsymbol{A}_s)^2 \succeq 0$, and the final inequality uses the assumption.

Hence, Eq. (17) is proved. Now choosing $g_i = f_i - f, \forall i\in[n]$, we obtain "AveSS $\Leftrightarrow$ AveHS". Additionally, letting $n=1$ and noting that $\left\|\left(\nabla^2 g_i(\boldsymbol{x})\right)^2\right\| = \|\nabla^2 g_i(\boldsymbol{x})\|^2$, we obtain "SS $\Leftrightarrow$ HS". The proof is finished. $\qquad\square$

## C  Concrete Complexity of Catalyst SVRP

Inherited from the computation of [33, Appendix G.1], we see that the total iterations of Catalyst SVRP is

$$\begin{aligned}
\mathbb{E}T_{\text{iter}}^{\text{total}} &= 8\sqrt{\frac{\mu+\gamma}{\mu}}\max\left\{\frac{\delta^2}{(\gamma+\mu)^2}, n\right\}\log\left(\frac{f(\boldsymbol{x}_0)-f(\boldsymbol{x}_*)}{\varepsilon}\cdot\frac{32(\mu+\gamma)}{\mu}\right)\log\iota, \\
\iota &:= A\left(\frac{2}{1-\rho} + \frac{2592\gamma}{\mu(1-\rho)^2(\sqrt{q}-\rho)^2}\right),
\end{aligned}$$

where $\rho = \sqrt{q}/2 = \frac{\sqrt{\mu/(\mu+\gamma)}}{2}\in(0,\frac{1}{2})$, $A = \frac{L+\gamma}{\mu+\gamma}\left(1+\frac{(\gamma+\mu)^2 n}{\delta^2}\right)$. Letting $\gamma = \max\left\{\frac{\delta}{\sqrt{n}}-\mu, 0\right\}$, we recover the complexity:

$$\begin{aligned}
\mathbb{E}T_{\text{iter}}^{\text{total}} &= 8\max\left\{n, n^{3/4}\sqrt{\frac{\delta}{\mu}}\right\}\log\left(\max\left\{32,\frac{32\delta}{\mu\sqrt{n}}\right\}\cdot\frac{f(\boldsymbol{x}_0)-f(\boldsymbol{x}_*)}{\varepsilon}\right)\log\iota, \\
\iota &= A\left(\frac{2}{1-\rho} + \frac{2592\gamma}{\mu(1-\rho)^2(\sqrt{q}-\rho)^2}\right) = \Theta\left(A\left(1+\frac{\gamma(\mu+\gamma)}{\mu^2}\right)\right) = \Theta\left(\frac{A(\mu+\gamma)^2}{\mu^2}\right).
\end{aligned}$$

When $\delta/\mu \le \sqrt{n}$, leading to $\gamma = 0$, then we get

$$\frac{A(\mu+\gamma)^2}{\mu^2} = \frac{L}{\mu}\left(1+\frac{\mu^2 n}{\delta^2}\right) = \Theta\left(\frac{L\mu n}{\delta^2}\right).$$

---

**Algorithm 3** Stochastic Variance-Reduced Sliding (SVRS)

---

1: **Input:** $\boldsymbol{w}_0 \in \mathbb{R}^d, p \in (0,1), \theta > 0, K \in \{1, 2, \dots\}$
2: **for** $k = 0, 1, 2, \dots, K - 1$ **do**
3: $\quad \boldsymbol{w}_{k+1} = \text{SVRS}^{\text{1ep}}(f, \boldsymbol{w}_k, \theta, p)$
4: **end for**
5: **Output:** $\boldsymbol{w}_K$

---

Thus, $\mathbb{E} T_{\text{iter}}^{\text{total}} = \mathcal{O}\left(\left(n + n^{3/4}\sqrt{\frac{\delta}{\mu}}\right) \log \frac{f(\boldsymbol{x}_0) - f(\boldsymbol{x}_*)}{\varepsilon} \log \frac{L\mu n}{\delta^2}\right)$.

When $\delta/\mu \geq \sqrt{n}$, i.e., $\max\left\{n, n^{3/4}\sqrt{\frac{\delta}{\mu}}\right\} = n^{3/4}\sqrt{\frac{\delta}{\mu}}$, we get $\gamma = \frac{\delta}{\sqrt{n}} - \mu \leq L - \mu$ (note that $L \geq \delta \geq \mu, n \geq 1$ by assumption), leading to

$$\frac{2L}{\mu} \leq \frac{A(\mu + \gamma)^2}{\mu^2} = \frac{2(L + \gamma)(\mu + \gamma)}{\mu^2} \leq \frac{4L^2}{\mu^2}.$$

Thus, $\mathbb{E} T_{\text{iter}}^{\text{total}} = \mathcal{O}\left(\left(n + n^{3/4}\sqrt{\frac{\delta}{\mu}}\right) \log \frac{f(\boldsymbol{x}_0) - f(\boldsymbol{x}_*)}{\varepsilon} \log \frac{L}{\mu}\right)$ (for small enough error $\varepsilon$).

## D Proofs for Section 3

The complete procedure of SVRS is presented in Algorithm 3. Before giving the omit proofs, we need the following one-step lemma.

**Lemma D.1** *Suppose Assumption 1 holds. If the step size $\theta \leq 1/(2\sqrt{n}\delta)$ in SVRS$^{\text{1ep}}$(Algorithm 1), then the following inequality holds for all $\boldsymbol{x} \in \mathbb{R}^d$ that is independent to the index $i_k$:*

$$\mathbb{E}_t[f(\boldsymbol{x}_{t+1}) - f(\boldsymbol{x})] \leq \mathbb{E}_t D_h(\boldsymbol{x}, \boldsymbol{x}_t) - \left(1 + \frac{\mu\theta/2}{1 + \sqrt{n}\theta\delta}\right) D_h(\boldsymbol{x}, \boldsymbol{x}_{t+1}) + \frac{2\theta^2\delta^2}{(1 - \sqrt{n}\theta\delta)^2} D_h(\boldsymbol{w}_0, \boldsymbol{x}_t)$$

$$+ \frac{2 + \mu\theta}{2\mu}\left[\left\|\nabla A_\theta^t(\boldsymbol{x}_{t+1})\right\|^2 - \frac{\mu}{20\theta}\left\|\boldsymbol{x}_t - \underset{\boldsymbol{x} \in \mathbb{R}^d}{\arg\min} A_\theta^t(\boldsymbol{x})\right\|^2\right]. \tag{18}$$

Proof: First, note that

$$\nabla A_\theta^t(\boldsymbol{x}) = \frac{\boldsymbol{x} - \boldsymbol{x}_t}{\theta} + \nabla f_1(\boldsymbol{x}) - \nabla f_1(\boldsymbol{x}_t) + \nabla f_{i_t}(\boldsymbol{x}_t) - [\nabla f_{i_t}(\boldsymbol{w}_0) - \nabla f(\boldsymbol{w}_0)]$$

$$= \nabla f(\boldsymbol{x}) + \nabla h(\boldsymbol{x}) - \nabla h(\boldsymbol{x}_t) + \nabla(f_{i_t} - f)(\boldsymbol{x}_t) - \nabla(f_{i_t} - f)(\boldsymbol{w}_0). \tag{19}$$

Now we begin from the strong convexity of function $f$ in Assumption 1,

$$\mathbb{E}_t[f(\boldsymbol{x}_{t+1}) - f(\boldsymbol{x})] \overset{(2)}{\leq} \mathbb{E}_t \langle \boldsymbol{x} - \boldsymbol{x}_{t+1}, -\nabla f(\boldsymbol{x}_{t+1}) \rangle - \frac{\mu}{2}\|\boldsymbol{x}_{t+1} - \boldsymbol{x}\|^2$$

$$\overset{(19)}{=} \mathbb{E}_t \langle \boldsymbol{x} - \boldsymbol{x}_{t+1}, \nabla h(\boldsymbol{x}_{t+1}) - \nabla h(\boldsymbol{x}_t) \rangle + \langle \boldsymbol{x} - \boldsymbol{x}_{t+1}, \nabla(f_{i_t} - f)(\boldsymbol{x}_t) - \nabla(f_{i_t} - f)(\boldsymbol{w}_0) \rangle$$

$$\quad - \langle \boldsymbol{x} - \boldsymbol{x}_{t+1}, \nabla A_\theta^t(\boldsymbol{x}_{t+1}) \rangle - \frac{\mu}{2}\|\boldsymbol{x} - \boldsymbol{x}_{t+1}\|^2$$

$$\overset{(i)}{=} \mathbb{E}_t D_h(\boldsymbol{x}, \boldsymbol{x}_t) - D_h(\boldsymbol{x}, \boldsymbol{x}_{t+1}) - D_h(\boldsymbol{x}_{t+1}, \boldsymbol{x}_t) + \langle \boldsymbol{x}_t - \boldsymbol{x}_{t+1}, \nabla(f_{i_t} - f)(\boldsymbol{x}_t) - \nabla(f_{i_t} - f)(\boldsymbol{w}_0) \rangle$$

$$\quad - \langle \boldsymbol{x} - \boldsymbol{x}_{t+1}, \nabla A_\theta^t(\boldsymbol{x}_{t+1}) \rangle - \frac{\mu}{2}\|\boldsymbol{x}_{t+1} - \boldsymbol{x}\|^2$$

$$\leq \mathbb{E}_t D_h(\boldsymbol{x}, \boldsymbol{x}_t) - D_h(\boldsymbol{x}, \boldsymbol{x}_{t+1}) - D_h(\boldsymbol{x}_{t+1}, \boldsymbol{x}_t)$$

$$\quad + \frac{1 - \sqrt{n}\theta\delta}{4\theta}\|\boldsymbol{x}_{t+1} - \boldsymbol{x}_t\|^2 + \frac{\theta}{1 - \sqrt{n}\theta\delta}\|\nabla(f_{i_t} - f)(\boldsymbol{x}_t) - \nabla(f_{i_t} - f)(\boldsymbol{w}_0)\|^2$$

$$\quad + \left[\frac{\mu}{4}\|\boldsymbol{x}_{t+1} - \boldsymbol{x}\|^2 + \frac{1}{\mu}\|\nabla A_\theta^t(\boldsymbol{x}_{t+1})\|^2\right] - \frac{\mu}{2}\|\boldsymbol{x}_{t+1} - \boldsymbol{x}\|^2 \tag{20}$$

where $(i)$ uses Eq. (14) and $\mathbb{E}_{i_t} \langle \boldsymbol{x}_t - \boldsymbol{x}, \nabla(f_{i_t} - f)(\boldsymbol{x}_t) - \nabla(f_{i_t} - f)(\boldsymbol{w}_0) \rangle = 0$ since $\boldsymbol{x}_t - \boldsymbol{x}$ is independent to $i_t$ and the final inequality uses $\langle \boldsymbol{a}, \boldsymbol{b} \rangle \leq t^2 \|\boldsymbol{a}\|^2 + \frac{\|\boldsymbol{b}\|^2}{4t^2}$ twice. Next, we continue

using Eq. (9) to convert $\|\cdot\|$ with $D_h(\cdot, \cdot)$ by assumption $\theta \leq 1/(2\sqrt{n}\delta)$:

$$\mathbb{E}_t[f(\boldsymbol{x}_{t+1}) - f(\boldsymbol{x})]$$

$$\overset{(20)(9)}{\leq} \mathbb{E}_t D_h(\boldsymbol{x}, \boldsymbol{x}_t) - \left(1 + \frac{\mu\theta/2}{1 + \sqrt{n}\theta\delta}\right) D_h(\boldsymbol{x}, \boldsymbol{x}_{t+1}) - \frac{1 - \sqrt{n}\theta\delta}{4\theta} \|\boldsymbol{x}_{t+1} - \boldsymbol{x}_t\|^2$$

$$+ \frac{\theta}{1 - \sqrt{n}\theta\delta} \|\nabla(f_{i_t} - f)(\boldsymbol{x}_t) - \nabla(f_{i_t} - f)(\boldsymbol{w}_0)\|^2 + \frac{1}{\mu} \left\|\nabla A_\theta^t(\boldsymbol{x}_{t+1})\right\|^2$$

$$\overset{(4)}{\leq} \mathbb{E}_t D_h(\boldsymbol{x}, \boldsymbol{x}_t) - \left(1 + \frac{\mu\theta/2}{1 + \sqrt{n}\theta\delta}\right) D_h(\boldsymbol{x}, \boldsymbol{x}_{t+1}) + \frac{\theta\delta^2}{1 - \sqrt{n}\theta\delta} \|\boldsymbol{x}_t - \boldsymbol{w}_0\|^2$$

$$- \frac{1 - \sqrt{n}\theta\delta}{4\theta} \|\boldsymbol{x}_{t+1} - \boldsymbol{x}_t\|^2 + \frac{1}{\mu} \left\|\nabla A_\theta^t(\boldsymbol{x}_{t+1})\right\|^2$$

$$\overset{(9)}{\leq} \mathbb{E}_t D_h(\boldsymbol{x}, \boldsymbol{x}_t) - \left(1 + \frac{\mu\theta/2}{1 + \sqrt{n}\theta\delta}\right) D_h(\boldsymbol{x}, \boldsymbol{x}_{t+1}) + \frac{2\theta^2\delta^2}{(1 - \sqrt{n}\theta\delta)^2} D_h(\boldsymbol{w}_0, \boldsymbol{x}_t)$$

$$- \frac{1 - \sqrt{n}\theta\delta}{4\theta} \|\boldsymbol{x}_{t+1} - \boldsymbol{x}_t\|^2 + \frac{1}{\mu} \left\|\nabla A_\theta^t(x_{t+1})\right\|^2 .$$

Finally, we show the error analysis if an approximate solution, i.e., $\|\nabla A_\theta^t(\boldsymbol{x}_{t+1})\| \neq 0$ is allowed. Using Proposition 2.5, we see that $f_1(\boldsymbol{x}) + \frac{\sqrt{n}\delta - \mu}{2} \|\boldsymbol{x}\|^2$ is a convex function, leading to $A_\theta^t(\boldsymbol{x})$ is $\left(\frac{1}{\theta} - \sqrt{n}\delta + \mu\right)$-strongly convex function. Let $\hat{\boldsymbol{x}}_{k+1} \in \arg\min_{\boldsymbol{x} \in \mathbb{R}^d} A_\theta^t(\boldsymbol{x})$, i.e., $\nabla A_\theta^t(\hat{\boldsymbol{x}}_{k+1}) = 0$. Since $\theta \leq 1/(2\sqrt{n}\delta)$, we can further bound the last two terms:

$$- \frac{1 - \sqrt{n}\theta\delta}{4\theta} \|\boldsymbol{x}_{t+1} - \boldsymbol{x}_t\|^2 + \frac{1}{\mu} \left\|\nabla A_\theta^t(\boldsymbol{x}_{t+1})\right\|^2 \leq \frac{1}{\mu} \left\|\nabla A_\theta^t(\boldsymbol{x}_{t+1})\right\|^2 - \frac{1}{8\theta} \|\boldsymbol{x}_{t+1} - \boldsymbol{x}_t\|^2$$

$$\leq \frac{1}{\mu} \left\|\nabla A_\theta^t(\boldsymbol{x}_{t+1})\right\|^2 + \frac{1}{8\theta} \|\boldsymbol{x}_{t+1} - \hat{\boldsymbol{x}}_{t+1}\|^2 - \frac{1}{16\theta} \|\hat{\boldsymbol{x}}_{t+1} - \boldsymbol{x}_t\|^2$$

$$\leq \frac{1}{\mu} \left\|\nabla A_\theta^t(\boldsymbol{x}_{t+1})\right\|^2 + \frac{\theta}{8(1 - (\sqrt{n}\delta - \mu)\theta)^2} \left\|\nabla A_\theta^t(\boldsymbol{x}_{t+1}) - \nabla A_\theta^t(\hat{\boldsymbol{x}}_{t+1})\right\|^2 - \frac{1}{16\theta} \|\hat{\boldsymbol{x}}_{t+1} - \boldsymbol{x}_t\|^2$$

$$\leq \frac{1}{\mu} \left\|\nabla A_\theta^t(\boldsymbol{x}_{t+1})\right\|^2 + \frac{\theta}{2} \left\|\nabla A_\theta^t(\boldsymbol{x}_{t+1})\right\|^2 - \frac{1}{16\theta} \|\boldsymbol{x}_t - \hat{\boldsymbol{x}}_{t+1}\|^2$$

$$= \frac{2 + \mu\theta}{2\mu} \left[\left\|\nabla A_\theta^t(\boldsymbol{x}_{t+1})\right\|^2 - \frac{\mu}{8\theta(2 + \mu\theta)} \left\|\boldsymbol{x}_t - \arg\min_{\boldsymbol{x} \in \mathbb{R}^d} A_\theta^t(\boldsymbol{x})\right\|^2\right]$$

$$\leq \frac{2 + \mu\theta}{2\mu} \left[\left\|\nabla A_\theta^t(\boldsymbol{x}_{t+1})\right\|^2 - \frac{\mu}{20\theta} \left\|\boldsymbol{x}_t - \arg\min_{\boldsymbol{x} \in \mathbb{R}^d} A_\theta^t(\boldsymbol{x})\right\|^2\right].$$

Therefore, Eq. (18) is proved. $\qquad\square$

## D.1 Proof of Lemma 3.1

Proof: Since $\theta = 1/(4\sqrt{n}\delta)$ satisfies the condition required in Lemma D.1, we get

$$\mathbb{E}_t[f(\boldsymbol{x}_{t+1}) - f(\boldsymbol{x})] \overset{(18)}{\leq} \mathbb{E}_t D_h(\boldsymbol{x}, \boldsymbol{x}_t) - \left(1 + \frac{2\mu\theta}{5}\right) D_h(\boldsymbol{x}, \boldsymbol{x}_{t+1}) + \frac{2}{9n} D_h(\boldsymbol{w}_0, \boldsymbol{x}_t).$$

Taking $t = T - 1$ with $T \sim \text{Geom}(p)$ and noting that $\boldsymbol{w}^+ = \boldsymbol{x}_T, \boldsymbol{w}_0 = \boldsymbol{x}_0$, by Lemma A.4, we get

$$\mathbb{E}[f(\boldsymbol{w}^+) - f(\boldsymbol{x})] = \mathbb{E}[f(\boldsymbol{x}_T) - f(\boldsymbol{x})]$$

$$\leq \mathbb{E} D_h(\boldsymbol{x}, \boldsymbol{x}_{T-1}) - D_h(\boldsymbol{x}, \boldsymbol{x}_T) - \frac{2\mu\theta}{5} D_h(\boldsymbol{x}, \boldsymbol{x}_T) + \frac{2}{9n} D_h(\boldsymbol{w}_0, \boldsymbol{x}_{T-1})$$

$$\overset{(15)}{=} \mathbb{E} \, p D_h(\boldsymbol{x}, \boldsymbol{x}_0) - p D_h(\boldsymbol{x}, \boldsymbol{x}_T) - \frac{2\mu\theta}{5} D_h(\boldsymbol{x}, \boldsymbol{x}_T)$$

$$+ \frac{2}{9n}[(1 - p) D_h(\boldsymbol{w}_0, \boldsymbol{x}_T) + p D_h(\boldsymbol{w}_0, \boldsymbol{x}_0)]$$

$$\leq \mathbb{E} \, p D_h(\boldsymbol{x}, \boldsymbol{w}_0) - p D_h(\boldsymbol{x}, \boldsymbol{w}^+) - \frac{2\mu\theta}{5} D_h(\boldsymbol{x}, \boldsymbol{w}^+) + \frac{2}{9n} D_h(\boldsymbol{w}_0, \boldsymbol{w}^+) \qquad (21)$$

$$\overset{(14)}{=} \mathbb{E} \, p \left\langle \boldsymbol{x} - \boldsymbol{w}_0, \nabla h(\boldsymbol{w}^+) - \nabla h(\boldsymbol{w}_0) \right\rangle - \frac{9pn - 2}{9n} D_h(\boldsymbol{w}_0, \boldsymbol{w}^+) - \frac{2\mu\theta}{5} D_h(\boldsymbol{x}, \boldsymbol{w}^+).$$

Thus, Eq. (11) is proved. $\qquad\square$

## D.2 Proof of Theorem 3.3

Proof: Choosing $\boldsymbol{x} = \boldsymbol{x}_*$ and $\boldsymbol{x} = \boldsymbol{w}_k$ in Eq. (21), which are all independent to indices $i_1, i_2 \ldots, i_T$ in $\mathrm{SVRS}^{\mathrm{1ep}}(f, \boldsymbol{w}_k, \theta, p)$, then we get

$$\mathbb{E}_k[f(\boldsymbol{w}_{k+1}) - f(\boldsymbol{x}_*)] \leq \mathbb{E}_k p D_h(\boldsymbol{x}_*, \boldsymbol{w}_k) - p D_h(\boldsymbol{x}_*, \boldsymbol{w}_{k+1}) - \frac{2\mu\theta}{5} D_h(\boldsymbol{x}_*, \boldsymbol{w}_{k+1}) + \frac{2}{9n} D_h(\boldsymbol{w}_k, \boldsymbol{w}_{k+1}).$$

$$\mathbb{E}_k[f(\boldsymbol{w}_{k+1}) - f(\boldsymbol{w}_k)] \leq \mathbb{E}_k p D_h(\boldsymbol{w}_k, \boldsymbol{w}_k) - p D_h(\boldsymbol{w}_k, \boldsymbol{w}_{k+1}) - \frac{2\mu\theta}{5} D_h(\boldsymbol{w}_k, \boldsymbol{w}_{k+1}) + \frac{2}{9n} D_h(\boldsymbol{w}_k, \boldsymbol{w}_{k+1}).$$

Adding both inequalities together, we could obtain

$$\mathbb{E}\left[2f(\boldsymbol{w}_{k+1}) - f(\boldsymbol{w}_k) - f(\boldsymbol{x}_*)\right] \leq \mathbb{E} p D_h(\boldsymbol{x}_*, \boldsymbol{w}_k) - \left(p + \frac{2\mu\theta}{5}\right) D_h(\boldsymbol{x}_*, \boldsymbol{w}_{k+1})$$
$$- \left(p + \frac{2\mu\theta}{5} - \frac{4}{9n}\right) D_h(\boldsymbol{w}_k, \boldsymbol{w}_{k+1}).$$

Noting that $p = 1/n$, thus $p + \frac{2\mu\theta}{5} - \frac{4}{9n} > 0$. Based on Eq. (9), after rearranging the terms, we get

$$\mathbb{E}[f(\boldsymbol{w}_{k+1}) - f(\boldsymbol{x}_*)] + \frac{1}{2}\left(p + \frac{2\mu\theta}{5}\right) D_h(\boldsymbol{x}_*, \boldsymbol{w}_{k+1}) \leq \mathbb{E} \frac{1}{2}[f(\boldsymbol{w}_k) - f(\boldsymbol{x}_*)] + \frac{p}{2} D_h(\boldsymbol{x}_*, \boldsymbol{w}_k).$$

Now we denote the potential function as

$$\Phi_k = \mathbb{E}[f(\boldsymbol{w}_k) - f(\boldsymbol{x}_*)] + \frac{1}{2}\left(p + \frac{2\mu\theta}{5}\right) D_h(\boldsymbol{x}_*, \boldsymbol{w}_k). \tag{22}$$

By $\theta = 1/(4\sqrt{n}\delta)$, we obtain

$$\mathbb{E}\Phi_{k+1} \leq \max\left\{1 - \frac{1}{2}, \left(1 + \frac{2\mu\theta}{5p}\right)^{-1}\right\} \mathbb{E}\Phi_k = \max\left\{1 - \frac{1}{2}, \left(1 + \frac{2\mu\sqrt{n}}{5\delta}\right)^{-1}\right\} \mathbb{E}\Phi_k.$$

When $\frac{2\mu\sqrt{n}}{5\delta} \geq 1$, we get $\mathbb{E}\Phi_{k+1} \leq \frac{1}{2}\mathbb{E}\Phi_k$. Otherwise, $\frac{2\mu\sqrt{n}}{5\delta} < 1$, by inequality $\frac{1}{1+x} \leq 1 - \frac{x}{2}, \forall 0 \leq x \leq 1$, we get $\left(1 + \frac{2\mu\sqrt{n}}{5\delta}\right)^{-1} \leq 1 - \frac{\mu\sqrt{n}}{5\delta}$. Hence, $\mathbb{E}\Phi_{k+1} \leq \left(1 - \frac{\mu\sqrt{n}}{5\delta}\right) \mathbb{E}\Phi_k$. Therefore, we obtain $\mathbb{E}\Phi_{k+1} \leq \max\left\{1 - \frac{1}{2}, 1 - \frac{\mu\sqrt{n}}{5\delta}\right\} \mathbb{E}\Phi_k$. Moreover, the initial term

$$\Phi_0 \overset{(22)}{=} f(\boldsymbol{w}_0) - f(\boldsymbol{x}_*) + \frac{1}{2}\left(p + \frac{2\mu\theta}{5}\right) D_h(\boldsymbol{x}_*, \boldsymbol{w}_0)$$

$$\overset{(9)}{\leq} f(\boldsymbol{w}_0) - f(\boldsymbol{x}_*) + \frac{1}{2}\left(p + \frac{2\mu\theta}{5}\right) \frac{1 + \sqrt{n}\theta\delta}{2\theta} \|\boldsymbol{x}_* - \boldsymbol{w}_0\|^2$$

$$\overset{(i)}{\leq} f(\boldsymbol{w}_0) - f(\boldsymbol{x}_*) + \frac{1}{2}\left(\frac{5\delta}{2\sqrt{n}} + \frac{\mu}{4}\right) \|\boldsymbol{w}_0 - \boldsymbol{x}_*\|^2 \overset{(ii)}{\leq} \left[1 + \frac{1}{2}\left(\frac{5\delta}{\mu\sqrt{n}} + \frac{1}{2}\right)\right] [f(\boldsymbol{w}_0) - f(\boldsymbol{x}_*)]$$

$$\leq 3\left(1 + \frac{\delta}{\mu\sqrt{n}}\right) [f(\boldsymbol{w}_0) - f(\boldsymbol{x}_*)],$$

where $(i)$ uses $\theta = \sqrt{p}/(4\delta)$ and $(ii)$ uses $\frac{\mu}{2} \|\boldsymbol{w}_0 - \boldsymbol{x}_*\|^2 \overset{(2)}{\leq} f(\boldsymbol{w}_0) - f(\boldsymbol{x}_*)$. Then we finally get

$$\mathbb{E} f(\boldsymbol{w}_k) - f(\boldsymbol{x}_*) \overset{(22)}{=} \mathbb{E}\Phi_k \leq \left(\max\left\{1 - \frac{1}{2}, 1 - \frac{\mu\sqrt{n}}{5\delta}\right\}\right)^k \cdot 3\left(1 + \frac{\delta}{\mu\sqrt{n}}\right) [f(\boldsymbol{w}_0) - f(\boldsymbol{x}_*)].$$

In order to make $\mathbb{E}\Phi_k \leq \varepsilon$, we need

$$\exp\left\{-\frac{k}{\max\left\{2, \frac{5\delta}{\mu\sqrt{n}}\right\}}\right\} \cdot 3\left(1 + \frac{\delta}{\mu\sqrt{n}}\right) [f(\boldsymbol{w}_0) - f(\boldsymbol{x}_*)] \leq \varepsilon,$$

which leads to $k \geq K_1 := \max\left\{2, \frac{5\delta}{\mu\sqrt{n}}\right\} \log \frac{3\left(1 + \frac{\delta}{\mu\sqrt{n}}\right)[f(\boldsymbol{w}_0) - f(\boldsymbol{x}_*)]}{\varepsilon}$.

Noting that one-epoch communication complexity in $\mathrm{SVRS}^{\mathrm{1ep}}$ is $\Theta(n)$ in expectation when $p = 1/n$ (shown in Section 3.1.1), we get total communication complexity is $\tilde{\mathcal{O}}(n + \sqrt{n}\delta/\mu)$. $\qquad\square$

## D.3 Proof of Lemma 3.5

Proof: Based on Lemma 3.1 and noting that $\boldsymbol{y}_{k+1} = \text{SVRS}^{1\text{ep}}(f, \boldsymbol{x}_{k+1}, \theta, p)$, we get

$$\mathbb{E}_k[f(\boldsymbol{y}_{k+1}) - f(\boldsymbol{x})] \overset{(11)}{\leq} \mathbb{E}_k p \langle \boldsymbol{x} - \boldsymbol{x}_{k+1}, \nabla h(\boldsymbol{y}_{k+1}) - \nabla h(\boldsymbol{x}_{k+1}) \rangle$$
$$- \frac{7p}{9} D_h(\boldsymbol{x}_{k+1}, \boldsymbol{y}_{k+1}) - \frac{2\mu\theta}{5} D_h(\boldsymbol{x}, \boldsymbol{y}_{k+1}). \tag{23}$$

Here $\boldsymbol{x} \in \mathbb{R}^d$ should be independent to random indices $i_1^{(k)}, i_2^{(k)}, \ldots, i_T^{(k)}$ in $\text{SVRS}^{1\text{ep}}(f, \boldsymbol{x}_{k+1}, \theta, p)$. Then we can apply interpolation $\boldsymbol{z}_k$ to derive

$$\mathbb{E}_k[f(\boldsymbol{y}_{k+1}) - f(\boldsymbol{x})] \overset{(23)}{\leq} \mathbb{E}_k \, p \langle \boldsymbol{z}_k - \boldsymbol{x}_{k+1}, \nabla h(\boldsymbol{y}_{k+1}) - \nabla h(\boldsymbol{x}_{k+1}) \rangle - \frac{7p}{9} D_h(\boldsymbol{x}_{k+1}, \boldsymbol{y}_{k+1})$$

$$+ p \langle \boldsymbol{x} - \boldsymbol{z}_k, \nabla h(\boldsymbol{y}_{k+1}) - \nabla h(\boldsymbol{x}_{k+1}) \rangle - \frac{2\mu\theta}{5} D_h(\boldsymbol{x}, \boldsymbol{y}_{k+1})$$

$$= \mathbb{E}_k \frac{1-\tau}{\tau} \cdot p \langle \boldsymbol{x}_{k+1} - \boldsymbol{y}_k, \nabla h(\boldsymbol{y}_{k+1}) - \nabla h(\boldsymbol{x}_{k+1}) \rangle - \frac{7p}{9} D_h(\boldsymbol{x}_{k+1}, \boldsymbol{y}_{k+1})$$

$$+ p \langle \boldsymbol{x} - \boldsymbol{z}_k, \nabla h(\boldsymbol{y}_{k+1}) - \nabla h(\boldsymbol{x}_{k+1}) \rangle - \frac{2\mu\theta}{5} D_h(\boldsymbol{x}, \boldsymbol{y}_{k+1})$$

$$\overset{(i)}{\leq} \mathbb{E}_k \frac{1-\tau}{\tau} \left[ f(\boldsymbol{y}_k) - f(\boldsymbol{y}_{k+1}) - \frac{7p}{9} D_h(\boldsymbol{x}_{k+1}, \boldsymbol{y}_{k+1}) \right] - \frac{7p}{9} D_h(\boldsymbol{x}_{k+1}, \boldsymbol{y}_{k+1})$$

$$+ p \langle \boldsymbol{z}_k - \boldsymbol{x}, \nabla h(\boldsymbol{x}_{k+1}) - \nabla h(\boldsymbol{y}_{k+1}) \rangle - \frac{2\mu\theta}{5} D_h(\boldsymbol{x}, \boldsymbol{y}_{k+1}) \tag{24}$$

where $(i)$ uses Eq. (23) with $\boldsymbol{x} = \boldsymbol{y}_k$, which is independent to indices in $\text{SVRS}^{1\text{ep}}(f, \boldsymbol{x}_{k+1}, \theta, p)$. We continue obtaining

$$\mathbb{E}_k[f(\boldsymbol{y}_{k+1}) - f(\boldsymbol{x})] \overset{(24)(9)}{\leq} \mathbb{E}_k \frac{1-\tau}{\tau} [f(\boldsymbol{y}_k) - f(\boldsymbol{y}_{k+1})] - \frac{7p}{9\tau} D_h(\boldsymbol{x}_{k+1}, \boldsymbol{y}_{k+1})$$

$$+ p \langle \boldsymbol{z}_k - \boldsymbol{x}, \nabla h(\boldsymbol{x}_{k+1}) - \nabla h(\boldsymbol{y}_{k+1}) \rangle - \frac{\mu(1 - \sqrt{n}\theta\delta)}{5} \|x - y_{k+1}\|^2$$

$$\overset{(i)}{\leq} \mathbb{E}_k \frac{1-\tau}{\tau} [f(\boldsymbol{y}_k) - f(\boldsymbol{y}_{k+1})] - \frac{7p}{9\tau} D_h(\boldsymbol{x}_{k+1}, \boldsymbol{y}_{k+1})$$

$$+ \mathbb{E}_k \left[ \mathbb{E}_{j_k} \langle \boldsymbol{z}_k - \boldsymbol{x}, \boldsymbol{\mathcal{G}}_{k+1} \rangle - \frac{3\mu}{20} \|\boldsymbol{x} - \boldsymbol{y}_{k+1}\|^2 \right]$$

$$\overset{(16)}{\leq} \mathbb{E}_k \frac{1-\tau}{\tau} [f(\boldsymbol{y}_k) - f(\boldsymbol{y}_{k+1})] - \frac{7p}{9\tau} D_h(\boldsymbol{x}_{k+1}, \boldsymbol{y}_{k+1})$$

$$+ \mathbb{E}_k \left[ \mathbb{E}_{j_k} \frac{\alpha}{2} \|\boldsymbol{\mathcal{G}}_{k+1}\|^2 + \frac{\|\boldsymbol{x} - \boldsymbol{z}_k\|^2}{2\alpha} - \frac{1 + 0.3\mu\alpha}{2\alpha} \|\boldsymbol{x} - \boldsymbol{z}_{k+1}\|^2 \right],$$

where $(i)$ uses $\mathbb{E}_{j_k} \boldsymbol{\mathcal{G}}_{k+1} = p[\nabla h(\boldsymbol{x}_{k+1}) - \nabla h(\boldsymbol{y}_{k+1})]$ and $\sqrt{n}\theta\delta = 1/4$.

Furthermore, we can estimate

$$\mathbb{E}_{j_k} \|\boldsymbol{\mathcal{G}}_{k+1}\|^2 = p^2 \mathbb{E}_{j_k} \|\nabla h(\boldsymbol{x}_{k+1}) - \nabla h(\boldsymbol{y}_{k+1}) + \nabla[f - f_{j_k}](\boldsymbol{x}_{k+1}) - \nabla[f - f_{j_k}](\boldsymbol{y}_{k+1})\|^2$$

$$\overset{(i)}{=} p^2 \mathbb{E}_{j_k} \|\nabla h(\boldsymbol{x}_{k+1}) - \nabla h(\boldsymbol{y}_{k+1})\|^2 + \|\nabla[f - f_{j_k}](\boldsymbol{x}_{k+1}) - \nabla[f - f_{j_k}](\boldsymbol{y}_{k+1})\|^2$$

$$\overset{(4)}{\leq} p^2 \mathbb{E}_{j_k} \|\nabla h(\boldsymbol{x}_{k+1}) - \nabla h(\boldsymbol{y}_{k+1})\|^2 + p^2\delta^2 \|\boldsymbol{x}_{k+1} - \boldsymbol{y}_{k+1}\|^2$$

$$\overset{(ii)}{\leq} \frac{2(1 + \sqrt{n}\theta\delta)p^2}{\theta} D_h(\boldsymbol{x}_{k+1}, \boldsymbol{y}_{k+1}) + \frac{2\theta p^2\delta^2}{1 - \sqrt{n}\theta\delta} D_h(\boldsymbol{x}_{k+1}, \boldsymbol{y}_{k+1})$$

$$= \frac{5p^2}{2\theta} D_h(\boldsymbol{x}_{k+1}, \boldsymbol{y}_{k+1}) + \frac{p^2}{6n\theta} D_h(\boldsymbol{x}_{k+1}, \boldsymbol{y}_{k+1}) \leq \frac{8p^2}{3\theta} D_h(\boldsymbol{x}_{k+1}, \boldsymbol{y}_{k+1})$$

where $(i)$ uses $\mathbb{E}_{j_k}\nabla[f-f_{j_k}](\boldsymbol{x}_{k+1})-\nabla[f-f_{j_k}](\boldsymbol{y}_{k+1})=\mathbf{0}$, $(ii)$ uses the convexity and smoothness of $h$ (e.g., [22, Theorem A.1 (iii)]) and Eq. (9). After rearrangement, we get

$$\mathbb{E}_k \frac{\alpha}{\tau}[f(\boldsymbol{y}_{k+1})-f(\boldsymbol{x})] \leq \mathbb{E}_k (1-\tau)\cdot\frac{\alpha}{\tau}[f(\boldsymbol{y}_k)-f(\boldsymbol{x})] + \frac{\|\boldsymbol{x}-\boldsymbol{z}_k\|^2}{2} - \frac{1+0.3\mu\alpha}{2}\|\boldsymbol{x}-\boldsymbol{z}_{k+1}\|^2$$
$$+\alpha\left(\frac{4\alpha p^2}{3\theta} - \frac{7p}{9\tau}\right)D_h(\boldsymbol{x}_{k+1},\boldsymbol{y}_{k+1}).$$

Hence, we see that once $2\tau\alpha p \leq \theta$, Eq. (12) holds. $\qquad\square$

### D.4 Proof of Theorem 3.6

Proof: Taking $\boldsymbol{x} = \boldsymbol{x}_*$ in Eq. (12), which is independent of any index during the process, we get

$$\mathbb{E}\frac{\alpha}{\tau}[f(\boldsymbol{y}_{k+1})-f(\boldsymbol{x}_*)]+\frac{(1+0.3\mu\alpha)\|\boldsymbol{x}_*-\boldsymbol{z}_{k+1}\|^2}{2} \leq \mathbb{E}(1-\tau)\cdot\frac{\alpha}{\tau}[f(\boldsymbol{y}_k)-f(\boldsymbol{x}_*)]+\frac{\|\boldsymbol{x}_*-\boldsymbol{z}_k\|^2}{2}.$$

Denote the potential function as

$$\Phi_k = [f(\boldsymbol{y}_k)-f(\boldsymbol{x}_*)] + \frac{\tau(1+0.3\mu\alpha)}{2\alpha}\|\boldsymbol{x}_*-\boldsymbol{z}_k\|^2.$$

We obtain

$$\mathbb{E}\Phi_{k+1} \leq \max\left\{1-\tau, \left(1+\frac{\mu\sqrt{n}}{\delta}\cdot\frac{3}{80\tau}\right)^{-1}\right\}\mathbb{E}\Phi_k.$$

When $\tau = \frac{1}{4} \leq \frac{1}{8}n^{1/4}\sqrt{\frac{\mu}{\delta}}$, then we have that

$$(1-\tau)\left(1+\frac{\mu\sqrt{n}}{\delta}\cdot\frac{3}{80\tau}\right) \geq (1-\tau)\left(1+\frac{3}{20\tau}\right) \geq 1 \Rightarrow \mathbb{E}\Phi_{k+1} \leq \left(1-\frac{1}{4}\right)\mathbb{E}\Phi_k.$$

When $\tau = \frac{n^{1/4}}{8}\sqrt{\frac{\mu}{\delta}} \leq \frac{1}{4}$, we get

$$t := \frac{\mu\sqrt{n}}{\delta}\cdot\frac{3}{80\tau} = \frac{3n^{1/4}}{10}\sqrt{\frac{\mu}{\delta}} \leq \frac{3}{5} \Rightarrow \frac{1}{1+t} \leq 1 - \frac{5t}{8} \Rightarrow \mathbb{E}\Phi_{k+1} \leq \left(1-\frac{n^{1/4}}{8}\sqrt{\frac{\mu}{\delta}}\right)\mathbb{E}\Phi_k.$$

Therefore, we finally obtain

$$\mathbb{E}\Phi_{k+1} \leq \max\left\{1-\frac{1}{4}, 1-\frac{n^{1/4}}{8}\sqrt{\frac{\mu}{\delta}}\right\}\mathbb{E}\Phi_k.$$

By the strong convexity of $f$ in Assumption 1 and the choice of $\tau$ and $\alpha$, the initial term

$$\begin{aligned}
\Phi_0 &= [f(\boldsymbol{y}_0)-f(\boldsymbol{x}_*)] + \frac{\tau(1+0.3\mu\alpha)}{2\alpha}\|\boldsymbol{x}_*-\boldsymbol{y}_0\|^2\\
&= [f(\boldsymbol{y}_0)-f(\boldsymbol{x}_*)] + \left(\frac{8\delta\tau^2}{\sqrt{n}\mu}+0.3\tau\right)\frac{\mu}{2}\|\boldsymbol{x}_*-\boldsymbol{y}_0\|^2\\
&\leq \left(1+\frac{1}{8}+\frac{0.3}{4}\right)[f(\boldsymbol{y}_0)-f(\boldsymbol{x}_*)] \leq 2[f(\boldsymbol{y}_0)-f(\boldsymbol{x}_*)].
\end{aligned}$$

To obtain $\varepsilon$-error solution, we need

$$k \geq K_2 = \max\left\{4, 8n^{-1/4}\sqrt{\frac{\delta}{\mu}}\right\}\log\frac{2[f(\boldsymbol{y}_0)-f(\boldsymbol{x}_*)]}{\varepsilon}.$$

Note that every call of Algorithm SVRS$^{\text{1ep}}$ requires $4n$ communication in expectation (shown in Section 3.1.1). The remaining communication in one iteration of AccSVRS need 4 communication (the master sends $\boldsymbol{x}_{k+1}$ and $\boldsymbol{y}_{k+1}$ to the client $j_k$, and then receives $\nabla f_{j_k}(\boldsymbol{x}_{k+1})$ and $\nabla f_{j_k}(\boldsymbol{y}_{k+1})$). Thus one iteration of AccSVRS is $\Theta(n)$ in expectation, leading to the total communication complexity for $\varepsilon$-error solution is $\tilde{\mathcal{O}}\left(n+n^{3/4}\sqrt{\frac{\delta}{\mu}}\right)$. $\qquad\square$

**Algorithm 4** Loopless Stochastic Variance-Reduced Sliding (SVRS)

---

1: **Input:** $\boldsymbol{w}_0 \in \mathbb{R}^d, p \in (0,1), \theta > 0, K \in \{1, 2, \dots\}$
2: Initialize $\boldsymbol{x}_0 = \boldsymbol{w}_0$ and compute $\nabla f(\boldsymbol{w}_0)$
3: **for** $k = 0, 1, 2, \dots, K-1$ **do**
4:    Sample $i_k \sim \text{Unif}([n])$ and compute $\boldsymbol{g}_k = \nabla f_{i_k}(\boldsymbol{w}_k) - \nabla f(\boldsymbol{w}_k)$
5:    Approximately solve the local proximal point problem:

$$\boldsymbol{x}_{k+1} \approx \arg\min_{\boldsymbol{x} \in \mathbb{R}^d} A_\theta^k(\boldsymbol{x}) := \langle \nabla f_{i_k}(\boldsymbol{x}_k) - \boldsymbol{g}_k - \nabla f_1(\boldsymbol{x}_k), \boldsymbol{x} - \boldsymbol{x}_k \rangle + \frac{1}{2\theta} \|\boldsymbol{x} - \boldsymbol{x}_k\|^2 + f_1(\boldsymbol{x})$$

6:    $\boldsymbol{w}_{k+1} = \begin{cases} \boldsymbol{x}_{k+1} & \text{with probability} \quad p \\ \boldsymbol{w}_k & \text{with probability} \quad 1-p \end{cases}$

7: **end for**
8: **Output:** $\boldsymbol{w}_K$

---

### D.5 Loopless SVRS

In this section, we describe the loopless SVRS (Algorithm 4). By simple facts shown in Proposition A.2, $\text{SVRS}^{1\text{ep}}(f, \boldsymbol{w}_k, \theta, p)$ can be viewed as the inter iteration until $\boldsymbol{w}_k$ in loopless SVRS is updated. Thus, the one-step variation in Lemma D.1 still holds. Hence, we can derive a similar convergence rate and communication complexity for loopless SVRS.

**Theorem D.2** *Suppose Assumption 1 holds. If in loopless SVRS (Algorithm 4), the hyperparameters are set as $\theta = 1/(4\sqrt{n}\delta), p = 1/n$, and the approximate solution in each proximal step satisfies Eq. (10), then for any error $\varepsilon > 0$, when*

$$k \geq K_1 := \max\left\{2n, \frac{11\sqrt{n}\delta}{\mu}\right\} \log \frac{3\left(1 + \frac{\delta}{\mu\sqrt{n}}\right)[f(\boldsymbol{x}_0) - f(\boldsymbol{x}_*)]}{\varepsilon},$$

*i.e., after $\tilde{\mathcal{O}}(n + \sqrt{n}\delta/\mu)$ communications, we can guarantee that $\mathbb{E}f(\boldsymbol{w}_k) - f(\boldsymbol{x}_*) \leq \varepsilon$.*

Proof: Noting that in each step of loopless SVRS, the anchor point is $\boldsymbol{w}_k$ instead of $\boldsymbol{w}_0$, thus Eq. (18) holds after replacing $\boldsymbol{w}_0$ to $\boldsymbol{w}_k$. Now choosing $\boldsymbol{x} = \boldsymbol{x}_*$ and $\boldsymbol{x} = \boldsymbol{w}_k$ in Eq. (18), which are all independent to index $i_k$, we get

$$\mathbb{E}_k[f(\boldsymbol{x}_{k+1}) - f(\boldsymbol{x}_*)] \leq \mathbb{E}_k D_h(\boldsymbol{x}_*, \boldsymbol{x}_k) - \left(1 + \frac{\mu\theta/2}{1 + \sqrt{n}\theta\delta}\right) D_h(\boldsymbol{x}_*, \boldsymbol{x}_{k+1}) + \frac{2\theta^2\delta^2}{(1 - \sqrt{n}\theta\delta)^2} D_h(\boldsymbol{w}_k, \boldsymbol{x}_k).$$

$$\mathbb{E}_k[f(\boldsymbol{x}_{k+1}) - f(\boldsymbol{w}_k)] \leq \mathbb{E}_k D_h(\boldsymbol{w}_k, \boldsymbol{x}_k) - \left(1 + \frac{\mu\theta/2}{1 + \sqrt{n}\theta\delta}\right) D_h(\boldsymbol{w}_k, \boldsymbol{x}_{k+1}) + \frac{2\theta^2\delta^2}{(1 - \sqrt{n}\theta\delta)^2} D_h(\boldsymbol{w}_k, \boldsymbol{x}_k).$$

Adding both inequalities together and noting that

$$\mathbb{E}_k D_h(\boldsymbol{w}_{k+1}, \boldsymbol{x}_{k+1}) = \mathbb{E}_k(1-p) D_h(\boldsymbol{w}_k, \boldsymbol{x}_{k+1}) + p D_h(\boldsymbol{x}_{k+1}, \boldsymbol{x}_{k+1}) = (1-p)\mathbb{E}_k D_h(\boldsymbol{w}_k, \boldsymbol{x}_{k+1}),$$

as well as

$$\mathbb{E}_k f(\boldsymbol{w}_{k+1}) = \mathbb{E}_k(1-p)f(\boldsymbol{w}_k) + pf(\boldsymbol{x}_{k+1}),$$

we could obtain

$$\mathbb{E} \frac{2}{p}[f(\boldsymbol{w}_{k+1}) - (1-p)f(\boldsymbol{w}_k)] - f(\boldsymbol{w}_k) - f(\boldsymbol{x}_*)$$

$$\leq \mathbb{E} D_h(\boldsymbol{x}_*, \boldsymbol{x}_k) - \left(1 + \frac{\mu\theta/2}{1 + \sqrt{n}\theta\delta}\right) D_h(\boldsymbol{x}_*, \boldsymbol{x}_{k+1}) + \left(1 + \frac{4\theta^2\delta^2}{(1 - \sqrt{n}\theta\delta)^2}\right) D_h(\boldsymbol{w}_k, \boldsymbol{x}_k) - D_h(\boldsymbol{w}_k, \boldsymbol{x}_{k+1})$$

$$= \mathbb{E} D_h(\boldsymbol{x}_*, \boldsymbol{x}_k) - \left(1 + \frac{\mu\theta/2}{1 + \sqrt{n}\theta\delta}\right) D_h(\boldsymbol{x}_*, \boldsymbol{x}_{k+1}) + \left(1 + \frac{4\theta^2\delta^2}{(1 - \sqrt{n}\theta\delta)^2}\right) D_h(\boldsymbol{w}_k, \boldsymbol{x}_k) - \frac{D_h(\boldsymbol{w}_{k+1}, \boldsymbol{x}_{k+1})}{1 - p}.$$

Rearranging the terms, we get

$$\mathbb{E}[f(\boldsymbol{w}_{k+1}) - f(\boldsymbol{x}_*)] + \frac{p}{2}\left(1 + \frac{\mu\theta/2}{1 + \sqrt{n}\theta\delta}\right) D_h(\boldsymbol{x}_*, \boldsymbol{x}_{k+1}) + \frac{p}{2(1-p)} D_h(\boldsymbol{w}_{k+1}, \boldsymbol{x}_{k+1})$$

$$\leq \quad \mathbb{E}(1 - \frac{p}{2})[f(\boldsymbol{w}_k) - f(\boldsymbol{x}_*)] + \frac{p}{2} D_h(\boldsymbol{x}_*, \boldsymbol{x}_k) + \frac{p}{2}\left(1 + \frac{4\theta^2\delta^2}{(1 - \sqrt{n}\theta\delta)^2}\right) D_h(\boldsymbol{w}_k, \boldsymbol{x}_k).$$

Now we denote the potential function as

$$\Phi_k = \mathbb{E}[f(\boldsymbol{w}_k) - f(\boldsymbol{x}_*)] + \frac{p}{2}\left(1 + \frac{\mu\theta/2}{1 + \sqrt{n}\theta\delta}\right) D_h(\boldsymbol{x}_*, \boldsymbol{x}_k) + \frac{p}{2(1-p)} D_h(\boldsymbol{w}_k, \boldsymbol{x}_k).$$

Then we obtain

$$\mathbb{E}\Phi_{k+1} \le \max\left\{1 - \frac{p}{2}, \left(1 + \frac{\mu\theta/2}{1 + \sqrt{n}\theta\delta}\right)^{-1}, \left(1 + \frac{4\theta^2\delta^2}{(1 - \sqrt{n}\theta\delta)^2}\right)(1-p)\right\}\mathbb{E}\Phi_k.$$

Since we choose $\theta = 1/(4\sqrt{n}\delta)$, we get $\theta\mu \le \sqrt{n}\theta\delta \le 1/4$ by Assumption 1, which shows that

$$\left(1 + \frac{\mu\theta/2}{1 + \sqrt{n}\theta\delta}\right)^{-1} = 1 - \frac{\mu\theta/2}{1 + \sqrt{n}\theta\delta + \mu\theta/2} = 1 - \frac{4\mu\theta}{11} = 1 - \frac{\mu}{11\delta\sqrt{n}}.$$

Additionally, by $p = 1/n$ and $\theta = 1/(4\delta\sqrt{n})$, we also have that

$$\left(1 + \frac{4\theta^2\delta^2}{(1 - \sqrt{n}\theta\delta)^2}\right)(1-p) = \left(1 + \left(\frac{\frac{1}{2\sqrt{n}}}{1 - \frac{1}{4}}\right)^2\right)(1-p) = \left(1 + \frac{4p}{9}\right)(1-p) \le 1 - \frac{5p}{9}.$$

Therefore, we obtain the ratio between $\mathbb{E}\Phi_{k+1}$ and $\mathbb{E}\Phi_k$:

$$\mathbb{E}\Phi_{k+1} \le \max\left\{1 - \frac{p}{2}, 1 - \frac{\mu}{11\delta\sqrt{n}}, 1 - \frac{5p}{9}\right\}\mathbb{E}\Phi_k \le \max\left\{1 - \frac{p}{2}, 1 - \frac{\mu}{11\delta\sqrt{n}}\right\}\mathbb{E}\Phi_k.$$

Moreover, the initial term

$$\Phi_0 = f(\boldsymbol{w}_0) - f(\boldsymbol{x}_*) + \frac{p}{2}\left(1 + \frac{\mu\theta/2}{1 + \sqrt{n}\theta\delta}\right) D_h(\boldsymbol{x}_*, \boldsymbol{x}_0)$$

$$\overset{(9)}{\le} f(\boldsymbol{x}_0) - f(\boldsymbol{x}_*) + \frac{p}{2}\left(1 + \frac{\mu\theta/2}{1 + \sqrt{n}\theta\delta}\right)\frac{1 + \sqrt{n}\theta\delta}{2\theta}\|\boldsymbol{x}_* - \boldsymbol{x}_0\|^2$$

$$\overset{(i)}{\le} f(\boldsymbol{x}_0) - f(\boldsymbol{x}_*) + \frac{p}{2}\left(\frac{2.5\delta}{\sqrt{p}} + \frac{\mu}{4}\right)\|\boldsymbol{x}_0 - \boldsymbol{x}_*\|^2 \overset{(ii)}{\le} \left[1 + \frac{p}{2}\left(\frac{5\delta}{\mu\sqrt{p}} + \frac{1}{2}\right)\right][f(\boldsymbol{x}_0) - f(\boldsymbol{x}_*)]$$

$$\le 3\left(1 + \frac{\delta}{\mu\sqrt{n}}\right)[f(\boldsymbol{x}_0) - f(\boldsymbol{x}_*)],$$

where $(i)$ uses $\theta = \sqrt{p}/(4\delta)$ and $(ii)$ uses $f(\boldsymbol{x}_0) - f(\boldsymbol{x}_*) \overset{(2)}{\ge} \frac{\mu}{2}\|\boldsymbol{x}_0 - \boldsymbol{x}_*\|^2$. Then we finally get

$$\mathbb{E}f(\boldsymbol{w}_k) - f(\boldsymbol{x}_*) \le \mathbb{E}\Phi_k \le \left(\max\left\{1 - \frac{1}{2n}, 1 - \frac{\mu}{11\delta\sqrt{n}}\right\}\right)^k \cdot 3\left(1 + \frac{\delta}{\mu\sqrt{n}}\right)[f(\boldsymbol{x}_0) - f(\boldsymbol{x}_*)].$$

In order to make $\mathbb{E}\Phi_k \le \varepsilon$, we need

$$\exp\left\{-\frac{k}{\max\left\{2n, \frac{11\sqrt{n}\delta}{\mu}\right\}}\right\} \cdot 3\left(1 + \frac{\delta}{\mu\sqrt{n}}\right)[f(\boldsymbol{x}_0) - f(\boldsymbol{x}_*)] \le \varepsilon,$$

which leads to

$$k \ge K_1 := \max\left\{2n, \frac{11\sqrt{n}\delta}{\mu}\right\}\log\frac{3\left(1 + \frac{\delta}{\mu\sqrt{n}}\right)[f(\boldsymbol{x}_0) - f(\boldsymbol{x}_*)]}{\varepsilon},$$

Noting that communication complexity in each iteration is $2p(n-1) + 2$ in expectation (by similar analysis in Section 3.1.1), we get communication complexity is $4$ in each iteration in expectation. Therefore, the total communication complexity is $\tilde{\mathcal{O}}(n + \frac{\sqrt{n}\delta}{\mu})$ in expectation. □

# E  Computation of Gradient Complexity

We show the detail omitted in Section 3.3. Let $\boldsymbol{x}_{t,*} = \arg\min_{\boldsymbol{x}\in\mathbb{R}^d} A_\theta^t(\boldsymbol{x})$. Noting that by [47, Theorem 2.2.2], we could obtain

$$\left\|\nabla A_\theta^t(\boldsymbol{x}_{t,s})\right\|^2 \leq 2L'\left(A_\theta^t(\boldsymbol{x}_{t,s}) - A_\theta^t(\boldsymbol{x}_{t,*})\right) \leq \frac{20\mu'L'\left\|\boldsymbol{x}_t - \boldsymbol{x}_{t,*}\right\|^2}{3}\left[e^{(s+1)/\sqrt{\kappa'}} - 1\right]^{-1}$$

if we start from $\boldsymbol{x}_t$ in the proximal step for optimizing $A_\theta^t(\boldsymbol{x})$, where $\kappa' = L'/\mu', L' = L+1/\theta, \mu' = -\sqrt{n}\delta + 1/\theta$ based on assumptions. Then Eq. (10) could be satisfied after $T_{\text{app}}$ iterations when

$$\frac{20\mu'L'\left\|\boldsymbol{x}_t - \boldsymbol{x}_{t,*}\right\|^2}{3}\left[e^{(T_{\text{app}}+1)/\sqrt{\kappa'}} - 1\right]^{-1} \leq \frac{\mu}{20\theta}\left\|\boldsymbol{x}_t - \boldsymbol{x}_{t,*}\right\|^2.$$

Note that $\theta = 1/(4\sqrt{n}\delta)$, which leads to

$$T_{\text{app}} = \mathcal{O}\left(\sqrt{\frac{1+\theta L}{1-\sqrt{n}\theta\delta}} \cdot \log\left(\frac{(1+\theta L)(1-\sqrt{n}\theta\delta)}{\mu\theta}\right)\right) = \mathcal{O}\left(\left(1+n^{-1/4}\sqrt{\delta/\mu}\right)\log\frac{\sqrt{n}\delta+L}{\mu}\right).$$

Hence, the total number of gradient calls in expectation is

$$\mathcal{O}(nT_{\text{app}}\cdot K_2) = \tilde{\mathcal{O}}\left[\left(n+n^{3/4}\sqrt{\frac{\delta}{\mu}}\right)\left(1+\frac{1}{n^{1/4}}\sqrt{\frac{L}{\delta}}\right)\right] = \tilde{\mathcal{O}}\left(n+n^{3/4}\left(\sqrt{\frac{\delta}{\mu}}+\sqrt{\frac{L}{\delta}}\right)+\sqrt{\frac{nL}{\mu}}\right).$$

Since $\delta \in [\mu, L]$, we obtain $\sqrt{\frac{\delta}{\mu}} + \sqrt{\frac{L}{\delta}} \leq \sqrt{\frac{L}{\mu}} + 1$, leading to

$$n + n^{3/4}\left(\sqrt{\frac{\delta}{\mu}}+\sqrt{\frac{L}{\delta}}\right)+\sqrt{\frac{nL}{\mu}} \leq 2\left(n+n^{3/4}\sqrt{\frac{L}{\mu}}\right).$$

Thus, the gradient complexity is $\tilde{\mathcal{O}}\left(n+n^{3/4}\sqrt{L/\mu}\right)$. Moreover, when $\delta = \Theta(\sqrt{\mu L})$, we obtain

$$n + n^{3/4}\left(\sqrt{\frac{\delta}{\mu}}+\sqrt{\frac{L}{\delta}}\right) = n + \Theta\left(n^{3/4}\left(\frac{L}{\mu}\right)^{1/4}\right)+\sqrt{\frac{nL}{\mu}} = \Theta\left(n+\sqrt{\frac{nL}{\mu}}\right).$$

Thus, the gradient complexity is $\tilde{\mathcal{O}}\left(n+\sqrt{nL/\mu}\right)$ in this time.

Note that Assumption 1 and smoothness of $f_1$ could only guarantee

$$\frac{1}{n}\sum_{i=1}^{n}\left\|\nabla f_i(\boldsymbol{x}) - \nabla f_i(\boldsymbol{y})\right\|^2 \overset{(4)}{\leq} \delta^2 + \left\|\nabla f(\boldsymbol{x}) - \nabla f(\boldsymbol{y})\right\|^2$$

$$\leq \delta^2 + 2\left\|\nabla[f-f_1](\boldsymbol{x})-\nabla[f-f_1](\boldsymbol{y})\right\|^2 + 2\left\|\nabla f_1(\boldsymbol{x})-\nabla f_1(\boldsymbol{y})\right\|^2 \leq \left[(2n+1)\delta^2+2L^2\right]\left\|\boldsymbol{x}-\boldsymbol{y}\right\|^2,$$

that is, $f_i$'s are $(2L + 2\sqrt{n}\delta)$-average smooth. Hence, the tightness of our gradient complexity holds for the average smooth setting only when $\sqrt{n}\delta = \mathcal{O}(L)$. Moreover, we can also compute

$$\left\|\nabla f_i(\boldsymbol{x}) - \nabla f_i(\boldsymbol{y})\right\|^2 \leq 2\left\|\nabla[f - f_i](\boldsymbol{x}) - \nabla[f - f_i](\boldsymbol{y})\right\|^2 + 2\left\|\nabla f(\boldsymbol{x}) - \nabla f(\boldsymbol{y})\right\|^2$$

$$\overset{(4)}{\leq} 2n\delta^2+4\left\|\nabla[f-f_1](\boldsymbol{x})-\nabla[f-f_1](\boldsymbol{y})\right\|^2 +4\left\|\nabla f_1(\boldsymbol{x})-\nabla f_1(\boldsymbol{y})\right\|^2 \leq \left[6n\delta^2+4L^2\right]\left\|\boldsymbol{x}-\boldsymbol{y}\right\|^2,$$

that is, $f_i$'s are $(2L + 3\sqrt{n}\delta)$-smooth. Hence, the tightness of our gradient complexity holds for the component smooth setting only when $\delta = \Theta(\sqrt{\mu L})$ and $n\mu = \mathcal{O}(L)$.

# F  Omitted Details of Section 4

In this section, we give the omitted details of Section 4 as well as their proofs.

## F.1  Formal Statement of Definition 4.1 and Discussion

In this subsection, we give the formal statement of Definition 4.1 and show that Algorithm 2 satisfies our definition.

We first introduce the two oracles: the incremental first-order oracle (IFO) [2, 63] and the Proximal Incremental First-order Oracle (PIFO)[11] [56, 25], which are defined as $h_{f_i}^{\mathrm{I}}(\boldsymbol{x}) = [f_i(\boldsymbol{x}), \nabla f_i(\boldsymbol{x})]$

---
[11]Although we have defined PIFO in Section 4.1, we restate it here for completement.

and $h_{f_i}^{\mathrm{P}}(\boldsymbol{x}, \gamma) = [f_i(\boldsymbol{x}), \nabla f_i(\boldsymbol{x}), \mathrm{prox}_{f_i}^{\gamma}(\boldsymbol{x})]$ with $\gamma > 0$ respectively. Here the proximal operator is

$$\mathrm{prox}_{f_i}^{\gamma}(\boldsymbol{x}) := \arg\min_{\boldsymbol{u}} \left\{ f_i(\boldsymbol{u}) + \frac{1}{2\gamma} \|\boldsymbol{x} - \boldsymbol{u}\|^2 \right\} = \arg\min_{\boldsymbol{u}} \left\{ \gamma f_i(\boldsymbol{u}) + \frac{1}{2} \|\boldsymbol{x} - \boldsymbol{u}\|^2 \right\}.$$

The IFO $h_{f_i}^{\mathrm{I}}(\boldsymbol{x})$ takes a point $\boldsymbol{x}$ and a component $f_i$ as input and returns the zero-order and first-order information of the component at $\boldsymbol{x}$. The PIFO $h_{f_i}^{\mathrm{P}}(\boldsymbol{x}, \gamma)$ has an additional input $\gamma > 0$, which can be viewed as the step size of the proximal operator. Besides the local zero-order and first-order information returned by $h_{f_i}^{\mathrm{I}}(\boldsymbol{x})$, $h_{f_i}^{\mathrm{P}}(\boldsymbol{x}, \gamma)$ also provides some global information of $f_i$ by means of the proximal operator. To see this, if we let $\gamma \to +\infty$, $\mathrm{prox}_{f_i}^{\gamma}(\boldsymbol{x})$ converges to the exact minimizer of $f_i$, irrelevant to the choice of $\boldsymbol{x}$. In practice, it could be hard to compute $\mathrm{prox}_{f_i}^{\gamma}(\boldsymbol{x})$ precisely. Nevertheless, since we only focus on communication complexity, it makes no difference to distinguish between the IFO and the PIFO[12]. Thus we assume the algorithm has access to the PIFO and the definition is as follows.

**Definition F.1 (Formal version of Definition 4.1)** *Consider a randomized algorithm $\mathcal{A}$ to solve problem* (1). *Suppose the number of communication rounds is $T$. Define information sets $\mathcal{I}_{t+1}$, $\mathcal{I}_{t+1}^0$ and $\mathcal{I}_{t+1}^1$. Here $\mathcal{I}_{t+1}$ denotes all the information $\mathcal{A}$ obtains after round $t$, while $\mathcal{I}_{t+1}^0$ and $\mathcal{I}_{t+1}^1$ denote the information before and after (possible) anchor point updating during round $t$, respectively. The algorithm updates the information set by the following procedure.*

1. *Choose a distribution $\mathcal{D}$ over $[n]$ with $q_i = \mathbb{P}_{Z \sim \mathcal{D}}(Z = i) > 0$, a positive number $p \le c_0/n$[13] and the initial points $\boldsymbol{x}_0$. Specify a master note 1 and assume $\max_{2 \le i \le n} q_i \le q_0/n$. Node 1 sends $\boldsymbol{x}_0$ to all the other nodes and other nodes send $h_{f_i}^{\mathrm{P}}(\boldsymbol{x}_0, \gamma_0)$ back to node 1. Initialize the information set as $\mathcal{I}_0 := \mathrm{span}\{\boldsymbol{x}_0, \nabla f_i(\boldsymbol{x}_0), \mathrm{prox}_{f_i}^{\gamma_0}(\boldsymbol{x}_0) \mid 1 \le i \le n\}$ and set $t = 0$ and $\tilde{\boldsymbol{x}}_0 = \boldsymbol{x}_0$.*

2. *Sample $i_t \sim \mathcal{D}$. Node 1 sends $\tilde{\boldsymbol{x}}_t$ to node $i_t$ and node $i_t$ sends $h_{f_{i_t}}^{\mathrm{P}}(\tilde{\boldsymbol{x}}_t, \gamma_t)$ back to node 1. Update the information set*

$$\mathcal{I}_{t+1}^0 := \mathrm{span}\{\boldsymbol{y}, \nabla f_{i_t}(\tilde{\boldsymbol{x}}_t), \mathrm{prox}_{f_{i_t}}^{\gamma_t}(\tilde{\boldsymbol{x}}_t) \mid \boldsymbol{y} \in \mathcal{I}_t\} \tag{25}$$

3. *Update the information set $\mathcal{I}_{t+1}^1$ and choose $\boldsymbol{x}_{t+1} \in \mathcal{I}_{t+1}^1$ following the linear-span protocol*

$$\boldsymbol{x}_{t+1} \in \mathcal{I}_{t+1}^1 := \mathrm{span}\{\boldsymbol{y}, \nabla f_1(\boldsymbol{z}), \mathrm{prox}_{f_1}^{\gamma_t'}(\boldsymbol{w}) \mid \boldsymbol{y}, \boldsymbol{z}, \boldsymbol{w} \in \mathcal{I}_{t+1}^0\}. \tag{26}$$

4. *Sample a Bernoulli random variable $a_t$ with expectation equal to $p$. If $a_t = 1$, go to step 5 (update the anchor point); otherwise, set $\tilde{\boldsymbol{x}}_{t+1} = \boldsymbol{x}_{t+1}$, $\mathcal{I}_{t+1} = \mathcal{I}_{t+1}^1$ and go to step 6 (do not update the anchor point).*

5. *Sample $j_t \sim \mathcal{D}$. Node 1 sends some $\boldsymbol{y}_{t+1} \in \mathcal{I}_{t+1}^1$ to node $j_t$ and node $j_t$ sends $h_{f_{j_t}}^{\mathrm{P}}(\boldsymbol{y}_{t+1}, \gamma_{t+1}'')$ back to node 1. Obtain the anchor point $\tilde{\boldsymbol{y}}_{t+1}$ by*

$$\tilde{\boldsymbol{y}}_{t+1} \in \mathrm{span}\{\boldsymbol{y}, \nabla f_1(\boldsymbol{z}), \mathrm{prox}_{f_1}^{\gamma_t'}(\boldsymbol{w}), \nabla f_{j_t}(\boldsymbol{y}_{t+1}), \mathrm{prox}_{f_{j_t}}^{\gamma_{t+1}''}(\boldsymbol{y}_{t+1}) \mid \boldsymbol{y}, \boldsymbol{z}, \boldsymbol{w} \in \mathcal{I}_{t+1}^1\}. \tag{27}$$

*Then node 1 sends the anchor point $\tilde{\boldsymbol{y}}_{t+1}$ to all the other nodes and other nodes send $h_{f_i}^{\mathrm{P}}(\tilde{\boldsymbol{y}}_{t+1}, \gamma_{t+1})$ back to node 1. Update the information set and obtain $\tilde{\boldsymbol{x}}_{t+1}$ by*

$$\tilde{\boldsymbol{x}}_{t+1} \in \mathcal{I}_{t+1} := \mathrm{span}\{\boldsymbol{y}, \tilde{\boldsymbol{y}}_{t+1}, \nabla f_i(\tilde{\boldsymbol{y}}_{t+1}), \mathrm{prox}_{f_i}^{\gamma_{t+1}}(\tilde{\boldsymbol{y}}_{t+1}) \mid \boldsymbol{y} \in \mathcal{I}_{t+1}^1, 1 \le i \le n\}. \tag{28}$$

6. *If $t = T - 1$, output some point in $\mathcal{I}_T$; otherwise, set $t \leftarrow t + 1$ and go back to step 2.*

*Here all the random variables $i_t$, $j_t$ and $a_t$ with $0 \le t \le T - 1$ are mutually independent, and the step sizes of the proximal operator $\gamma_t$, $\gamma_t'$ and $\gamma_t''$ are positive numbers.*

---

[12]See Lemma F.2

[13]To include catalyst accelerated algorithms, we also need $p \ge c_1/n$ for some $c_1 > 0$ (see footnote 17). To analyze Algorithm 2, $p \le c_0/n$ is enough.

Now we explain this definition and show that Algorithm 2 (with Algorithm 1 as a part) satisfies our definition.

**Initialization.** In our definition, step 1 is the initialization step. Without loss of generality, we can assume $x_0 = 0$ and node 1 is the master node. Otherwise, it suffices to consider $\{\tilde{f}_i(x) = f_i(x + x_0)\}_{i=1}^n$ and exchange the indices between node 1 and the master node. In Algorithm 2, the distribution $\mathcal{D}$ is $\mathrm{Unif}([n])$[14], and $p = 1/n$. In the initialization stage, the algorithm needs to calculate the full gradient of the initial point $x_0$, whose communication cost is $2(n-1)$.

We note that Definition F.1 enjoys a loopless structure while Algorithm 2 has two loops. In fact, when $p$ is fixed, a loopless algorithm is equivalent to a two-loop one with the inner loop size obeying $\mathrm{Geom}(p)$[15].

**Analysis of one communication round.** In each communication round, whether to calculate the full gradient depends on a coin toss with success probability $p$, as shown in step 4.

**The case $a_t = 0$.** We first focus on the case where the full gradient need not be calculated. Such a scenario corresponds to an iteration of Algorithm 1. Each communication round start with step 2. In this step, the algorithm samples a local node, with which the master node communicates. And the communication cost is 2. $\tilde{x}_t$ in this step corresponds to $x_t$ in Algorithm 1. In step 3, the master node calculates the next point based on the current information set $\mathcal{I}_{t+1}^0$ as well as the PIFO $h_{f_1}^{\mathrm{P}}$. This corresponds to line 7 in Algorithm 1. Indeed, the subproblem (7) can be rewritten as finding

$$\arg\min_{x \in \mathbb{R}^d} A_\theta^t(x) = \arg\min_{x \in \mathbb{R}^d} \left\{ \frac{1}{2\theta} \|x - x_t + \theta[\nabla f_{i_t}(x_t) - g_t - \nabla f_1(x_t)]\|^2 + f_1(x) \right\}$$
$$= \mathrm{prox}_{f_1}^\theta (x_t - \theta[\nabla f_{i_t}(x_t) - g_t - \nabla f_1(x_t)]) .$$

If the algorithm has access to the PIFO $h_{f_1}^{\mathrm{P}}$, then the subproblem (7) can be exactly solved by one step of (26). Otherwise, one can apply (26) recursively without the proximal information (i.e., only using the IFO $h_{f_1}^{\mathrm{I}}$), e.g., (accelerated) gradient methods, to find an approximate solution of (7)[16]

**The case $a_t = 1$.** When $a_t = 1$ in step 4, the algorithm needs to perform step 5, which corresponds to an outer iteration of Algorithm 2. Before calculating the full gradient, the algorithm first samples a local node $j_t$ again and the master node communicates the information about $y_{t+1}$ with this node. Here $y_{t+1}$ corresponds to $y_{k+1}$ in Algorithm 2, and the communication cost is 2. Then the master node calculates $\tilde{y}_{t+1}$ by (27), which corresponds to $x_{k+1}$ (of the next iteration) in Algorithm 2. That is to say, lines 7, 8 and 4 (of the next iteration) in Algorithm 2 can be summarized as (27). Then the master node communicates with all the other nodes the information about $\tilde{x}_{t+1}$, and the communication cost is $2(n-1)$. In (28), the algorithm picks up $\tilde{x}_{t+1}$ as the starting point of the next round. In Algorithm 2, $\tilde{x}_{t+1}$ is the same to $\tilde{y}_{t+1}$.

**Communication cost.** From the above analysis, the communication cost in step 5 is $2(n-1)$. Since we assume $q \leq c_0/n$, step 5 is performed infrequently and the expected communication cost is (at most) $2(n-1) \cdot p \approx 2c_0$ for a sufficiently large $n$. As a result, the total communication cost of a round is roughly $2 + 2c_0$ in expectation. After $T$ rounds, the expected communication cost is roughly $2(n-1) + 2(1+c_0)T$. As a result, we can use the number of rounds to measure communication complexity.

**The linear-span protocol and information set.** In Definition F.1, we focus on loopless algorithms based on the linear-span protocol. One can check that many methods, e.g., KatyushaX [6], L-SVRG and L-Katyusha [34], Loopless SARAH [42] and SVRP[17] [33], satisfy our definition. And this class of algorithms is sufficiently large in that the upper and lower bounds have matched for most cases

---

[14]When analyzing computational complexity, this distribution can also depend on the smoothness of each component function [57, 6]

[15]See Proposition A.2.

[16]Such a modification makes no difference to subsequent analysis. See Remark F.3.

[17]For Catalyzed SVRP in their paper, we can slightly modify it without affecting the gradient or communication complexity. Specifically, we remove the full gradient step at the beginning of the inner loop and do not update the current point until the full gradient is calculated. The number of additional communication rounds is $1/p = \Theta(n)$ in expectation, as long as $p = \Theta(1/n)$. Since in each inner loop, the algorithm must calculate the full gradient, whose gradient or communication complexity is also $\Theta(n)$, such a modification would not affect the total complexity.

[25]. Built on the linear-span protocol, the information set $\mathcal{I}_{t+1}$, a linear subspace of the whole space, gathers all the gradient and proximal information obtained by $t$ rounds of communication and includes all the possible points generated by the algorithm after round $t$. Clearly, the sequence $\{\mathcal{I}_t\}_{t=0}^T$ is nondecreasing in the sense that $\mathcal{I}_t \subseteq \mathcal{I}_{t'}$ for any $t' > t$.

## F.2 Details of Section 4.2

Recall that in Section 4.2, we consider the following class of matrices

$$
\boldsymbol{B}(m,\zeta) = \begin{bmatrix} 1 & -1 & & & \\ & 1 & -1 & & \\ & & \ddots & \ddots & \\ & & & 1 & -1 \\ & & & & \zeta \end{bmatrix} \in \mathbb{R}^{m \times m}.
$$

And one can check the matrix $\boldsymbol{A}(m,\zeta)$ is a tridiagonal matrix, i.e.,

$$
\boldsymbol{A}(m,\zeta) := \boldsymbol{B}(m,\zeta)^\top \boldsymbol{B}(m,\zeta) := \begin{bmatrix} 1 & -1 & & & \\ -1 & 2 & -1 & & \\ & \ddots & \ddots & \ddots & \\ & & -1 & 2 & -1 \\ & & & -1 & \zeta^2+1 \end{bmatrix} \in \mathbb{R}^{m \times m}.
$$

With the hard instance constructed in (13), we have the following lemma, which is a modification of Lemma 6.1 in Han et al. [25] in that the partitions of the index sets are slightly different.

**Lemma F.2** *Suppose that $n \geq 3$, $m \geq 3$, $\gamma$ is an arbitrarily positive number and $\boldsymbol{x} \in \mathcal{F}_k$ for some $0 \leq k < m$. If $k = 0$, we have*

$$
\nabla r_i(\boldsymbol{x}),\ \mathrm{prox}_{r_i}^\gamma(\boldsymbol{x}) \in \begin{cases} \mathcal{F}_1, & \text{if } i = 1, \\ \mathcal{F}_0, & \text{otherwise.} \end{cases}
$$

*If $k > 0$, we have*

$$
\nabla r_i(\boldsymbol{x}),\ \mathrm{prox}_{r_i}^\gamma(\boldsymbol{x}) \in \begin{cases} \mathcal{F}_{k+1}, & \text{if } k \in \mathcal{L}_i, \\ \mathcal{F}_k, & \text{otherwise.} \end{cases}
$$

*Here $\{\mathcal{F}_k\}_{k=0}^m$ are defined as $\mathcal{F}_0 = \{\boldsymbol{0}\}$ and $\mathcal{F}_k = \mathrm{span}\{\boldsymbol{e}_1, \boldsymbol{e}_2, \ldots, \boldsymbol{e}_k\}$ for $1 \leq k \leq m$, and we omit the parameters of $r_i$ to simplify the notation.*

Lemma F.2 tells us that if the current point $\boldsymbol{x}$ lies in some subspace of $\mathbb{R}^m$, only one component can provide the information of the next dimension by gradient or proximal information. In this sense, PIFO cannot provide more information than IFO. Thus, when we focus on the communication complexity of an algorithm, it makes no difference to distinguish between IFO and PIFO. And we can assume the algorithm has access to PIFO without loss of generality. Moreover, when $k > 0$, the oracle of $r_1$ can never provide any information on the next dimension. The proof of Lemma F.2 is deferred to Appendix F.4.

With Lemma F.2, Lemma 4.3 is a natural corollary and the proof is deferred to Appendix F.5.

**Remark F.3** *Recall that in steps 2 and 3, the difference between $\mathcal{I}_{t+1}^1$ and $\mathcal{I}_{t+1}^0$ only resides in $h_{f_1}^{\mathrm{P}}(\boldsymbol{x},\gamma)$ (or $h_{r_1}^{\mathrm{P}}(\boldsymbol{x},\gamma)$ when we consider problem (13)) for $\boldsymbol{x} \in \mathcal{I}_t^0$, while Lemma F.2 implies that $h_{r_1}^{\mathrm{P}}(\boldsymbol{x},\gamma)$ would not expand the information set as long as $\boldsymbol{x} \neq \boldsymbol{0}$[18]. This demonstrates that applying (26) recursively would not affect the analysis of communication complexity.*

The next result is a corollary of Lemma 4.3 and the proof is deferred to Appendix F.6.

---

[18]In the proof of Lemma 4.3 in Appendix F.5, we show that $\mathcal{I}_{t+1}^1 = \mathcal{I}_{t+1}^0$

**Corollary F.4** *Define the random variables $T_0 = -1$,*

$$T_k := \min_t \{t : t > T_{k-1}, 3k-2 \in \mathcal{L}_{i_t} \text{ or } a_t = 1\} \text{ for } 1 \le k \le (m-1)/3, \qquad (29)$$

*and $Y_k := T_k - T_{k-1}$. Then we have (i) $\mathcal{I}_{t+1} \subseteq \mathcal{F}_{3k-2}$ for any $t < T_k$; (ii) the $Y_k$ are mutually independent; (iii) $Y_k \sim \text{Geom}(q_{k'} + p - pq_{k'})$ with $k' \equiv 3k-1 \pmod{(n-1)}$, $2 \le k' \le n$.*

Corollary F.4 claims that $T_k$ is the smallest index of the communication round after which the information set can be expanded to $\mathcal{F}_{3k+2}$. Moreover, $T_k$ can be decomposed into the sum of independent geometric random variables. With Lemma A.6, which gives a concentration result for the sum of geometric random variables, we have the following proposition, whose proof is deferred to Appendix F.7.

**Proposition F.5** *Let $0 \le M \le (m-2)/3$ and $N = \frac{n(M+1)}{4(q_0+c_0)} + 1$ with $q_0$ and $c_0$ defined in Definition 4.1. Suppose we use an algorithm $\mathcal{A}$ satisfying Definition F.1 to solve problem (13). After $N$ round of communication, the algorithm obtains the information set $\mathcal{I}_N$. Then we have $\mathcal{I}_N \subseteq \mathcal{F}_{3M+1} \subset \mathcal{F}_{m-1}$. Moreoever, if*

$$\min_{\boldsymbol{x} \in \mathcal{F}_{3M+1}} r(\boldsymbol{x}) - \min_{\boldsymbol{x} \in \mathbb{R}^m} r(\boldsymbol{x}) \ge 9\epsilon, \qquad (30)$$

*we have*

$$\mathbb{E} \min_{\boldsymbol{x} \in \mathcal{I}_N} r(\boldsymbol{x}) - \min_{\boldsymbol{x} \in \mathbb{R}^m} r(\boldsymbol{x}) \ge \epsilon.$$

Proposition F.5 specifies the number of communication rounds needed to find an $\epsilon$-suboptimal solution under the condition (30). Roughly speaking, the condition requires that the exact solution of problem (13) does not lie in some subspace of $\mathcal{F}_{m-1}$

Now we come back to the hard instance (13). Recall that $r$ is $c$-strongly convex and $r_i$'s satisfy $\sqrt{8n+4}$-aveSS. We need to properly scale the function class $\{r_i\}_{i=1}^n$ such that it satisfies Assumption 1. Note that rescaling does not influence Lemma F.2. Thus Proposition F.5 still holds for any rescaled version of problem (13). Specially, we consider the following problem

$$\min_{\boldsymbol{x} \in \mathbb{R}^m} f^{\mathrm{h}}(\boldsymbol{x}) := \frac{1}{n} \sum_{i=1}^n f_i^{\mathrm{h}}(\boldsymbol{x}) \quad \text{where} \quad f_i^{\mathrm{h}}(\boldsymbol{x}) := \lambda\, r(\boldsymbol{x}/\beta; m, \zeta, c),$$

$$\lambda = \frac{4\Delta}{\rho-1}, \ \beta = \frac{4}{\rho-1}\sqrt{\frac{\Delta}{\mu(\rho+1)}}, \ \zeta = \sqrt{\frac{2}{1+\rho}} \ \text{ and } \ c = \frac{4}{\rho^2-1} \ \text{ with } \ \rho = \sqrt{\frac{2\delta/\mu}{\sqrt{2n+1}}}+1. \qquad (31)$$

Here $n, \delta, \mu$ and $\Delta$ are given parameters. As shown in the next Proposition, $\delta$ is the AveSS parameter, $\mu$ is the strong convexity parameter and $\Delta$ is the function value gap between the initial point and the solution.

**Proposition F.6** *The problem defined in (31) with $n \ge 3$ and $m \ge 3$ has the following properties.*

1. *$f^{\mathrm{h}}$ is $\mu$-strongly convex and $f_i^{\mathrm{h}}$'s satisfy $\delta$-AveSS.*

2. *Let $q = \frac{\rho-1}{\rho+1}$. The minimizer of $f^{\mathrm{h}}$ is $\boldsymbol{x}_* = \frac{\beta(\rho+1)}{2}(q, q^2, \ldots, q^m)^\top$ and $f^{\mathrm{h}}(\boldsymbol{0}) - f^{\mathrm{h}}(\boldsymbol{x}_*) = \Delta$.*

3. *For $0 \le k \le m-1$, we have*

$$\min_{\boldsymbol{x} \in \mathcal{F}_k} f^{\mathrm{h}}(\boldsymbol{x}) - f^{\mathrm{h}}(\boldsymbol{x}_*) \ge \Delta q^{2k} \quad \text{and} \quad \min_{\boldsymbol{x} \in \mathcal{F}_k} \|\boldsymbol{x} - \boldsymbol{x}_*\|^2 \ge \frac{4\Delta}{\mu(\rho+1)} q^{2k}. \qquad (32)$$

Property 2 shows that the minimizer of problem (31) has all elements nonzero. Thus, it does not lie in any subspace $\mathcal{F}_k$ for $k < m$. As a result, we cannot obtain an approximate solution up to an arbitrarily small accuracy, unless we get an iterate with the last element nonzero, as claimed by Property 3. This implies problem (31) satisfies the condition 30.

Combining Propositions F.5 and F.6, we can establish the lower bound of the communication complexity.

**Theorem F.7 (Formal version of Theorem 4.4)** *Suppose we use any algorithm $\mathcal{A}$ satisfying Definition F.1 to solve the minimization problem* (31) *and the following conditions hold*

$$n \geq 3 \ \ and \ \ \epsilon \leq \frac{\Delta}{9} \cdot q^3, \ with \ q = \frac{\rho - 1}{\rho + 1}, \rho = \sqrt{\frac{2\delta/\mu}{\sqrt{2n+1}}} + 1.$$

*Set $m = \left\lfloor \frac{\log(\Delta/(9\epsilon))}{2\log(1/q)} + 2 \right\rfloor$. In order to find $\hat{x}$ such that $\mathbb{E} f^{\mathrm{h}}(\hat{x}) - \min_{x \in \mathbb{R}^m} f^{\mathrm{h}}(x) < \epsilon$, the communication complexity in expectation is*

$$\begin{cases} \Omega\left(n + n^{3/4}\sqrt{\delta/\mu}\log(1/\epsilon)\right), & for \ \frac{\delta}{\mu} = \Omega(\sqrt{n}), \\ \Omega\left(n + \frac{n\log(1/\epsilon)}{1+(\log(\mu\sqrt{n}/\delta))_+}\right), & for \ \frac{\delta}{\mu} = \mathcal{O}(\sqrt{n}). \end{cases}$$

Recall that in (32), $\min_{x \in \mathcal{F}_k} f^{\mathrm{h}}(x) - f^{\mathrm{h}}(x_*)$ and $\min_{x \in \mathcal{F}_k} \|x - x_*\|^2$ are both lower bounded by $q^{2k}$ multiplied with some constants. If we want to find $\hat{x}$ such that $\mathbb{E}\|\hat{x} - x_*\|^2 < \epsilon$, the communication complexity is the same. The proof of Theorem F.7 is deferred to Appendix F.9.

### F.3  Proof of Proposition 4.2

Proof: For convenience of notation, we omit the dependence of $r_i$, $r$, $B$ and $b_l$ on the parameters $m$, $\zeta$ and $c$. With the definition of $\{r_i\}_{i=1}^n$ and $r$, we have

$$\nabla(r_i - r)(x) = \begin{cases} -\sum_{l=1}^m b_l b_l^\top x - (n-1)e_1, & i = 1, \\ n\sum_{l \in \mathcal{L}_i} b_l b_l^\top x - \sum_{l=1}^m b_l b_l^\top x + e_1, & i \neq 1. \end{cases} \tag{33}$$

From the definition of $b_l$, we have

$$b_l^\top b_l = \begin{cases} 2, & 1 \leq l \leq m-1, \\ \zeta^2, & l = m, \end{cases} \qquad b_l^\top b_{l+1} = \begin{cases} -1, & 1 \leq l \leq m-2, \\ -\zeta, & l = m-1, \end{cases} \tag{34}$$

and $b_l^\top b_{l'} = 0$ for any $|l - l'| \geq 2$. Since $n \geq 3$, this implies $b_l^\top b_{l'} = 0$ for any $l, l' \in \mathcal{L}_i$ and $l \neq l'$. Define $b_0 = b_{m+1} = 0$ for ease of notation. Let $A_i = \sum_{l \in \mathcal{L}_i} b_l b_l^\top$, $A = \sum_{i=2}^n A_i = \sum_{l=1}^m b_l b_l^\top$. Then for any $x, y \in \mathbb{R}^m$ and $u = x - y$, we have

$$\sum_{i=1}^n \|\nabla(r_i - r)(x) - \nabla(r_i - r)(y)\|^2 \overset{(33)}{=} \|Au\|^2 + \sum_{i=2}^n \|nA_i u - Au\|^2$$

$$= \|Au\|^2 + \sum_{i=2}^n n^2\|A_i u\|^2 - 2n\sum_{i=2}^n (A_i u)^\top Au + (n-1)\|Au\|^2 = n^2\sum_{i=2}^n \|A_i u\|^2 - n\|Au\|^2.$$

Note that by Eq. (34), $\|A_i u\|^2 = \left\|\sum_{l \in \mathcal{L}_i} b_l b_l^\top u\right\|^2 = \sum_{l \in \mathcal{L}_i} (b_l^\top u)^2 b_l^\top b_l$ and

$$\|Au\|^2 = \left\|\sum_{l=1}^m b_l b_l^\top u\right\|^2 = \sum_{l=1}^m (b_l^\top u)^2 b_l^\top b_l + (b_l^\top u)(b_{l+1}^\top u)b_{l+1}^\top b_l + (b_l^\top u)(b_{l-1}^\top u)b_{l-1}^\top b_l$$

$$= \sum_{l=1}^m (b_l^\top u)^2 b_l^\top b_l + 2(b_l^\top u)(b_{l+1}^\top u)b_{l+1}^\top b_l,$$

where the final equality uses $b_0 = b_{m+1} = 0$. Hence, we get

$$\frac{1}{n}\sum_{i=1}^n \|\nabla(r_i - r)(x) - \nabla(r_i - r)(y)\|^2$$

$$= n\sum_{l=1}^m (b_l^\top u)^2 b_l^\top b_l - \left[\sum_{l=1}^m (b_l^\top u)^2 b_l^\top b_l + 2(b_l^\top u)(b_{l+1}^\top u)b_{l+1}^\top b_l\right]$$

$$= (n-1)\sum_{l=1}^m (b_l^\top u)^2 b_l^\top b_l - 2\sum_{l=1}^m (b_l^\top u)(b_{l+1}^\top u)b_{l+1}^\top b_l. \tag{35}$$

Recall that $0 < \zeta \le \sqrt{2}$. Then (34) implies $\boldsymbol{b}_l^\top \boldsymbol{b}_l \le 2$ and $|\boldsymbol{b}_l^\top \boldsymbol{b}_{l+1}| \le \sqrt{2}$ for any $l$. Substituting these into (35) and using Cauchy's inequality, we have

$$\frac{1}{n}\sum_{i=1}^{n}\|\nabla(r_i-r)(\boldsymbol{x})-\nabla(r_i-r)(\boldsymbol{y})\|^2 \quad \le \quad 2(n-1)\sum_{l=1}^{m}(\boldsymbol{b}_l^\top \boldsymbol{u})^2 + \sqrt{2}\sum_{l=1}^{m}\left[(\boldsymbol{b}_l^\top \boldsymbol{u})^2 + (\boldsymbol{b}_{l+1}^\top \boldsymbol{u})^2\right]$$

$$\le \quad \left(2n+2\sqrt{2}-2\right)\sum_{l=1}^{m}(\boldsymbol{b}_l^\top \boldsymbol{u})^2 \le (2n+1)\sum_{l=1}^{m}(\boldsymbol{b}_l^\top \boldsymbol{u})^2.$$

Notice that $(\boldsymbol{b}_l^\top \boldsymbol{u})^2 = (u_l - u_{l+1})^2 \le 2(u_l^2 + u_{l+1}^2)$ for $1 \le l \le m-1$ and $(\boldsymbol{b}_m^\top \boldsymbol{u})^2 = (\zeta u_m)^2 \le 2u_m^2$. This implies

$$\frac{1}{n}\sum_{i=1}^{n}\|\nabla(r_i-r)(\boldsymbol{x})-\nabla(r_i-r)(\boldsymbol{y})\|^2 \le 4(2n+1)\|\boldsymbol{u}\|^2 = (8n+4)\|\boldsymbol{x}-\boldsymbol{y}\|^2.$$

As a result, $r_i$'s satisfy $\sqrt{8n+4}$-AveSS. $\qquad\square$

### F.4 Proof of Lemma F.2

Proof: For convenience of notation, we omit the dependence of $r_i$, $r$, $\boldsymbol{B}$ and $\boldsymbol{b}_l$ on the parameters $m$, $\zeta$ and $c$.

1) First, we focus on the gradient of the $r_i$. Recall that

$$r_i(\boldsymbol{x})=\begin{cases} \frac{c}{2}\|\boldsymbol{x}\|^2 - n\langle \boldsymbol{e}_1, \boldsymbol{x}\rangle, & \text{for } i = 1, \\ \frac{n}{2}\sum_{l\in\mathcal{L}_i}\boldsymbol{x}^\top \boldsymbol{b}_l \boldsymbol{b}_l^\top \boldsymbol{x} + \frac{c}{2}\|\boldsymbol{x}\|^2, & \text{for } i \ne 1. \end{cases} \Rightarrow \nabla r_i(\boldsymbol{x})=\begin{cases} c\boldsymbol{x} - n\boldsymbol{e}_1, & \text{for } i = 1, \\ n\sum_{l\in\mathcal{L}_i}\boldsymbol{b}_l \boldsymbol{b}_l^\top \boldsymbol{x} + c\boldsymbol{x}, & \text{for } i \ne 1. \end{cases}$$

(i): If $\boldsymbol{x} \in \mathcal{F}_0$, i.e., $\boldsymbol{x} = \boldsymbol{0}$, we have $\nabla r_1(\boldsymbol{0}) = -n\boldsymbol{e}_1 \in \mathcal{F}_1$ and $\nabla r_i(\boldsymbol{0}) = \boldsymbol{0}$ for $i \ne 1$. (ii): If $\boldsymbol{x} = (x_1, \ldots, x_m) \in \mathcal{F}_k$ for $1 \le k < m$, we have $\nabla r_1(\boldsymbol{x}) = c\boldsymbol{x} - n\boldsymbol{e}_1 \in \mathcal{F}_k$. As for $i \ne 1$, we need to examine $\boldsymbol{b}_l \boldsymbol{b}_l^\top \boldsymbol{x}$. One can check

$$\boldsymbol{b}_l \boldsymbol{b}_l^\top \boldsymbol{x} = \begin{cases} (x_l - x_{l+1})(\boldsymbol{e}_l - \boldsymbol{e}_{l+1}), & 1 \le l \le m-1, \\ \zeta^2 x_m \boldsymbol{e}_m, & l = m. \end{cases}$$

For $\boldsymbol{x} \in \mathcal{F}_k$, we have

$$\boldsymbol{b}_l \boldsymbol{b}_l^\top \boldsymbol{x} \in \begin{cases} \mathcal{F}_k, & l \ne k, \\ \mathcal{F}_{k+1}, & l = k. \end{cases} \tag{36}$$

As a result, if $k \in \mathcal{L}_i$, then $\nabla r_i(\boldsymbol{x}) \in \mathcal{F}_{k+1}$; otherwise, $\nabla r_i(\boldsymbol{x}) \in \mathcal{F}_k$.

2) Now we turn to the proximal operator. (i) For $i = 1$, it is easy to verify $\text{prox}_{r_1}^\gamma(\boldsymbol{x}) = (1/\gamma + c)^{-1}(\boldsymbol{x}/\gamma + n\boldsymbol{e}_1)$. Thus, if $\boldsymbol{x} \in \mathcal{F}_0$, $\text{prox}_{r_1}^\gamma(\boldsymbol{x}) \in \mathcal{F}_1$; if $\boldsymbol{x} \in \mathcal{F}_k$ for $k \ge 1$, $\text{prox}_{r_1}^\gamma(\boldsymbol{x}) \in \mathcal{F}_k$. (ii) For $i \ne 1$, we define $\boldsymbol{u}_i := \text{prox}_{r_i}^\gamma(\boldsymbol{x})$ for simplicity. Then $\boldsymbol{u}_i$ satisfies the following equation

$$\left[n\gamma \boldsymbol{B}_i^\top \boldsymbol{B}_i + (c\gamma+1)\boldsymbol{I}\right]\boldsymbol{u}_i = \boldsymbol{x}, \quad \boldsymbol{B}_i := \sum_{l\in\mathcal{L}_i}\boldsymbol{e}_l \boldsymbol{b}_l^\top.$$

Note that $\boldsymbol{B}_i^\top \boldsymbol{B}_i = \sum_{l\in\mathcal{L}_i}\boldsymbol{b}_l \boldsymbol{b}_l^\top$. By the Sherman-Morrison-Woodbury formula, we get

$$\left(\boldsymbol{I} + \tilde{c}\boldsymbol{B}_i^\top \boldsymbol{B}_i\right)^{-1} = \boldsymbol{I} - \boldsymbol{B}_i^\top \left(\frac{1}{\tilde{c}}\boldsymbol{I} + \boldsymbol{B}_i \boldsymbol{B}_i^\top\right)^{-1}\boldsymbol{B}_i, \forall \tilde{c} \ne 0.$$

In the proof of Proposition 4.2, we have shown that $\boldsymbol{b}_l^\top \boldsymbol{b}_{l'} = 0$ for any $|l - l'| \ge 2$ and consequently $\boldsymbol{b}_l^\top \boldsymbol{b}_{l'} = 0$ for any $l, l' \in \mathcal{L}_i$ and $l \ne l'$. Thus, $\boldsymbol{B}_i \boldsymbol{B}_i^\top = \sum_{l\in\mathcal{L}_i}\boldsymbol{b}_l^\top \boldsymbol{b}_l \boldsymbol{e}_l \boldsymbol{e}_l^\top$ is a diagonal matrix. Then we can denote $\boldsymbol{D}_i = \left(\frac{c\gamma+1}{n\gamma}\boldsymbol{I} + \boldsymbol{B}_i \boldsymbol{B}_i^\top\right)^{-1} = \sum_{l=1}^{m}d_{i,l}\boldsymbol{e}_l \boldsymbol{e}_l^\top$ and obtain

$$\boldsymbol{u}_i = \left[n\gamma \boldsymbol{B}_i^\top \boldsymbol{B}_i + (c\gamma+1)\boldsymbol{I}\right]^{-1}\boldsymbol{x} = \frac{1}{c\gamma+1}\left(\boldsymbol{I} + \frac{n\gamma}{c\gamma+1}\boldsymbol{B}_i^\top \boldsymbol{B}_i\right)^{-1}\boldsymbol{x} = \frac{\boldsymbol{x} - \boldsymbol{B}_i^\top \boldsymbol{D}_i \boldsymbol{B}_i \boldsymbol{x}}{c\gamma+1}.$$

Then we have $\boldsymbol{B}_i^\top \boldsymbol{D}_i \boldsymbol{B}_i \boldsymbol{x} = \sum_{l\in\mathcal{L}_i}d_{i,l}\boldsymbol{b}_l \boldsymbol{b}_l^\top \boldsymbol{x}$. For $\boldsymbol{x} \in \mathcal{F}_k$, by (36), if $k \in \mathcal{L}_i$, then $\boldsymbol{u}_i \in \mathcal{F}_{k+1}$; otherwise, $\boldsymbol{u}_i \in \mathcal{F}_k$. This completes the proof. $\qquad\square$

## F.5 Proof of Lemma 4.3

Proof: Since we can assume $x_0 = 0$, Lemma F.2 implies $\nabla r_1 \in \mathcal{F}_1$. Then from step 1, we have $\mathcal{I}_0 = \mathcal{F}_1$[19].

Then we focus on the second claim and examine how many dimensions of the information set we can increase after a round of communication.

Since we choose node 1 as the master node, we have access to $\nabla r_1$ and $\mathrm{prox}_{r_1}^\gamma$ in each communication round. Recall that we set $\mathcal{L}_1$ as the empty set. Lemma F.2 guarantees that the information provided by $r_1$ can never expand the information set unless the information set only contains $0$. Thus, (26) does not affect the information set, i.e., $\mathcal{I}_{t+1}^1 = \mathcal{I}_{t+1}^0$ for any $t \geq 0$.

When $a_t = 0$, only (25) can expand the information set. By Lemma F.2, if $i_t$ satisfies $k \in \mathcal{L}_{i_t}$, we have $\mathcal{I}_{t+1}^1 = \mathcal{I}_{t+1}^0 \subseteq \mathcal{F}_{k+1}$. Otherwise, we still have $\mathcal{I}_{t+1}^1 = \mathcal{I}_{t+1}^0 \subseteq \mathcal{F}_k$. Then step 4 in Definition F.1 implies $\mathcal{I}_{t+1} = \mathcal{I}_{t+1}^1$.

When $a_t = 1$, from the above analysis, we always have $\mathcal{I}_{t+1} \subseteq \mathcal{F}_{k+1}$. By Lemma F.2, (27) could expand the information set by at most one dimension. It follows that $\tilde{y}_{t+1} \in \mathcal{F}_{k+2}$. Using Lemma F.2 again yields $\mathcal{I}_{t+1} \subseteq \mathcal{F}_{k+3}$. $\qquad\square$

## F.6 Proofs of Corollary F.4

Proof: We prove the first claim by induction on $t$. That is to say, we prove that for any integer $t \geq -1$, $\mathcal{I}_{t+1} \subseteq \mathcal{F}_{3k-2}$ for any $k$ satisfying $t < T_k$. Define $k(t)$ as the positive integer such that $T_{k(t)-1} \leq t < T_{k(t)}$. From the monotonicity of $\mathcal{F}_\cdot$, it suffices to prove $\mathcal{I}_{t+1} \subseteq \mathcal{F}_{3k(t)-2}$.

By Lemma F.2, we have $\mathcal{I}_0 = \mathcal{F}_1$ and $k(-1) = 1$. The claim holds for $t = -1$. Suppose that $\mathcal{I}_{t+1} \subseteq \mathcal{F}_{3k(t)-2}$. If $3k(t) - 2 \in \mathcal{L}_{i_{t+1}}$ or $a_{t+1} = 1$, Lemma 4.3 together with (29) implies $k(t+1) = k(t) + 1$ and $\mathcal{I}_{t+2} \subseteq \mathcal{F}_{3k(t)+1} = \mathcal{F}_{3k(t+1)-1}$. Otherwise, we still have $k(t+1) = k(t)$ and $\mathcal{I}_{t+2} \subseteq \mathcal{F}_{3k(t)-1} = \mathcal{F}_{3k(t+1)-1}$.

For the second claim, the independence of $\{Y_k\}_{k \geq 1}$ is natural consequence of the independence of $\{(i_t, a_t)\}_{t \geq 1}$.

For the last one, note that $3k - 2 \in \mathcal{L}_{i_t}$ is equivalent to $i_t \equiv 3k-1 (\mathrm{mod}\ (n-1))$ for $2 \leq i_t \leq n$. Then we have for $k' \equiv 3k-1 (\mathrm{mod}\ (n-1)), 2 \leq k' \leq n$,

$$
\begin{aligned}
\mathbb{P}(T_k - T_{k-1} = s) \;=\; & \mathbb{P}(i_{T_{k-1}+1} \neq k', \ldots, i_{T_{k-1}+s-1} \neq k', a_{T_{k-1}+1} = \cdots = a_{T_{k-1}+s-1} = 0, \\
& i_{T_{k-1}+s} = k' \text{ or } a_{T_{k-1}+s} = 1) \\
\overset{(i)}{=} \;& \left[(1 - q_{k'})(1 - p)\right]^{s-1} \left[1 - (1 - q_{k'})(1 - p)\right],
\end{aligned}
$$

where $(i)$ is due to the independence of $\{(i_t, a_t)\}_{t \geq 1}$. So $Y_k = T_k - T_{k-1}$ is a geometric random variable with success probability $1 - (1 - q_{k'})(1 - p) = q_{k'} + p - q_{k'}p$. $\qquad\square$

## F.7 Proof of Proposition F.5

Proof: By Corollary F.4, if $N - 1 < T_{M+1}$, then $\mathcal{I}_N \subseteq \mathcal{F}_{3M+1} \subseteq \mathcal{F}_{m-1}$. Thus we have

$$
\mathbb{E} \min_{x \in \mathcal{I}_N} r(x) - \min_{x \in \mathbb{R}^m} r(x) \geq \mathbb{E}\left[\min_{x \in \mathcal{I}_N} r(x) - \min_{x \in \mathbb{R}^m} r(x) \Big| N - 1 < T_{M+1}\right] \mathbb{P}(N - 1 < T_{M+1})
$$

$$
\geq \mathbb{E}\left[\min_{x \in \mathcal{F}_{3M+1}} r(x) - \min_{x \in \mathbb{R}^m} r(x) \Big| N - 1 < T_{M+1}\right] \mathbb{P}(N - 1 < T_{M+1}) \geq 9\epsilon\, \mathbb{P}(N - 1 < T_{M+1}).
$$

By Corollary F.4 again, $T_{M+1}$ can be written as $T_{M+1} = \sum_{l=1}^{M+1} Y_l$, where $\{Y_l\}_{1 \leq l \leq M+1}$ are independent random variables, and $Y_l \sim \mathrm{Geom}(\tilde{q}_l)$ with $\tilde{q}_l = q_{l'} + p - pq_{l'}$, $l' \equiv 3l-1 (\mathrm{mod}\ (n-1))$ and $2 \leq l' \leq n$. Moreover, Definition F.1 guarantees $\max_{2 \leq l' \leq n} q_l' \leq q_0/n$ and $p \leq c_0/n$. Then we have $\sum_{l=1}^{M+1} \tilde{q}_l \leq (q_0 + c_0)(M+1)/n$. Therefore, by Lemma A.6, we have

$$
\mathbb{P}(T_{M+1} > N - 1) = \mathbb{P}\left(\sum_{l=1}^{M+1} Y_l > \frac{n(M+1)}{4(q_0 + c_0)}\right) \geq \mathbb{P}\left(\sum_{l=1}^{M+1} Y_l > \frac{(M+1)^2}{4\sum_{l=1}^{M+1} \tilde{q}_l}\right) \geq \frac{1}{9},
$$

---

[19]In the definition of the information set, each $f_i$ is replaced by $r_i$ here.

which implies our desired result. $\qquad\square$

## F.8 Proof of Proposition F.6

Proof: **Property 1.** By Proposition 4.2, we have $f^{\mathrm{h}}$ is $\lambda c/\beta^2$-strongly convex and $f_i^{\mathrm{h}}$'s satisfy $\lambda\sqrt{8n+4}/\beta^2$-AveSS. One can check $\lambda c/\beta^2 = \mu$ and $\lambda\sqrt{8n+4}/\beta^2 = \delta$.

**Property 2.** Let $\xi := \lambda/\beta^2 = \mu(\rho^2-1)/4$. We have

$$f^{\mathrm{h}}(\boldsymbol{x}) = \frac{\xi}{2}\boldsymbol{x}^\top \boldsymbol{A}(m,\zeta)\,\boldsymbol{x} + \frac{\mu}{2}\,\|\boldsymbol{x}\|^2 - \xi\beta\,\langle\boldsymbol{e}_1,\boldsymbol{x}\rangle.$$

Letting $\nabla f^{\mathrm{h}}(\boldsymbol{x}) = \boldsymbol{0}$ yields $(\xi\boldsymbol{A}(m,\zeta) + \mu\boldsymbol{I})\,\boldsymbol{x} = \xi\beta\boldsymbol{e}_1$, or equivalently.

$$\begin{bmatrix} 1+\frac{\mu}{\xi} & -1 & & & & \\ -1 & 2+\frac{\mu}{\xi} & -1 & & & \\ & \ddots & \ddots & \ddots & & \\ & & -1 & 2+\frac{\mu}{\xi} & -1 \\ & & & -1 & \zeta^2+1+\frac{\mu}{\xi} \end{bmatrix}\boldsymbol{x} = \begin{bmatrix} \beta \\ 0 \\ \vdots \\ 0 \\ 0 \end{bmatrix}. \tag{37}$$

Since $q = \frac{\rho-1}{\rho+1}$, we get $2+\frac{\mu}{\xi} = \frac{2\rho^2+2}{\rho^2-1} = q+\frac{1}{q}$ and $\zeta^2+1+\frac{\mu}{\xi} = \frac{\rho+1}{\rho-1} = \frac{1}{q}$. We solve (37) by

$$x_{m-1} - \frac{x_m}{q} = 0,\ x_k - \left(q+\frac{1}{q}\right)x_{k+1} + x_{k+2} = 0,\ k\in[m-2],\ \left(q+\frac{1}{q}-1\right)x_1 - x_2 = \beta.$$

Thus, $\boldsymbol{x}_* = \frac{\beta}{1-q}(q,q^2,\dots,q^m)^\top$ and $f^{\mathrm{h}}(\boldsymbol{x}_*) = -\frac{\xi\beta\langle\boldsymbol{e}_1,\boldsymbol{x}_*\rangle}{2} = -\frac{\xi\beta^2 q}{2(1-q)} = -\frac{\lambda(\rho-1)}{4} \overset{(31)}{=} -\Delta$.

**Property 3.** If $\boldsymbol{x}\in\mathcal{F}_k$, $1\le k < m$, then $x_{k+1} = x_{k+2} = \cdots = x_m = 0$. Let $\boldsymbol{y}$ denote the first $k$ coordinates of $\boldsymbol{x}$ and $\boldsymbol{A}_k$ denote the first $k$ rows and columns of $\boldsymbol{A}(m,\zeta)$. Then for any $\boldsymbol{x}\in\mathcal{F}_k$, we can rewrite $f^{\mathrm{h}}(\boldsymbol{x})$ as

$$\hat{f}^{\mathrm{h}}(\boldsymbol{y}) := f^{\mathrm{h}}(\boldsymbol{x}) = \frac{\xi}{2}\boldsymbol{y}^\top \boldsymbol{A}_k\boldsymbol{y} + \frac{\mu}{2}\,\|\boldsymbol{x}\|^2 - \xi\beta\,\langle\hat{\boldsymbol{e}}_1,\boldsymbol{y}\rangle,$$

where $\hat{\boldsymbol{e}}_1$ is the first $k$ coordinates of $\boldsymbol{e}_1$. Let $\nabla f_k(\boldsymbol{y}) = \boldsymbol{0}$, that is

$$\begin{bmatrix} 1+\frac{\mu}{\xi} & -1 & & & & \\ -1 & 2+\frac{\mu}{\xi} & -1 & & & \\ & \ddots & \ddots & \ddots & & \\ & & -1 & 2+\frac{\mu}{\xi} & -1 \\ & & & -1 & 2+\frac{\mu}{\xi} \end{bmatrix}\boldsymbol{y} = \begin{bmatrix} \beta \\ 0 \\ \vdots \\ 0 \\ 0 \end{bmatrix}.$$

Similarly, we need to solve

$$x_{m-1} = \left(q+\frac{1}{q}\right)x_m,\ x_k - \left(q+\frac{1}{q}\right)x_{k+1} + x_{k+2} = 0,\ k\in[m-2],\ \left(q+\frac{1}{q}-1\right)x_1 - x_2 = \beta.$$

By some computation, one can check the solution is

$$\boldsymbol{y}_* = \frac{\beta q^{k+1}}{2(q-1)(1+q^{2k+1})}\left(q^{-k}-q^k, q^{-(k-1)}-q^{k-1},\dots,q^{-1}-q^1\right)^\top.$$

Thus, we have

$$f^{\mathrm{h}}(\boldsymbol{y}_*) = -\frac{\xi\beta\,\langle\hat{\boldsymbol{e}}_1,\boldsymbol{y}_*\rangle}{2} = \frac{\xi\beta^2 q}{2(1-q)}\cdot\frac{1-q^{2k}}{1+q^{2k+1}} = \frac{\lambda(\rho-1)}{4}\cdot\frac{1-q^{2k}}{1+q^{2k+1}} = \frac{(1-q^{2k})\Delta}{1+q^{2k+1}},$$

and by $q < 1$, we further have that

$$\min_{\boldsymbol{x}\in\mathcal{F}_k} f^{\mathrm{h}}(\boldsymbol{x}) - \min_{\boldsymbol{x}\in\mathbb{R}^m} f^{\mathrm{h}}(\boldsymbol{x}) = f^{\mathrm{h}}(\boldsymbol{y}_*) - f^{\mathrm{h}}(\boldsymbol{x}_*) = \Delta\left(1-\frac{1-q^{2k}}{1+q^{2k+1}}\right) = \frac{(1+q)q^{2k}\Delta}{1+q^{2k+1}} \ge \Delta q^{2k}.$$

Moreover, recall that $\boldsymbol{x}_* = \frac{\beta(\rho+1)}{2}(q,q^2,\dots,q^m)^\top$. Then we have

$$\min_{\boldsymbol{x}\in\mathcal{F}_k} \|\boldsymbol{x}-\boldsymbol{x}^\star\|^2 = \frac{\beta^2(\rho+1)^2}{4}\sum_{i=k+1}^m q^{2i} \ge \frac{\beta^2(\rho+1)^2 q^2}{4}q^{2k} = \frac{4\Delta}{\mu(\rho+1)}q^{2k}.$$

This completes the proof. $\qquad\square$

## F.9 Proof of Theorem F.7

Proof: Let $q = \frac{\rho-1}{\rho+1}$ and $M = \left\lfloor \frac{\log(\Delta/9\epsilon)}{6\log 1/q} - \frac{1}{3} \right\rfloor$. From the condition on $\epsilon$ and the definition of $m$, one can check $0 \le M \le (m-2)/3$ and $m \ge 3$. Moreover, we have $3M+1 \le \frac{\log(9\epsilon/\Delta)}{2\log q}$. Then by Proposition F.5, after $N = \frac{n(M+1)}{4(q_0+c_0)} + 1$ rounds of communication, the information set satisfies $\mathcal{I}_N \subseteq \mathcal{F}_{3M+1} \subseteq \mathcal{F}_{m-1}$. The third property of Propostion F.6 implies

$$\min_{\boldsymbol{x} \in \mathcal{F}_{3M+1}} f^{\mathrm{h}}(\boldsymbol{x}) - \min_{\boldsymbol{x} \in \mathbb{R}^m} f(\boldsymbol{x}) \ge \Delta q^{6M+2} \ge 9\epsilon.$$

Then by Proposition F.5 again, in order to find $\hat{\boldsymbol{x}}$ such that $\mathbb{E}f^{\mathrm{h}}(\hat{\boldsymbol{x}}) - \min_{\boldsymbol{x} \in \mathbb{R}^m} f^{\mathrm{h}}(\boldsymbol{x}) < \epsilon$, the algorithm $\mathcal{A}$ needs at least $N$ communication rounds.

Now we give a lower bound $N$. According to whether $2\delta/\mu$ is larger than $\sqrt{2n+1}$, we divide the analysis into two cases.

**Case 1:** $2\delta/\mu \ge \sqrt{2n+1}$. Then $\rho \ge \sqrt{2}$. By inequality $0 < \log(1+x) \le x, \forall x > 0$, we get

$$\frac{1}{\log \frac{1}{q}} = \frac{1}{\log\left(1 + \frac{2}{\rho-1}\right)} \ge \frac{1}{\frac{2}{\rho-1}} = \frac{\rho-1}{2} = \frac{1}{2}\left(\sqrt{\frac{2\delta/\mu}{\sqrt{2n+1}} + 1} - 1\right) \ge \frac{\sqrt{2\delta/\mu}}{6\sqrt[4]{2n+1}},$$

where the final inequality uses $\sqrt{t+1} - 1 \ge \sqrt{t}/3, \forall t \ge 1$. Moreover, the condition on $\epsilon$ implies $\log \frac{\Delta}{9\epsilon} \ge 3\log \frac{1}{q}$. Then we have

$$M+1 \ge \frac{\log \frac{\Delta}{9\epsilon}}{6\log \frac{1}{q}} - \frac{1}{3} \ge \frac{\log \frac{\Delta}{9\epsilon}}{18\log \frac{1}{q}} \ge \frac{\sqrt{2\delta/\mu}}{108\sqrt[4]{2n+1}} \log \frac{\Delta}{9\epsilon} = \Omega\left(\frac{\sqrt{\delta/\mu}}{n^{1/4}} \log \frac{1}{\epsilon}\right).$$

It follows that $N = \frac{n(M+1)}{4(q_0+c_0)} + 1 = \Omega\left(n^{3/4}\sqrt{\delta/\mu} \log(1/\epsilon)\right)$. The total communication cost in expectation is of the order $\Theta(n+N) = \Omega\left(n + n^{3/4}\sqrt{\delta/\mu} \log(1/\epsilon)\right)$.

**Case 2:** $2\delta/\mu < \sqrt{2n+1}$. In this case, we have $1 \le \rho < \sqrt{2}$ and consequently

$$\log \frac{1}{q} = \log\left(1 + \frac{2}{\rho-1}\right) \overset{(i)}{\le} \log\left(1 + \frac{3\sqrt{2n+1}}{\delta/\mu}\right) \le \log\left(\frac{7\mu\sqrt{2n+1}}{2\delta}\right) \le 2 + \log\left(\frac{\mu\sqrt{2n+1}}{2\delta}\right),$$

where $(i)$ uses $\sqrt{t+1} - 1 \ge t/3, \forall 0 < t \le 1$ and $\rho = \sqrt{\frac{2\delta/\mu}{\sqrt{2n+1}} + 1}$, Moreover, the condition on $\epsilon$ implies $\log \frac{\Delta}{9\epsilon} \ge 3\log \frac{1}{q}$. Then we have

$$M+1 \ge \frac{\log \frac{\Delta}{9\epsilon}}{6\log \frac{1}{q}} - \frac{1}{3} \ge \frac{\log \frac{\Delta}{9\epsilon}}{18\log \frac{1}{q}} = \Omega\left(\frac{\log(1/\epsilon)}{1 + (\log(\mu\sqrt{n}/\delta))_+}\right).$$

where $(a)_+$ denote $\max\{a, 0\}$. It follows that $N = \frac{n(M+1)}{4(q_0+c_0)} + 1 = \Omega\left(\frac{n\log(1/\epsilon)}{1+(\log(\mu\sqrt{n}/\delta))_+}\right)$. The total communication cost in expectation is of the order $\Theta(n+N) = \Omega\left(n + \frac{n\log(1/\epsilon)}{1+(\log(\mu\sqrt{n}/\delta))_+}\right)$. $\quad\square$

# G Experiment Details

We show some detail of our numerical experiments in this section. The computation of problem-dependent parameters is defined as follows. Since the objective is

$$f(\boldsymbol{x}) = \frac{1}{n} \sum_{i=1}^{n} \left[ f_i(\boldsymbol{x}) := \frac{1}{m} \sum_{j=1}^{m} \left(\boldsymbol{z}_{i,j}^{\top} \boldsymbol{x} - y_{i,j}\right)^2 + \frac{\mu}{2} \|\boldsymbol{x}\|^2 \right],$$

Let $\boldsymbol{Z}_i = (\boldsymbol{z}_{i,1}, \cdots, \boldsymbol{z}_{i,m}) / \sqrt{m/2}$, $\boldsymbol{y}_i = (y_{i,1}, \ldots, y_{i,m})^{\top} / \sqrt{m/2}$. We reformulate $f_i, i \in [n]$ into

$$f_i(\boldsymbol{x}) = \frac{1}{2} \left\| \boldsymbol{Z}_i^{\top} \boldsymbol{x} - \boldsymbol{y}_i \right\|^2 + \frac{\mu}{2} \|\boldsymbol{x}\|^2, \quad \nabla^2 f_i(\boldsymbol{x}) = \boldsymbol{Z}_i \boldsymbol{Z}_i^{\top} + \mu \boldsymbol{I}_d.$$

Table 2: The choices of interpolation $\tau = s\tau_0$ in experiments.

|  |  | Katyusha X | | | AccSVRS | | |
| --- | --- | --- | --- | --- | --- | --- | --- |
| Synthetic data | $\mu$ | 1 | 0.1 | 0.01 | 1 | 0.1 | 0.01 |
| | s | 1 | 2 | 5 | 2 | 5 | 10 |
| Real data | $\mu$ | 0.1 | 0.01 | 0.001 | 0.1 | 0.01 | 0.001 |
| | s | 1 | 1 | 2 | 2 | 0.5 | 0.5 |

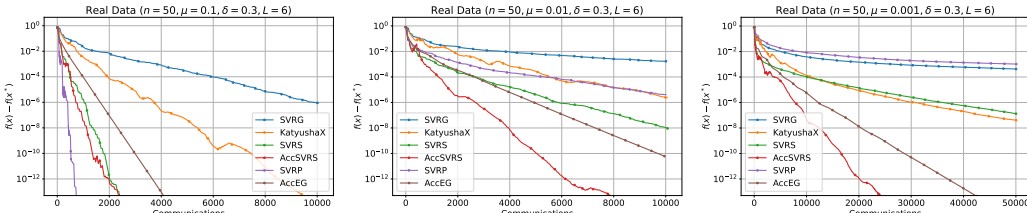

Figure 2: Numerical experiments on real data. The corresponding coefficients are shown in the title of each graph. We plot the function gap on a log scale versus the number of communication steps, where one exchange of vectors counts as a communication step.

Thus, we obtain the smoothness of each $f_i$ is $L_i = \|\boldsymbol{Z}_i\|^2 + \mu$, and $L := \max_{i \in [n]} L_i$. Obviously, $f$ is $\mu$-strongly convex.

For the synthetic data, we first generate a random symmetric matrix $\boldsymbol{Z}_0 \in \mathbb{R}^{d \times d}$ with $d = 100$ and $\|\boldsymbol{Z}_0\| = 3000$, then we add a perturbed symmetric matrix $\boldsymbol{N}_i, \forall i \in [n]$ with $n = 400$, $\|\boldsymbol{N}_i\| \approx 30$ to obtain $\boldsymbol{Z}_i = \boldsymbol{Z}_0 + \boldsymbol{N}_i$. We also add a correction $\lambda_{\min}(\boldsymbol{Z}_i)\boldsymbol{I}_d$ to $\boldsymbol{Z}_i$ to further make $\boldsymbol{Z}_i \succeq 0$. Finally, we recompute the center matrix $\boldsymbol{Z} = \sum_{i=1}^{n} \boldsymbol{Z}_i / n$ and $\delta$-average similarity coefficient following AveHS in Eq. (6) as

$$\delta = \sqrt{\frac{1}{n} \sum_{i=1}^{n} \|\boldsymbol{Z}_i - \boldsymbol{Z}\|^2}.$$

We use the analytic solution obtained by the proximal step since

$$\mathrm{prox}_{f_1}^{\theta}(\boldsymbol{x}_0) := \arg\min_{\boldsymbol{x} \in \mathbb{R}^d} f_1(\boldsymbol{x}) + \frac{1}{2\theta} \|\boldsymbol{x} - \boldsymbol{x}_0\|^2 = \left[\boldsymbol{Z}_1 \boldsymbol{Z}_1^\top + \left(\mu + \frac{1}{\theta}\right) \boldsymbol{I}_d\right]^{-1} \left(\boldsymbol{Z}_1 \boldsymbol{y} + \frac{\boldsymbol{x}_0}{\theta}\right).$$

For Katyusha X [6, Fact 4.2], and AccSVRS (Thm 3.6), we scale the interpolation coefficient $\tau = s\tau_0$ with $s \in \{0.5, 1, 2, 5, 10\}$ and $\tau_0$ is the theoretical value. The finally used scaling $s$ is shown in Table 2. The initial points of all methods are the same, which are sampled from $\mathrm{Unif}(\mathcal{S}^{d-1})$.

We also run the real data 'a9a' from LIBSVM library [16], where we split it into $n = 50$ datasets with the data size $m = 600$. The results are shown in Figure 2, and we can observe similar behavior of our methods.