# OpenReview forum: "Stochastic Distributed Optimization under Average Second-order Similarity: Algorithms and Analysis"
_NeurIPS.cc/2023/Conference — NeurIPS 2023 poster_

### Official Review · Reviewer_Y65R · 2023-07-01

**Soundness:** 2 fair
**Presentation:** 2 fair
**Contribution:** 2 fair
**Rating:** 4
**Confidence:** 3

**Summary:**

**Summary**

The paper studies finite-sum minimization ($\min_{x} f(x):= \frac{1}{n} \sum_{i= 1}^n f_i(x)$) in the distributed setting with a central node and $n-1$ non-central (client) nodes. The setup and the paper's assumptions are as follows.

1. Each function $f_i$ is held in the $i$-th client node, and the first node is designated the central node. The machines on the non-central nodes can communicate function values and gradients with each other as well as with the central node.

2. The different $f_i$'s are assumed to be related to each other via the notion of "second-order similarity", which essentially means that the Hessians of $f_i$ and $f$ evaluated at the same point do not differ from each other by too much in the operator norm. A benefit of this assumption is that different clients do not need to all send their Hessian information.

3. The $f_i$'s are all convex, $f$ is $\mu$-strongly convex, and the $f_i$'s all satisfy $\delta$-average-second-order-similarity as described in 2.

The paper's results are three-fold:

1. A non-accelerated algorithm called SVRS, which attains a communication cost of $\widetilde{O}(n + \sqrt{n} \delta/\mu)$. This result improves upon the previous best result of Khaled and Jin when $\delta \geq \sqrt{n} \mu$.

2. An accelerated algorithm called AccSVRS, which attains a communication cost of $O((n+ n^{3/4} \sqrt{\delta/\mu})\log|\epsilon|)$, which improves upon the previous best accelerated rate by Khaled and Jin by a factor of $\log(L/\mu)$. Therefore, this new rate is "smoothness-free".

3. Lower bounds for 2.

The paper's technique is based, broadly, on the use of Bregman-SVRG: instead of calculating exactly in each iteration, we calculate the approximate entities every time and periodically offset the error by exact calculations; additionally, through the use of a Bregman divergence term, the update rule ensures that the next iterate is not too far from the current one in some chosen distance metric.



**Strengths:**


**Strengths**
I think the paper scores highly on the story-telling aspect: it reads quite well! I also think the paper studies an important problem (faster communication complexity for distributed optimization).


**Weaknesses:**



In results:

1. Line 64: The paper's result from SVRS beats that of SVRP by Khaled and Jin when $\delta \geq \sqrt{n} \mu$. As of now it's not clear to me what the scope of this assumption is: are there many cases where this inequality holds? It appears to me that Lines 124 and 198 allude to this but it's not entirely clear to me. Can the authors please elaborate?

2. Line 68 - 70: The paper's result with acceleration shaves a log factor from the previous best result as well as removes a component-wise strong convexity assumption from the previous best result. It would be nice to see some intuition for why the removal of log factor here is important and why the component-wise strong convexity is that much stronger than total strong convexity.


In the writing:

1. In lines 14 - 17, there are no references for these stated applications. It would be much more convincing to have citations for each of the stated applications.

2. The problem parameter $\mu$ should be introduced separately, rather than as part of the related work in lines 40 - 48, since it's an important parameter that comes up repeatedly.

3. The inclusion of Lemma $3.1$ in the main body does not serve (in my opinion) much purpose because it's mathematically quite dense to parse Equation $(10)$. Theorem $3.2$ is ok since it clearly gives the claimed rate. Instead of Lemma $3.1$, I'd have preferred to have a proof sketch for Theorem $3.2$.



**Questions:**

**Questions**


1. In Algorithm 1, should line 4 be $g_t = \nabla f_{i_t} (w_t) - \nabla f(w_t)$ and should there be an update rule for $w_t$ somewhere? (That's what Line 176 seems to suggest.) If not, it would be great to have intuition for why the same $w_0$ is used through the entire algorithm.

2. Please see the Weaknesses section above for more questions.

---

> ### Author Rebuttal · Authors · 2023-08-07
>
> Thank you for your advice. We reply to your comments one by one below.
>
> 1. Reply to Weaknesses in Results 1: We are sorry for the confusing argument. Indeed, SVRS is always no worse than SVRP by Khaled and Jin since
> $O(n+\delta^2/\mu^2)=O(n)$ and $O(n+\sqrt{n}\delta/\mu) = O(n)$ when $\delta<\sqrt{n}\mu$.
> We emphasize the case of $\delta \geq \sqrt{n}\mu$ to show the benefit of our method in the ill-conditioning case that $\mu$ is very small.
> 2. Reply to Weaknesses in Results 2: Thank you for your suggestions. Not only do we remove the log factor, but also we give a directly accelerated method.
> As mentioned in the paper of KatyushaX (see reference [6, Section 1.2] in our paper), the Catalyst SVRP needs to run each SVRP until a very accurate point is obtained since the error propagates.
> Moreover, to optimize the complexity, one needs to terminate each call of SVRP at a different accuracy, while this may be difficult for a random algorithm. Finally, one needs to tune three parameters in a Catalyzed method.
> Thus, we wonder whether a directly accelerated method exists. The contribution should not only be limited to removing the log factor (though it is also important from the poor theoretical view) but also the directly accelerated method.
> As for the improvements from the component-wise strong convexity to the total strong convexity, much literature has mentioned the reason. Overall, when we remove the component-wise strong convexity, then each component in the finite-sum objective could even be non-convex! This greatly expands the suitable scene. The most famous example is the shift-and-invert approach to solving PCA [1,2] (or see Sec 1.1 Motivating Examples in the paper of KatyushaX), where each component is smooth and non-convex, but the average function is convex. Thus, we consider our improvement meaningful.
> 3. Reply to Weaknesses in writing: Thank you for the advice, we will add references and adjust the structure in the later version.
> 4. Reply to Question 1: this is not a typo. Note that the Algorithm is one epoch (1ep in short) version of our SVRS. Thus, in the one epoch of SVRS, the anchor point is fixed just like SVRG. Line 176 shows the multi-epoch SVRS, where we update the anchor point $w_t$ based on 1ep SVRS.
>
>
> [1] Youcef Saad. Numerical methods for large eigenvalue problems. Manchester University Press, 1992.
>
> [2] Garber, Dan, et al. Faster eigenvector computation via shift-and-invert preconditioning. International Conference on Machine Learning. PMLR, 2016.

---

> ### Author Response · Authors · 2023-08-15
> **Looking Forward to Your Reply**
>
> Dear Reviewer Y65R,
>
> We understand that the review process can be time-consuming and demanding.
> We would greatly appreciate it if you could let us know whether you agree with our reply.

---

> > ### Comment · Reviewer_Y65R · 2023-08-18
> > **Acknowledgement of response**
> >
> > Dear authors,
> >
> > Thank you for your time and effort in a detailed rebuttal. I am keeping my score.
> >
> > Thanks!

---

### Official Review · Reviewer_ZtXX · 2023-07-06

**Soundness:** 3 good
**Presentation:** 2 fair
**Contribution:** 2 fair
**Rating:** 5
**Confidence:** 4

**Summary:**

The authors consider distributed minimization problems under data similarity (hessian similarity). The authors consider stochastic methods that reduce communication complexity via device sampling. In particular, from the stochastic point of view, the variance reduction techniques: SVRG and Katyusha, are taken. The sliding (stochastic preconditioning/mirror descent with unusual Bregman divergence) technique is used for dealing with similarity. The authors obtain record results in communication complexity (the previous ones were beaten by the logarithmic factor). To complete the picture, the authors provide lower bounds that give the optimality of their upper bounds. Synthetic experiments are also given.

**Strengths:**

1) Direct acceleration (without envelops) is an interesting and important result. In theory it removes the extra logarithmic factor, and in practice it works better.

2) The lower bounds complete the picture.



**Weaknesses:**

1) I think the literature review is not complete. In particular I find two papers also about the hessian similarity, which also use the variance reduction technique and obtain such non-accelerated results as the authors have. A detailed comparison in approaches and results is needed.

Beznosikov, A., & Gasnikov, A. (2023). Similarity, Compression and Local Steps: Three Pillars of Efficient Communications for Distributed Variational Inequalities. arXiv preprint arXiv:2302.07615.

Beznosikov, A., & Gasnikov, A. (2022, September). Compression and data similarity: Combination of two techniques for communication-efficient solving of distributed variational inequalities. In International Conference on Optimization and Applications (pp. 151-162). Cham: Springer Nature Switzerland.

2) The lower bounds are a good supplement to the upper bounds, but they are to be expected. The idea of getting them is also known. Unfortunately, here the authors  don not  also give a complete summary of the literature: the problem with a matrix A is classical (the authors note it), but the partition of the problem into columns is also classical - see the papers:

Zhang, M., Shu, Y., & He, K. (2020). Tight Lower Complexity Bounds for Strongly Convex Finite-Sum Optimization. arXiv preprint arXiv:2010.08766.

Han, Y., Xie, G., & Zhang, Z. (2021). Lower complexity bounds of finite-sum optimization problems: The results and construction. arXiv preprint arXiv:2103.08280.

Kovalev, D., Beznosikov, A., Sadiev, A., Persiianov, M., Richtárik, P., & Gasnikov, A. (2022). Optimal algorithms for decentralized stochastic variational inequalities. Advances in Neural Information Processing Systems, 35, 31073-31088.

3) Experiments for me are not the most important thing in this paper, which is primarily theoretical. But I would still like to see real datasets (3 for symmetry with synthetic data). I would also advise authors to change not \mu, but to change \delta in synthetic experiments, this way the effect of similarity will be noticeable.

Summary: For me, this is a borderline paper. For now I put a (weak) rejection, but I hope the authors will take part in discussion and make the changes I asked for.



**Questions:**

-

**Limitations:**

The paper is theoretical, therefore there is no need to discuss the social negative impact.

---

> ### Author Rebuttal · Authors · 2023-08-08
>
> Thank you for your review and for pointing out these interesting and meaningful references. We are sorry for the incompleteness of the reference.
> We will add the reference you posted in a later version because revision is not allowed in the rebuttal period this year.
> In addition, we find there are still some differences compared to these works.
> 1. Reply to Weakness 1: Despite the similar assumptions and techniques with similar communication bounds, we find that our paper still has some differences compared to Beznosikov, A., & Gasnikov, A. (2022, 2023):
> - We do not assume the smoothness and convexity of the component or the objective but replace it with a more general proximal approximately solvable assumption (see Eq. (9) in our main paper), which could even cover some nonsmooth and non-convex but proximal trackable component functions.
> We consider such an assumption more essential since the local update step in the paper of Beznosikov, A., & Gasnikov, A. (2022, 2023) can be viewed as partially solving the proximal step.
> - Our algorithm (Alg. 1) is more concise with easy-choosing parameters. Particularly, the choice of hyper-parameters, such as learning rate, is totally smoothness-free. This can also be viewed as the benefit of our proximal solvable assumption.
> - More importantly, except for the diversity in the non-accelerated method, we also give a simple directly accelerated method with a better communication complexity bound.
> However, Beznosikov, A., & Gasnikov, A. (2022, 2023) provide really meaningful work by considering a more general setting for solving variational inequalities, which could cover our simple minimization setup and many common problems.
> Moreover, they also adopt the famous compression technique to further reduce communication complexity.
> Due to these challenges, the algorithms introduced by Beznosikov, A., & Gasnikov, A. (2022, 2023) are more complex their ours.
>
> 2. Reply to Weakness 2: Our way of partitioning the matrix $A$ is inspired by Han, Y., et al. (2021) (as we claimed in Section 4) and indeed closely related to Kovalev, D., et al. (2022), but different from Zhang, M., et al. (2020), where the authors duplicate the matrix $n$ times instead of partitioning it. Moreover, our settings are different from these papers. Zhang, M., et al. (2020) and Han, Y., et al. (2021) both focus on the gradient complexity, while our concern is the communication complexity. Kovalev, D., et al. (2022) study the communication complexity (as well as the gradient complexity) for smooth variational inequalities, while we aim to give a more refined analysis of communication complexity for minimization problems even without the smoothness assumption, though we find that constructing a smooth hard instance suffices to yield the desired lower bound. Despite the similarity in construction, our results and theirs are not directly comparable.
>
> 3. Reply to Weakness 3: Since the communication bound in all literature is related to $\delta/\mu$, so we only need to change one parameter and fix another to see the effect. Meanwhile, adjusting the strong-convex coefficient $\mu$ is rather easy, but the similarity coefficient would change the dataset entirely.
> Hence, we adopt an easy-tuning way to conduct experiments for horizontal comparison of the different condition numbers with the same prefixed dataset.

---

> ### Author Response · Authors · 2023-08-15
> **Looking Forward to Your Reply**
>
> Dear Reviewer ZtXX,
>
> We understand that the review process can be time-consuming and demanding.
> We would greatly appreciate it if you could let us know whether you agree with our reply.

---

> > ### Comment · Reviewer_ZtXX · 2023-08-16
> >
> > Thanks to the authors for the response!
> >
> > I still recommend considering points 1 and 2 and reflecting them in the paper to understand the paper's place in the literature.
> >
> > The paper remains borderline for me, I will raise my score a bit (in hopes that the authors will consider points 1 and 2). The overall impression of the paper remains the same. The results about the algorithms are interesting (but the idea of using variance reduction is not new), the upper estimates are record-breaking (but only on the logarithmic factor), the lower bounds are a nice addition (but repeat the idea and technique of lower estimates for the unallocated finite sum).

---

> > > ### Author Response · Authors · 2023-08-16
> > > **Further Response to Reviewer ZtXX**
> > >
> > > Thank you for raising the score. We will definitely incorporate the points 1 and 2 into the revision.   Here we would again emphasize the differences between these nice works and our work.
> > >
> > > 1. First,  our work has large differences compared to Beznosikov \& Gasnikov (2022, 2023),  despite the similar assumptions and techniques.
> > >
> > > * We do not assume the smoothness and convexity of the component or the objective but replace it with a more general proximal approximately solvable assumption, which could even cover some nonsmooth and non-convex but proximal trackable component functions. We consider such an assumption more essential because the local update step in  Beznosikov \& Gasnikov (2022, 2023) can be viewed as partially solving the proximal step.
> > >
> > > * Our algorithm (Alg. 1) is more concise with easy-choosing parameters. Particularly, the choice of hyper-parameters, such as learning rate, is totally smoothness-free. This can also be viewed as the benefit of our proximal solvable assumption.
> > >
> > > * More importantly, except for the diversity in the non-accelerated method, we also give a simple directly accelerated method (Alg. 2) with an optimal communication complexity bound.
> > >
> > > *  Beznosikov \& Gasnikov (2022, 2023) provide really meaningful work by considering a more general setting for solving variational inequalities.
> > > They also adopt the famous compression technique to further reduce communication complexity.
> > > Nevertheless, despite that their setup is more complicated, our results are also valuable and not directly comparable to theirs.
> > > Although minimization problems, on which we focus, are just a subclass of variational inequalities,
> > > they are extremely attractive due to their simple structure. And we indeed fill a gap in the literature.
> > > Moreover, since minimization problems enjoy much better properties than variational inequalities, the optimal communication complexity bounds for the two kinds of problems are generally different.
> > >
> > > 2. Second, with regard to the lower bound  which makes our results complete,  our way of partitioning the matrix
> > > is inspired by Han, Xie, and Zhang (2021) (as we claimed in Section 4) and indeed closely related to Kovalev, D., et al. (2022), but different from Zhang, Shu, and He (2020), where the authors duplicate the matrix $n$ times instead of partitioning it. Moreover, our settings are different from these papers. Zhang, Shu, and He (2020) and Han, Xie, and Zhang (2021) both focus on the gradient complexity, while our concern is the communication complexity. Kovalev, D., et al. (2022) study the communication complexity (as well as the gradient complexity) for smooth variational inequalities, while we aim to give a more refined analysis of communication complexity for minimization problems even without the smoothness assumption, though we find that constructing a smooth hard instance suffices to yield the desired lower bound.
> > > As we mentioned in the first point, despite the similarity in construction, the optimal communication complexity bounds for the two kinds of problems are not directly comparable.

---

> > > > ### Comment · Reviewer_ZtXX · 2023-08-16
> > > >
> > > > I rather care about the commonalities with Beznosikov & Gasnikov (2022, 2023) rather than the differences. Most importantly, the two papers use the idea of variance reduction, as does the paper under review.
> > > >
> > > > "a more general proximal approximately solvable assumption" - questionable, I wouldn't say there is much generality, it's just different assumptions: smoothness vs. proximal friendliness. As the authors correctly pointed out there are proximal friendly non-smooth functions. But there are also smooth but not proximally friendly functions, such as logistic regression. Please, be honest and correct here when discussing these assumptions.
> > > >
> > > > "non-accelerated method" - absolute truth, if they had done an accelerated rate for the minimization problem, I wouldn't have given this work (the paper under review) a 5 ;-)
> > > >
> > > > Good luck with other rebuttals!

---

> > > > > ### Author Response · Authors · 2023-08-18
> > > > > **Further Response to Reviewer ZtXX 2**
> > > > >
> > > > > Thank you for your patience and time.
> > > > > We agree that smoothness vs. proximal friendliness are two different assumptions in general. However, note that we use $\|\|x- x_t\|\|^2$ as a proximal term (see Eq. (7) in Alg 1). So once $f_1$ is smooth (like in logistic regression), then our objective function in Eq. (7) is strongly convex and smooth. The resulting problem is approximately solvable by applying some accelerated algorithms, e.g., Nesterov's accelerated gradient. We would emphasize that we don't need a closed-form solution. Thus, our proximal friendliness assumption could cover the smoothness assumption in our case. Do you agree?

---

> > > > > > ### Comment · Reviewer_ZtXX · 2023-08-21
> > > > > >
> > > > > > Unfortunately not, if you compute the proximal operator inexactly, it introduces inexactness in further computations as well, this needs to be formally analyzed and the contribution of the inaccuracy of the computation studied. This is a rather complex issue, including from the point of view of local computations, what the local complexity will be, whether it will be optimal, and so on. Therefore, I suggest that the authors strongly modify the paper, taking into account everything I wrote above. Or write honestly about the problems of proximal-friendly assumption and analysis (e.g., "our results are not fully applicable to logistic regression, but if we assume that we solve the proximal operator inexactly, it works, but we didn't provide an honest analysis for it").

---

> > > > > > > ### Author Response · Authors · 2023-08-21
> > > > > > > **Further Response to Reviewer ZtXX 3**
> > > > > > >
> > > > > > > We would emphasize that computation complexity (or gradient complexity) and communication complexity
> > > > > > > are two different measurements of complexity.
> > > > > > > Your issues are related to computation complexity, while our main concern is communication complexity.
> > > > > > > We could honestly say that how to solve the proximal point problem in Eq.(7) would not affect the communication complexity at all because the update is at a local device.
> > > > > > > As long as we could approximately solve it up to the criterion Eq.(9), our optimal communication complexity bound does hold.

---

### Official Review · Reviewer_FHTz · 2023-07-15

**Soundness:** 3 good
**Presentation:** 3 good
**Contribution:** 2 fair
**Rating:** 5
**Confidence:** 1

**Summary:**

The paper presents a novel algorithm for distributed optimization, named Accelerated Stochastic Variance-Reduced Sliding (ASVRS). The authors focus on the problem of minimizing the average of a large number of smooth and strongly convex functions, a common scenario in machine learning and data analysis. The proposed ASVRS algorithm combines the techniques of gradient-sliding and variance reduction, aiming to improve the convergence rate and reduce the communication cost in distributed settings.

**Strengths:**

The ASVRS algorithm is a novel contribution that combines gradient-sliding and variance reduction techniques in a unique way. This combination appears to be original and innovative. The authors provide detailed proofs for the convergence rate and communication complexity of the ASVRS algorithm, demonstrating its theoretical advantages over existing methods. I didn't check the details but the result seems reasonable.


**Weaknesses:**

The theoretical analysis relies on several assumptions, such as the strong convexity of the functions. This makes the application of the work rather limited.  It would be helpful to discuss the implications of these assumptions and how the algorithm's performance might be affected if they are not met.

**Questions:**

How sensitive is the ASVRS algorithm to its parameters? Could you provide some guidance on how to set the parameters in practice.

**Limitations:**

The limitation has been adequately addressed.

---

> ### Author Rebuttal · Authors · 2023-08-06
>
> Thank you for your review and advice.
>
> The assumptions in our paper include 1) the finite-sum objective is strongly convex, 2) the finite-sum objective satisfies average second-order similarity, and 3) the proximal operator of just one part (or each part) is approximately solvable.
> These assumptions also appear in previous work (e.g., references [29, 31] in our paper).
> Assumption 1) is a common assumption in convex optimization, which could be satisfied after adding regularization for the common convex loss function. Assumption 2) is the core assumption in our setup, which appears in a large body of work, and has the practical scene from statistical learning, as well as the intuition of similar data in each client. Assumption 3) is related to the proximal operator, which also has much literature, particularly on nonsmooth optimization.
> Assumption 1) guarantees the benign property of the finite-sum objective, which serves as the whole property;
> Assumption 2) shows the connection between each component in the objective, which serves as the property between parts;
> Finally, Assumption 3) shows the requirement of one component (or each component), which serves as the part property.
>
> The hyper-parameters in ASVRS include interpolation coefficients $\tau, \alpha$, learning rate $\theta$, and full gradient step probability $p$. ASVRS is an accelerated method.
> Hence, ASVRS may display the oscillation when the learning rate is large and the interpolation coefficients are improper. Our theorems already give a proper combination of these parameters.
> However, in practice, the parameter in theory may not be optimal. Here we list one common tuning method based on our experience in experiments. We can choose $p=1/n$ and fix the relationship between interpolation coefficients $\tau$ and $\alpha$ as in Theorem 3.5, and then fine-tune the learning rate $\theta$ and only one interpolation coefficient $\tau \in (0, 1)$. In detail, we may scale $\theta$ and $\tau$ based on its theoretical value.

---

> ### Author Response · Authors · 2023-08-15
> **Looking Forward to Your Reply**
>
> Dear Reviewer FHTz,
>
> We understand that the review process can be time-consuming and demanding.
> We would greatly appreciate it if you could let us know whether you agree with our reply.

---

### Official Review · Reviewer_LQ8y · 2023-07-16

**Soundness:** 3 good
**Presentation:** 3 good
**Contribution:** 4 excellent
**Rating:** 8
**Confidence:** 3

**Summary:**

The paper considers distributed strongly convex optimization problems in the setting where the communication between nodes is bottleneck. The authors propose new methods, SVRS and AccSVRS, that guarantee new communication complexities. Also, they proved the lower bound that ensures the optimality of the AccSVRS method.

**Strengths:**

I think that the paper is strong. The authors provide new theoretical guarantees in the considered setting. They improve the previous methods. I haven't checked the proofs in detail and I can miss some essential parts, but the theory sounds to me.

**Weaknesses:**

It is well known that the (Loopless-)Katyusha method converges after $n + \sqrt{n \frac{L}{\mu}}$ iterations. By applying Katyusha to the authors' problem, we can get the communication complexity $n + \sqrt{n \frac{L}{\mu}},$ since Katyusha requires only one gradient in each iteration. In the regime when $\delta = L$ the Katyusha method has better communication complexity than $n + n^{3/4} \sqrt{\frac{L}{\mu}}.$ Why doesn't it contradict the lower bound (Theorem G.7 and Theorem 4.4)? Does it mean that your method is only better in the regimes when $\delta \ll L$?

Minor comments:

The paper's setup can be slightly confusing. Many other papers (e.g., \[1,2\]) assume that the nodes can do calculations and send vectors *in parallel,* meaning that they count each round as *one* communication, *one round = one communication*. In comparison, this paper assumes that *one round = $n$ communications* in the full participation regime. Can the authors write a small text *in the paper* explaining the difference between the setups? It seems that we have at least two different setups that both have the right to life.

Typos:

Eq. (18): $\nabla$ is missed.

\[1\]: https://arxiv.org/abs/2202.09357
\[2\]: https://arxiv.org/abs/2304.04169

**Questions:**

-

**Limitations:**

-

---

> ### Author Rebuttal · Authors · 2023-08-06
>
> Thank you for your review. We consider there are some misleading due to the unclearness in our paper.
> **Our setting is unable to be compared with the classical smooth and strongly convex setting generally.**
> First of all, we reclarify the framework we studied: 1) the finite-sum objective is $\mu$-strongly convex, 2) the finite-sum objective satisfies $\delta$-average second-order similarity, and 3) the proximal operator of just one part (or each part) is approximately solvable (e.g., Eq. (9) in our paper).
> Under assumptions 1), 2), and 3), we improve the communication complexity compared to the previous work.
> Thus, *the gradient complexity is not the main contribution.*
> Indeed, the computation complexity (i.e., gradient complexity) is *not available* in our main theorems since we do not know the gradient complexity of getting an approximate solution of the proximal operator without further assumptions.
> **Our lower bound is also in the framework under 1), 2) and 3)**, but we enhance 3) from approximately solvable into entirely solvable, i.e., we can obtain the solution of the proximal operator of each part function. This is fine since we consider the lower bound there.
> In summary, the main results in our paper are built on 1), 2), and 3), and are only related to the communication complexity.
> Moreover, since the gradient complexity is also important in the machine learning and optimization community, we turn to consider a more common setup by additionally assuming 4) each part (or just one part) of the finite-sum objective is $L$-smooth in Sec 3.3, which appears in many previous works (e.g., the references [29, 31] in our paper).
>
> Now we turn to answer the reviewer's questions.
> 1. Katyusha admits optimal gradient complexity under the assumption 1) and 4), which differs from our setup. So the direct comparison is unsuitable. However, if we study the setup under 1), 2), and 4), both algorithms could apply since 2) and 4) could guarantee 3) in our setup.
> At this time, we need to recognize that *our current method (AccSVRS) is not optimal in the gradient complexity generally*, meaning that our method could be worse than Katyusha, but when the coefficient $\delta$ in assumption 2) lies in the proper range: $\delta = \Theta(\sqrt{\mu L})$, our AccSVRS could nearly recover the optimal gradient complexity under assumptions 1) and 4) and is comparable to Katyusha.
> Finally, as we reply to the first reviewer, the discussion in Sec 3.3 only confirms that our algorithm could recover the optimal computation complexity of the classical (average-)smooth and strongly convex case *under some relationship between smoothness and similarity constants*. The discussion in Section 3.3 is to show the minor benefit of gradient complexity under 4) in some restricted cases, instead of affirming that our method is optimal in gradient complexity, since under Assumptions 1), 2), 3), gradient complexity is even not available.
> 2. Now we explain why the communication complexity of (Loopless-)Katyusha does not contradict our lower bounds. It is worth emphasizing that (Loopless-)Katyusha requires that each component is $L'$-smooth (we adopt a different symbol here to avoid ambiguity). Then the gradient complexity and the communication complexity of Katyusha are indeed the same, i.e., $n + \sqrt{n \frac{L'}{\mu} }$. Meanwhile, one can check that the hard instance constructed for our lower bound only satisfies that each component is $c \sqrt{n} \delta$-smooth for some constant $c>0$ (if necessary, we will add the detailed computation in the later version).  If $\delta = L$, applying Katyusha to our hard instance in fact yields the $n + n^{3/4} \sqrt{\frac{L}{\mu}}$ communication complexity instead of $n + \sqrt{ n \frac{L}{\mu} }$. This implies that for the special case with $L'  = \Theta( \sqrt{n} \delta )$, Katyusha attains the optimal communication complexity. As a comparison, the optimality of our method in terms of communication complexity does not require the smoothness of component functions or the relationship between the smoothness and similarity constants.
> 3. We thank the reviewer for pointing out the two setups in distributed optimization. We consider both setups important.
> We only list the case of our setting. For example, in a business network or communications network, the communication between any two nodes could produce the charge and the risk, which indeed could not be viewed as one communication in parallel at all.
> Moreover, the entire parallel is not available since the environment behind each node is roughly different and complex.
> However, if the environment is similar, the parallel is possible and could save the total time. We will add some explanation in a later version.

---

> > ### Comment · Reviewer_LQ8y · 2023-08-10
> > **Final decision**
> >
> > Thank you! I quickly went through the comment. It seems that my questions have been addressed. Thank you for the explanation. I will slightly increase the score. Good luck with other rebuttals.

---

### Official Review · Reviewer_pJR8 · 2023-07-29

**Soundness:** 3 good
**Presentation:** 3 good
**Contribution:** 3 good
**Rating:** 7
**Confidence:** 4

**Summary:**

This work considers finite-sum (distributed) optimization problems in the strongly convex and second-order similar regime. The authors proposed SVRS and its acceleration, AccSVRS, to solve the problem and provided the corresponding communication and computation complexities, which outperform existing works in several perspectives. The authors further characterized the lower bound for solving such problems, which validates the near optimality of the proposed AccSVRS algorithm.

**Strengths:**

1. The study is comprehensive in general, covering both upper and lower bounds
2. Proposed algorithm achieves near-optimal communication/computation complexity.
3. The design of the algorithm, which incorporates PPA, VR, gradient sliding and Katyusha, is interesting.
4. The paper is well-written, and the flow is clear.

**Weaknesses:**

1. The design of the AccSVRS, as the core algorithm achieving near optimality, can be further elaborated. For now, I do not have a clear understanding on Steps 5 and 6 in AccSVRS. Some discussions similar to Section 3.1 connecting Katyusha X paper will be appreciated.
2. Some more questions below.

**Questions:**

1. Line 232, you mentioned you recovered the optimal complexity of the average smooth setting. While with the additional AveSS setting, I may expect the complexity possibly can be (strictly) better than the classical $L$-smooth $\mu$-strongly convex case (assuming all components are $L$-smooth here for convenience).

   In your proof of Appendix E, you mentioned the obtained complexity is only smaller than that of the average smooth setting (Line 590), but should it be equivalence (rather than smaller only) here to be verified?

2. With the current result in Section 3.3, can we argue that possibly second-order similarity does not help in the gradient complexity of the classical case?

3. Line 591 in Appendix E (or Line 234), why do you drop the $O(n^{3/4}(L/\mu)^{1/4})$ term? Compared to the last $O(\sqrt{nL/\mu})$ term, is there a possibility that the $n^{3/4}$ term will dominate? Do I miss anything here?

---

> ### Author Rebuttal · Authors · 2023-08-06
>
> Thank you for your review and suggestions. Here we list the reply point by point.
>
> 1. Reply to Weakness 1: We are sorry for the unclearness due to the space limit of pages. We need to recognize that Steps 5 and 6 in AccSVRS are somehow tricky, so we only list some shallow understanding here.
> **The difference between AccSVRS and KatyushaX is due to the different choice of distance space.**
> Recapping the connection with Bregman-SVRG in Section 3.1, we adopt the reference function $f_1(\cdot)+\frac{1}{2\theta}||\cdot||^2$ to control $f(\cdot)$ instead of $\frac{1}{2\theta}||\cdot||^2$ in the original SVRG, leading to the distance space in our proof is produced by the Bregman divergence of $h(\cdot)=f_1(\cdot)+\frac{1}{2\theta}||\cdot||^2-f(\cdot)$ shown in Lemma 3.1 and thereafter.
> Now we turn back to AccSVRS, which is motivated by the general framework of KatyushaX (see (4.1) in its arxiv version: https://arxiv.org/pdf/1802.03866.pdf). Thus you can see that AccSVRS (Alg. 2) shares the same structure as KatyushaX.
> The main difference is that in our setup, the gradient mapping step for computing $\mathcal G_{k+1}$ is not based on the original standard norm $\frac{1}{2\theta}||\cdot||^2$, but the space produced by Bregman divergence $D_h(\cdot, \cdot)$.
> Hence, the gradient mapping should be $\nabla h(x_{k+1}) - \nabla h(y_{k+1})$ instead of $\frac{x_{k+1} -  y_{k+1}}{\theta}$.
> Next, noting that $\nabla h(x) = \nabla f_1(x) - \nabla f(x) + \frac{x}{\theta}$ could introduce the heavy gradient computing part $\nabla f(x)$, so we further adopt its stochastic version $\nabla f_1(x) - \nabla f_{j_k}(x) + \frac{x}{\theta}$ by uniformly sampling $j_k \sim \mathrm {Unif} ([n])$ to reduce the communication complexity.
> (P.S. Indeed, using $\nabla h(x)$ directly for computing $\mathcal G_{k+1}$ in AccSVRS is fine since the communication complexity of $\mathrm {SVRS^{1ep}}$ is $\Theta(n)$ when $p=1/n$. Here we employ a stochastic version to further reduce the communication complexity.)
> If the reviewer considers such an explanation helpful, we will add it to the later version.
> 2. Reply to Questions 1& 2: We consider there are some misleading in our description. **Our setting is unable to be compared with the classical smooth and strongly convex setting generally.**
> First, we reclarify the entire assumption of our problem: 1) the finite-sum objective is strongly convex, 2) the finite-sum objective satisfies average second-order similarity, and 3) the proximal operator of just one part (or each part) is approximately solvable, meaning that we can get an approximate solution of the proximal step (like Eq. (9) in our paper). And our main results are related to the communication complexity.
> As you can see, the computation complexity (i.e., gradient complexity) is not available in our main theorems since we do not know the gradient complexity of getting an approximate solution of the proximal operator without further assumptions.
> (P.S. Note that our lower bound is also in this framework, but we enhance 3) from approximately solvable to analytically solvable. This is fine since we consider the lower bound there.)
> Due to the importance of total computation in the machine learning and optimization community, we turn to consider a more common setup by assuming 4) each part (or just one part) of the finite-sum objective is smooth, which appears in many previous works (e.g., the references [29, 31] in our paper).
> The computation complexity in Sec 3.3 is a by-product of our results with a smaller function class due to the additional smoothness assumption 4). Meanwhile, compared to the classical smooth and strongly convex case, we further need to assume 2). Thus, we may wonder if the computation complexity in this stricter setting is *at least as good as* the optimal complexity excluding 2), since such a setting is well-studied.
> The discussion in Sec 3.3 only confirms that our algorithm could recover the optimal computation complexity of the classical (average-)smooth and strongly convex case *under some relationship between smoothness and similarity constant*.
> We do not know the optimal gradient complexity under 1), 2), and 4) up to now, but we are willing to see the relative work or try such a setting in the future.
> We agree with the reviewer's conjecture that the computation complexity may be better after assuming 2), but it is not able to conclude that second-order similarity does not help in the gradient complexity based on our paper.
> 3. Reply to Question 3: Thank you for your careful review even on the Appendix. Noting that $n+\sqrt{n L/\mu} \geq 2\sqrt{n \cdot \sqrt{n L/\mu}} = 2 n^{3/4}(L/\mu)^{1/4}$, we could drop the term $n^{3/4}(L/\mu)^{1/4}$ compared to the remaining term $n+\sqrt{n L/\mu}$ in the final complexity after hidding some constants.

---

> > ### Comment · Reviewer_pJR8 · 2023-08-14
> >
> > Thank you for the reply, my questions have been addressed. I will keep my score here.

---

### Decision · Program_Chairs · 2023-09-21

**Decision:**

Accept (poster)

**Comment:**

The paper presents and analyzes the SVRS and AccSVRS methods in the distributed finite-sum setting, under a second-order similarity condition. The techniques, based on gradient sliding and variance reduction, lead to rates which improve upon previous results, and these improvements are complemented by nearly matching lower bounds. While the use of variance reduction for these types of settings may not be entirely new, the results provide a meaningful contribution in terms of better communication complexity, and so acceptance is recommended.